# Quantum Theory of the Classical: Einselection, Envariance, Quantum Darwinism and Extantons

**DOI:** 10.3390/e24111520

**Published:** 2022-10-24

**Authors:** Wojciech Hubert Zurek

**Affiliations:** Theory Division, Mail Stop B213, LANL, Los Alamos, NM 87545, USA; whz@lanl.gov

**Keywords:** decoherence, einselection, quantum jumps, Born’s rule, envariance, quantum Darwinism, quantum-classical transition, existential interpretation, extantons

## Abstract

Core quantum postulates including the superposition principle and the unitarity of evolutions are natural and strikingly simple. I show that—when supplemented with a limited version of predictability (captured in the textbook accounts by the repeatability postulate)—these core postulates can account for all the symptoms of classicality. In particular, both objective classical reality and elusive information about reality arise, via quantum Darwinism, from the quantum substrate. This approach shares with the Relative State Interpretation of Everett the view that collapse of the wavepacket reflects perception of the state of the rest of the Universe *relative* to the state of observer’s records. However, our “let quantum be quantum” approach poses questions absent in Bohr’s Copenhagen Interpretation that relied on the preexisting classical domain. Thus, one is now forced to seek preferred, predictable, hence effectively classical but ultimately quantum states that allow observers keep reliable records. Without such *(i) preferred basis* relative states are simply “too relative”, and the ensuing *basis ambiguity* makes it difficult to identify events (e.g., measurement outcomes). Moreover, universal validity of quantum theory raises the issue of *(ii) the origin of Born’s rule*, $p_{k}={\left|\psi_{k}\right|}^{2}$, relating probabilities and amplitudes (that is simply postulated in textbooks). Last not least, even preferred pointer states (defined by *einselection*—*e*nvironment—*in*duced super*selection*)—are still quantum. Therefore, unlike classical states that exist objectively, quantum states of an individual system cannot be found out by an initially ignorant observer through direct measurement without being disrupted. So, to complete the ‘quantum theory of the classical’ one must identify *(iii) quantum origin of objective existence* and explain how the information about objectively existing states can appear to be essentially inconsequential for them (as it does for states in Newtonian physics) and yet matter in other settings (e.g., thermodynamics). I show how the mathematical structure of quantum theory supplemented by the only uncontroversial measurement postulate (that demands immediate repeatability—hence, predictability) leads to preferred states. These *(i) pointer states* correspond to measurement outcomes. Their stability is a prerequisite for objective existence of effectively classical states and for events such as quantum jumps. Events at hand, one can now enquire about their probability—the probability of a pointer state (or of a measurement record). I show that the symmetry of entangled states—*(ii) entanglement—assisted invariance* or *envariance*—implies Born’s rule. Envariance also accounts for the loss of phase coherence between pointer states. Thus, decoherence can be traced to symmetries of entanglement and understood without its usual tool—reduced density matrices. A simple and manifestly noncircular derivation of $p_{k}={\left|\psi_{k}\right|}^{2}$ follows. Monitoring of the system by its environment in course of decoherence typically leaves behind multiple copies of its pointer states in the environment. Only pointer states can survive decoherence and can spawn such plentiful information-theoretic progeny. This *(iii) quantum Darwinism* allows observers to use *environment as a witness*—to find out pointer states indirectly, leaving systems of interest untouched. Quantum Darwinism shows how epistemic and ontic (coexisting in *epiontic* quantum state) separate into robust objective existence of pointer states and detached information about them, giving rise to *extantons*—composite objects with system of interest in the core and multiple records of its pointer states in the halo comprising of environment subsystems (e.g., photons) which disseminates that information throughout the Universe.



**Contents**
 **1**      

**Introduction**


**3**

1.1      Core Quantum Postulates......................................................................................................................................................................................... 3
1.2      Quantum States, Information, and Existence......................................................................................................................................................... 5
1.3      Interpreting Relative States Interpretation............................................................................................................................................................. 7
1.4      Preview....................................................................................................................................................................................................................... 9 **2**      

**Quantum Jumps and Einselection from Information Flows and Predictability**


**10**

2.1      Repeatability and the Quantum Origin of Quantum Jumps.................................................................................................................................. 11
2.2      Mixed States of the “Target”...................................................................................................................................................................................... 13
2.3      Predictability Killed the (Schrödinger’s) Cat........................................................................................................................................................... 15
2.4      Records and Branches: Degenerate “Control”......................................................................................................................................................... 15
                          Repeatability and Actionable Information................................................................................................................................................17
2.5      Pointer Basis, Information Transfer, and Decoherence.......................................................................................................................................... 19
2.6      Irreversibility of Perceived Events, or “Don’t Blame the 2nd Law—Wavepacket Collapse Is Your Own Fault!”...........................................21
            2.6.1      Classical Measurement Can Be Reversed Even when Record of the Outcome is Kept.......................................................................21
            2.6.2      Quantum Measurement Can’t Be Reversed when the Record of the Outcome is Kept......................................................................22
2.7      Summary: Events, Irreversibility, and Perceptions................................................................................................................................................ 23 **3**      

**Born’s Rule from the Symmetries of Entanglement**


**24**

3.1      Envariance................................................................................................................................................................................................................... 26
3.2      Decoherence as a Result of Envariance.................................................................................................................................................................... 27
3.3      Swaps, Counterswaps, and Equiprobability............................................................................................................................................................ 28
3.4      Born’s Rule from Envariance..................................................................................................................................................................................... 31
            3.4.1      Additivity of Probabilities from Envariance.............................................................................................................................................33
            3.4.2      Algebra of Records as the Boolean Algebra of Events.............................................................................................................................35
3.5      Inverting Born’s Rule: Why Is the Amplitude a Square Root of the Frequency of Occurrence?....................................................................... 36
3.6      Relative Frequencies from Relative States............................................................................................................................................................... 39
3.7      Envariance—An Overview........................................................................................................................................................................................ 41
            3.7.1      Implications and the Scope of Envariance: Why Entanglement? Why Schmidt States?......................................................................41
            3.7.2      Towards the Experimental Verification of Envariance............................................................................................................................43 **4**      

**Quantum Darwinism**


**45**

4.1      Mutual Information, Redundancy, and Discord..................................................................................................................................................... 48
            4.1.1      Mutual Information.....................................................................................................................................................................................48
            4.1.2      Quantum Discord........................................................................................................................................................................................50
            4.1.3      Evidence and Its Redundancy....................................................................................................................................................................52
            4.1.4      Mutual Information, Pure Decoherence, and Branching States.............................................................................................................54
            4.1.5      Surplus Decoherence and Redundant Decoherence................................................................................................................................56
            4.1.6      Information Gained by Pure and Mixed Environments..........................................................................................................................57
            4.1.7      Environment as a Communication Channel.............................................................................................................................................58
            4.1.8      Quantum Darwinism and Amplification Channels.................................................................................................................................59
4.2      Quantum Darwinism in Action................................................................................................................................................................................. 59
            4.2.1       C-Nots and Qubits......................................................................................................................................................................................60
            4.2.2      Central Spin Decohered by Noninteracting Spins....................................................................................................................................61
            4.2.3      Quantum Darwinism in a Hazy Environment..........................................................................................................................................64
            4.2.4      Quantum Darwinism and Pointer States...................................................................................................................................................65
            4.2.5      Redundancy vs. Relaxation in the Central Spin Model...........................................................................................................................68
            4.2.6      Quantum Darwinism in Quantum Brownian Motion.............................................................................................................................71
            4.2.7      Huge Redundancy in Scattered Photons..................................................................................................................................................74
4.3      Experimental Tests of Quantum Darwinism........................................................................................................................................................... 78
4.4      Summary: Environment as an Amplification Channel........................................................................................................................................... 80 **5**      

**Quantum Darwinism and Objective Existence: Photohalos and Extantons**


**82**

5.1      Anatomy of an Extanton............................................................................................................................................................................................ 82
            5.1.1      Extantons and “The Classical”...................................................................................................................................................................83
            5.1.2      Photohalos, Photoextantons, and Information Detached from Existence.............................................................................................84
            5.1.3      Photohalos and the Quantum Origins of Irreversibility.........................................................................................................................85
5.2      Quantum Darwinism and the Existential Interpretation....................................................................................................................................... 85
5.3      From Quantum Core Postulates to Objective Classical Reality............................................................................................................................. 86
5.4      Extantons and the Existential Interpretation.......................................................................................................................................................... 87
5.5      Decoherence and Information Processing............................................................................................................................................................... 88
5.6      Quantum Darwinism and “Life as We Know It”.................................................................................................................................................... 89
                           Seeing Is Believing......................................................................................................................................................................................90
5.7      Bohr, Everett, and Wheeler........................................................................................................................................................................................ 90
5.8      Closing Remarks......................................................................................................................................................................................................... 92 

**References**


**94**



## 1. Introduction

Quantum mechanics is often regarded as an essentially probabilistic theory, where the random collapses of the wavepacket with probabilities governed by the rule conjectured by Max Born (1926) [1] play a central role. Yet, evolution dictated by the Schrödinger Equation is deterministic. This clash of quantum determinism of the unitary evolutions of the fundamental quantum theory with the quantum randomness of its phenomenological practice is at the heart of the interpretational controversies.

The aim of this review is to assess the progress made in the wake of the earlier developments (including in particular theory decoherence and einselection) since the beginning of this millennium. This includes the realization that selection of preferred states—einselection of the pointer states usually justified using decoherence—is a consequence of the tension between the linearity of quantum theory and the nonlinearity of copying processes involved in the acquisition of information. Derivation of Born’s rule based on the symmetries of entangled quantum states shores up and simplifies foundations of quantum theory.

Quantum Darwinism will be discussed especially carefully, but nevertheless with significant omissions that are inevitable in reviewing a rapidly evolving field. In such a case one is faced with a “moving target”—the most recent developments are inevitably either left out or treated only in the superficial manner (since assessing their impact on the future development of the field is difficult).

We will also reconsider the status of the quantum measurement problem [2]. I shall claim that perception of the objective classical reality is accounted for by the developments mentioned briefly above and discussed in more detail below.

We shall start by reviewing the assumptions—postulates of quantum theory—and by selecting from their textbook version core postulates that are consistent and can be used to address the issues usually dealt with via measurement axioms that are also included in the textbook presentations but are inconsistent with the quantum core. More detailed preview of the content of this review can be found at the end of this introductory section.

### 1.1. Core Quantum Postulates

The difficulty of reconciling quantum determinism with quantum randomness is reflected in the postulates that provide textbook summary of quantum mechanics (see, e.g., Dirac, 1958) [3]. We list them starting with four uncontroversial core postulates, cornerstones of the *quantum theory of the classical* we shall develop. Two are very familiar:

(i) *The state of a quantum system is represented by a vector in its Hilbert space* $H_{S}$.

(ii) *Evolutions are unitary (i.e., generated by the Schrödinger Equation).*

They imply, respectively, the *quantum superposition principle* and the *unitarity of evolutions*, and we shall often refer to them by citing their physical consequences. They provide an almost complete summary of the formal structure of the theory.

One more postulate should be added to (i) and (ii) to complete the mathematics of quantum mechanics:

(o) *Quantum state of a composite system is a vector in a tensor product of the Hilbert spaces of its subsystems.*

Postulate (o) (von Neumann, 1932 [4]; Nielsen and Chuang, 2000 [5]) is often omitted from textbooks as obvious. However, composite systems are essential, as in absence of subsystems Schrödinger Equation provides a *deterministic* description of the evolution of an indivisible Universe, and the measurement problem disappears [6,7]. In absence of at least a measured system and a measuring apparatus questions about the outcomes cannot be even posed. We shall need at least one more ingredient—an environment—to address them.

The measurement problem arises because a quantum state of a collection of systems can evolve from a Cartesian product (where definite state of the whole implies definite states of each subsystem) into an entangled state represented by tensor product: State of the whole is still definite and pure, but states of the subsystems are indefinite. By contrast, in classical settings completely known (pure) composite states are always represented by Cartesian products of pure states—state of each subsystem is also perfectly known.

Postulates (o)–(ii) provide a complete summary of the *mathematics* of quantum theory. They contain no premonition of either collapse or probabilities. Using them and the obvious additional ingredients (initial states and Hamiltonians) one can justify and carry out every quantum calculation. However, in order to relate quantum theory to experiments one needs to establish a correspondence between abstract state vectors in $H_{S}$ and experiments. This task starts with the *repeatability postulate*:

(iii) *Immediate repetition of a measurement yields the same outcome.*

Postulate (iii) is idealized—it is hard to perform such non-demolition measurements, but in principle it can be done. Yet—as a fundamental postulate—it is also indispensable. The very concept of a “state” embodies predictability that requires axiom (iii): The role of states is to allow for predictions, and the most basic prediction is that a state is what it is known to be. Repeatability postulate asserts that confirmation of this prediction is in principle possible.

Postulate (iii) is also uncontroversial: Repeatability is taken for granted in the classical setting where it follows from the assumption that one can find out an unknown state without perturbing it. This classical version is a much stronger assumption than the repeatability postulated above in (iii). It is responsible for the familiar “objective reality” of the classical world: It detaches existence of classical states from what is known about them.

Quantum measurement problem arises because—by contrast—unknown quantum states are re-prepared by the attempts to find out what they are. So, quantum repeatability postulate (iii) signals a significant weakening of the role states play in our quantum Universe: Repeatability guarantees only that the existence of a *known* quantum state can be confirmed, but it no longer implies their objective existence: Unlike classical states, unknown quantum state cannot be simply found out independently by many initially ignorant observers through direct measurements.

This quantum intertwining of the epistemic and ontic function of a state is the central quantum feature regarded as a key interpretational problem. One of our goals is to understand how (as a consequence of quantum Darwinism) one can recover objective existence—states that survive discovery by an initially ignorant observer, so others can confirm their identity.

We will show that the essence of the remaining textbook postulates can be deduced from the above quantum core that includes the mathematical postulates (o)–(ii) and the repeatability postulate (iii) that begins to deal with the experimental consequences of quantum theory such as information transfers, including the measurements.

### 1.2. Quantum States, Information, and Existence

So far, we have outlined a consistent set of core quantum postulates, (o)–(iii). They will serve as a basis for the derivation of the emergence of classical behavior in a quantum Universe. In this subsection, we consider textbook axioms (iv) and (v) that are at odds with the quantum core. The whole (o)–(v) list is, of course, given by textbooks. The inconsistency is usually “resolved” through some version of Bohr’s strategy. That is, textbooks assume that quantum theory can be applied only to a part of the Universe. The rest of the Universe—including observers and measuring devices—must be classical, or at the very least out of quantum jurisdiction. Our aim will be to show that the classical domain need not be postulated, and that the measurement process (the focus of axioms (iv) and (v)) can be accounted for by using the quantum core postulates (o)–(iii).

In contrast to classical physics (where an unknown preexisting state can be found out by an initially ignorant observer) the very next textbook axiom explicitly limits predictive attributes of quantum states:

(iv) *Measurement outcome is an eigenstate of the Hermitian operator corresponding to the measured observable.*

Thus, in general, a measurement will return something else than the preexisting state of the system. Repeatability postulate (iii) is in a sense an exception to this quantum undermining of the predictive role of states. Axiom (iv) can be usefully subdivided into:

(iva) *Allowed measurement outcomes correspond to the eigenstates of a Hermitian operator.*

(ivb) *Only one outcome is seen in each run.*

This splitting may seem pedantic, but it is useful. Textbooks often separate our (iv) into such two axioms.

We emphasize that already (iva) limits predictive attributes of quantum states: When the Hermitian operator representing the measured observable does not have, as one of its eigenstates, the preexisting state of the system, the outcome cannot be predicted with certainty even when the preexisting state is perfectly known (pure).

Nevertheless, repeatability means that when the same measurement is immediately repeated on the very same system, the outcome will be the same. This is, operationally, the essence of the collapse: The preexisting pure state will give an unpredictable result that can be, however, confirmed and reconfirmed by re-measurement of the outcome. What you saw you will get, again and again. Therefore, as soon as (iva) can be accounted for (which we shall do in Section 2), then—in combination with the repeatability of (iii)—the symptoms of the “wavepacket collapse” postulated by (ivb) can be also recovered.

*Collapse axiom* is the first truly controversial item in the textbook list. In its literal form it is inconsistent with the first two postulates: Starting from a general state $|\psi_{S}\rangle$ in a Hilbert space of the system (postulate (i)), an initial state $|A_{0}\rangle$ of the apparatus A, and assuming unitary evolution (postulate (ii)) one is led to a superposition of outcomes;
(1.1)$$|\psi_{S}\rangle |A_{0}\rangle =\left(\sum_{k}a_{k}|s_{k}\rangle \right)|A_{0}\rangle \Rightarrow \sum_{k}a_{k}|s_{k}\rangle |A_{k}\rangle ,$$
which is in apparent contradiction with (iv).

The impossibility to account—starting with the core quantum postulates (o)–(iii)—for the literal collapse to a single state postulated by (ivb) was appreciated since Bohr (1928) [8] and von Neumann (1932) [4]. It was—and often still is—regarded as an indication of the insolubility of the measurement problem. It is straightforward to extend such insolubility demonstrations to various more realistic situations, e.g., by allowing the state of the apparatus to be initially mixed. As long as the superposition and unitarity postulates (i) and (ii) hold, one is forced to admit that the quantum state of AS after they interacted contains a superposition of many alternative outcomes rather than just one of them as the literal reading of the collapse axiom (and our immediate experience) suggest (see Figure 1).

Given this clash between the mathematical structure of the theory and the expectation of the literal collapse (that captures the subjective impressions of what happens in the real-world measurements), one is tempted to accept—following Bohr—primacy of our immediate experience and blame the inconsistency of (iv) with the core of quantum formalism (superposition principle and unitarity, (i) and (ii)) on the nature of the apparatus: Copenhagen Interpretation regards apparatus, observer, and, generally, macroscopic objects as *ab initio* classical. They do not abide by the quantum principle of superposition—their evolutions need not be unitary. Therefore, according to Copenhagen Interpretation, the unitarity postulate (ii) does not apply to measurements, and the literal collapse can happen on the border between quantum and classical.

Uneasy coexistence of the quantum and the classical postulated by Bohr is a challenge to the unification instinct of physicists. Yet, it has proven surprisingly durable.

At the heart of many approaches to the measurement problem is the desire to reduce the relation between existence and information about what exists to what could have been taken for granted in a world where the fundamental theory was Newtonian physics. There, classical systems had real states that existed independently of what was known about them. They could be found out by measurements. Many initially ignorant observers could measure the same system without perturbing it. Their records would agree, reflecting reality of the underlying state and confirming its objective existence.

Immunity of classical states to measurements suggested that, in classical settings, the information was unphysical. Information was a mere immaterial shadow of real physical states. It was irrelevant for physics.

This dismissive view of information run into problems already when Newtonian classical physics confronted classical thermodynamics. Clash of these two classical theories led to Maxwell’s demon, and is implicated in the origins of the arrow of time.

The specter of information was haunting classical physics since XIX century. The seemingly unphysical shadowy record state was beginning to play a role reserved for the “real” state.

Attempts to solve measurement problem often follow the strategy where the underlying state of the quantum system somehow becomes classical. Even decoherence can be, in a sense, regarded as a completely quantum version of such a strategy, with the effective classicality arising in the world that is fundamentally quantum. Other proposals assert supremacy of existence over information and suggest modifications of quantum evolution equations (e.g., abandoning unitarity) as discussed by Weinberg (2012) [9].

It is conceivable that, one day, we may find discrepancies of quantum theory with experiments. However, evidence to date supports view that our Universe is quantum to the core, and we have to reconcile superposition principle, unitarity and their consequences—illustrated, e.g., by the violation of Bell’s inequality—with our perceptions. Nonlocality of quantum states and other experimental manifestations of quantumness are here to stay.

The strategy adopted by the program discussed in this review is to start with the core quantum postulates (o)–(iii). They have the simplicity that rivals postulates of special relativity. Given this “let quantum be quantum” starting point we shall show how (and to what extent) both attributes of the familiar classical world—objective existence and information about it—emerge from the epiontic quantum substrate.

### 1.3. Interpreting Relative States Interpretation

The alternative to Bohr’s Copenhagen Interpretation and a new approach to the measurement problem was proposed by Hugh Everett III, student of John Archibald Wheeler, over half a century ago (Everett, 1957 [10,11]; Wheeler, 1957 [12]; DeWitt and Graham, 1973 [13]). The basic idea was to abandon the literal view of collapse and recognize that a measurement (including the appearance of the collapse) is already implicit in Equation (1.1). One just needs to include an observer in the wavefunction, and consistently interpret the consequences of this step.

The obvious problem raised by (ivb)—“Why don’t I, the observer, perceive such splitting, but register just one outcome at a time?”—is then answered by asserting that while the right-hand side of Equation (1.1) contains all the possible outcomes, the observer who recorded outcome #17 will (from then on) perceive “branch #17” that is consistent with the outcome reflected in his records. In other words, when the global state of the Universe is $|\Upsilon \rangle$, and my state is $|I_{17}\rangle$, for me the state of the rest of the Universe collapses to $|\gamma_{17}\rangle ∼\langle I_{17}|\Upsilon \rangle$. Since this is the only state I (actually, $|I_{17}\rangle$!) am aware of, following the correlation, I should renormalize the state vector $|\gamma_{17}\rangle$ of the rest of the Universe to reflect my certainty about my branch—this is now my only Universe[note 1].

This “let quantum be quantum” view of the collapse is supported by the repeatability postulate (iii); upon immediate re-measurement, the same state will be found. Everett’s assertion: “The discontinuous jump into an eigenstate is thus only a relative proposition, dependent on the mode of decomposition of the total wave function into the superposition, and relative to a particularly chosen apparatus-coordinate value...”. is consistent with the core quantum postulates: In the superposition of Equation (1.1) record state $|A_{17}\rangle$ can indeed imply detection of the corresponding state of the system, $|s_{17}\rangle$.

Two questions immediately arise. The first one concerns the part (iva) of the collapse postulate: What constrains the set of outcomes—the preferred states of the apparatus or the observer. By the principle of superposition (postulate (i)) the state of the system or of the apparatus after the measurement can be written in infinitely many ways, each corresponding to one of the unitarily equivalent bases in the Hilbert space of the pointer of an apparatus (or a memory cell of an observer);
(1.2)$$\sum_{k}a_{k}|s_{k}\rangle |A_{k}\rangle =\sum_{k}a_{k}^{\prime }|s_{k}^{\prime }\rangle |A_{k}^{\prime }\rangle =\sum_{k}a_{k}^{\prime \prime }|s_{k}^{\prime \prime }\rangle |A_{k}^{\prime \prime }\rangle =...$$
This *basis ambiguity* is not limited to the pointers of measuring devices or cats, which for Schrödinger (1935) [20] play a role of the apparatus (see Figure 1). One can show that also very large systems (such as satellites of planets) can evolve into very nonclassical superpositions on surprisingly short timescales [21,22,23]. In reality, this does not seem to happen. So, there is something that (in spite of the egalitarian superposition principle enshrined in (i)) picks out certain preferred quantum states, and makes them effectively classical while banishing their superpositions.

Postulate (iva) anticipates this need for preferred states—destinations for quantum jumps: Before there is a collapse (as in (ivb)), a set of preferred states (one of which is selected by the collapse) must be somehow chosen. Indeed, discontinuity of quantum jumps Everett emphasizes in the quote above would be impossible without some underlying discontinuity in the set of the possible choices. Yet, there is nothing in Everett’s writings that would provide a criterion for such preferred outcomes states, and nothing to even hint that he was aware of this question. We shall show how such discontinuities arise in the framework defined by the core quantum postulates (o)–(iii).

The second question concerns probabilities: How likely it is that—after I, the observer, measure S—I will become $|I_{17}\rangle$? Everett was very aware of its significance.

*The preferred basis problem* was settled by the *pointer basis* that is singled out by the environment—induced superselection (*einselection*), a consequence of decoherence (Zurek, 1981; 1982 [24,25]).

As emphasized by Dieter Zeh (1970) [26], apparatus, observers, and other macroscopic objects are immersed in their environments. The problem of preferred basis was not pointed out at that time, perhaps because this issue is never pointed out by Everett which motivated Zeh’s paper. Indeed, it appears Everettians, (e.g., DeWitt, [14,15]) did not fully appreciate its importance until the advent of the pointer basis.

Decoherence leads to monitoring of the system by its environment, described by analogy with Equation (1.1). When this monitoring is focused on a specific observable of the system, its eigenstates form a *pointer basis*: They entangle least with the environment (and, therefore, are least perturbed by it). This resolves basis ambiguity. Pointer basis and einselection [24,25] were developed and are discussed elsewhere [6,7,24,25,27,28,29,30,31,32,33]. However, their original derivation comes at a price that would have been unacceptable to Everett: Theory of decoherence, as it is usually practiced, employs reduced density matrices. Their physical significance derives from averaging (Landau, 1927 [34]; Nielsen and Chuang, 2000 [5]; Zurek 2003 [35]) and is thus based on probabilities that follow from Born’s rule:

(v) *Probability $p_{k}$ of finding an outcome $|s_{k}\rangle$ in a measurement of a quantum system that was previously prepared in the state $|\psi \rangle$ is given by $|\langle s_{k}|\psi \rangle {|}^{2}$*.

Born’s rule (1926) [1] completes standard textbook discussions of the foundations of quantum theory. In contrast to the wavepacket collapse of axiom (iv), axiom (v) is not in obvious contradiction with the core postulates (o)–(iii), so one can adopt the view that Born’s rule is a part of the axiomatics of quantum theory. One can then use core postulates (o)–(iii) plus Born’s rule to justify preferred basis and explain the symptoms of collapse through decoherence and einselection. This is the usual practice of decoherence (Zurek, 1991 [27]; 1998 [36]; 2003 [7]; Paz and Zurek, 2001 [28]; Joos et al., 2003 [29]; Schlosshauer, 2005 [31]; 2006 [37]; 2007 [32]; 2019 [33]). It relies, however, on the statistical interpretation of the reduced density matrices that depends on accepting Born’s rule.

Nevertheless, (as Everett argued) axiom (v) is inconsistent with the spirit of the “let quantum be quantum” approach. Therefore, one might guess, he would not have been satisfied with the usual approach to decoherence and its consequences. Indeed, Everett attempted to derive Born’s rule from the other quantum postulates. We shall follow his lead, although not his strategy which—as is now known—was flawed (DeWitt, 1971 [15]; Kent, 1990 [38]; Squires, 1990 [39]).

### 1.4. Preview

Our first goal is to shore up quantum foundations—to understand the emergence of stable classical states from the quantum substrate, and to deduce the origin of the rules governing randomness at the quantum-classical border. To this end, in the next two sections we shall derive collapse axiom (iva) and Born’s rule (v) from the core postulates (o)–(iii). We shall then, in Section 4, account for the “objective existence” of pointer states. This succession of results provides a wholly quantum account of the emergence of classical reality.

We start with a derivation of the preferred set of *pointer states*—(iva), the business end of the collapse postulate. We will show that the nature of the information transfer—nature of the coupling to the measuring device—determines this preferred set, and that any set of orthogonal states will do. We will also see how these states are (ein)selected by the dynamics of the process of information acquisition, thus following the spirit of Bohr’s approach which emphasized the ability to communicate the results of measurements. Orthogonality of outcomes implies that repeatedly measurable quantum observable must be Hermitian. We shall then compare this approach (obtained without resorting to reduced density matrices or any other appeals to Born’s rule) with a decoherence-based approach to pointer states and the usual view of einselection.

Pointer states—terminal states for quantum jumps—are determined by the dynamics of information transfer. They define the outcomes independently of the instantaneous reduced density matrix of the system and of its initial state. Fixed outcomes define events, and call for the derivation of probabilities. In Section 3 we also take a fresh and very fundamental look at decoherence: It arises—along with Born’s rule—from the symmetries of entangled quantum states.

Given Born’s rule and preferred pointer states one is still faced with a problem. Quantum states are fragile. An initially ignorant observer cannot find out an unknown quantum state without endangering its existence: Collapse postulate means that selection of what to measure implies a set of outcomes. Therefore, only a lucky guess of an observable could let the observer find out an unknown state without repreparing it. The criterion for pointer states implied by postulates (o)–(iii) turns out to be equivalent to their stability under decoherence, and still leaves one with the same difficulty: How to find out an effectively classical but ultimately quantum pointer state and leave it intact?

The answer turns out to be surprisingly simple: Continuous monitoring of S by its environment results in redundant records of its pointer states in E. Thus, observers can find out the state of the system indirectly, from small fragments of the same E that caused decoherence. Recent and still ongoing studies discussed in Section 4 show how this replication selects the “fittest” states that can survive monitoring, and yields copious qmemes[note 2], their information-theoretic offspring: Quantum Darwinism favors pointer observables at the expense of their complements. Objectivity of the preferred states is quantified by their redundancy—by the number of copies of the state of the system deposited in E. Stability in spite of the interaction with the environment is clearly a prerequisite for large redundancy. Pointer states do best in this information—theoretic “survival of the fittest”.

The classical world we perceive consists predominantly of macroscopic objects. Bohr decreed their states were classical “by fiat”, so that information about them could be acquired without perturbing them, thus restoring classical independence of existence from information. We recognize instead that quantum theory is universal. States of macroscopic objects become effectively classical (as Bohr wanted), but as a consequence of decoherence and einselection. Objects are immersed in the decohering environment consisting of subsystems (such as photons). Superpositions of pointer states are unstable, quickly turning into their mixtures. Thus, predictably evolving quantum states of macroscopic objects are restricted to stable, effectively classical pointer states einselected by decoherence. In the course of decoherence fragments of the environment that monitors them become inscribed with the data about their pointer states.

*Extanton* is a composite entity with the object of interest in its core embedded in the information-laden halo, part of the environment that monitors its pointer states. Information about them is heralded by the fragments of the environment, and disseminated throughout the Universe. Fragments of the halo intercepted by observers inform about the state of its core. Extanton combines the source of information (extanton core) with the means of its transmission (halo, often consisting of photons).

John Bell (1975 [40]; 1987 [41]) imagined “beables” (as in “to be or not to be”). In contrast to observables, beables were supposed to be robust, much like states of macroscopic objects in the classical domain posited by Bohr. They would exist, and (in contrast to quantum states), their states would be immune to observation.

Extantons are quantum, but fulfill these desiderata. Environment determines pointer states through einselection. Pointer states of extanton cores persist (hence, exist) and the environment broadcasts information about them. That information reaches observers, revealing the pointer state of the macroscopic system at the extanton core without the need for direct measurement (hence, without disrupting the state preselected by the decoherence).

We are immersed in such extaton halos, inundated with the information about pointer states in their cores. This is how the classical world we perceive emerges from within our quantum Universe.

As we shall see, several steps based on interdependent insights are needed to account for quantum jumps, for the appearance of the collapse, for preferred pointer states, for the probabilities and Born’s rule, and, finally, for the consensus, the essence of objective reality—for the emergence of ‘the classical’ from within a quantum Universe. It is important to take these steps in the right order, so that each step is based only on what is already established. This is our aim, and this order has determined the structure of this paper: The next three sections describe three crucial steps. Nevertheless, each section can be read separately: Preceding sections are important to provide the right setting, but are generally not essential as a background. An overview of the resulting quantum theory of the classical is presented and the interpretational implications are discussed in Section 5.2.

## 2. Quantum Jumps and Einselection from Information Flows and Predictability

This section shows how the core quantum postulates (o)–(iii) lead to the discreteness we regard as characteristic of the quantum world. In textbooks this discreteness is introduced via the collapse axiom (iva) designating the eigenstates of the measured observable as the only possible outcomes. Here, we show that discontinuous quantum jumps between a restricted set of orthogonal states turn out to be a consequence of symmetry breaking that resolves the tension between the unitarity of quantum evolutions and repeatability. We shall also see how preferred Hermitian observables defined by the resulting orthogonal basis are related to the familiar pointer states.

Unitary evolution of a general initial state of a system S interacting with an apparatus A leads—as illustrated by Equation (1.1)—to an entangled state of SA. Thus, there is no single outcome—no literal collapse—and an apparent contradiction with our immediate experience. It may seem that the measurement problem cannot be addressed unless unitarity is somehow circumvented (e.g., along the *ad hoc* lines of the Copenhagen Interpretation).

We start with the same assumptions and follow similar steps, but arrive at a different conclusion. This is because instead of demanding a single outcome we shall only require that the result of the measurement can be confirmed (by a re-measurement), or communicated (by making a copy of the record). In either case, copying some state (of the system or of the apparatus) is essential. As “perception” and “consciousness” presumably depend on copying and other such information processing tasks (as they undoubtedly do) then the necessity to deal with the Universe “one branch at a time” can produce symptoms of collapse while bypassing the need for it to be “literal”.

Amplification—the ability to make copies, qmemes of the original—is the essence of the repeatability postulate (iii). It calls for nonlinearity (one needs to replicate the original state, or at least its salient features) that would appear to be in conflict with the unitarity (hence, linearity) demanded by postulate (ii).

As we shall see, copying is possible for orthogonal subsets of states of the original. Each such subset is determined by the measurement device—by the unitary evolution that implements copying. When, beforehand, the system is not in one of such copying eigenstates, its state is not preserved. This shows (Zurek, 2007 [42]; 2013 [43]) why one cannot find out an unknown quantum state. Most importantly, we reach this conclusion (where the role of the copying device parallels function of the classical apparatus in Bohr’s Copenhagen Interpretation) without calling on the collapse axiom (iv) or on Born’s rule, axiom (v).

### 2.1. Repeatability and the Quantum Origin of Quantum Jumps

Consider a quantum system S interacting with another quantum system E (which can be an apparatus, or—as the present notation suggests—an environment). Let us suppose (in accord with the repeatability postulate (iii)) that there are states of S that remain unperturbed by this operation, e.g., that this interaction implements a measurement—like information transfer from S to E:(2.1)$$|s_{k}\rangle |\varepsilon_{0}\rangle \;⟹\;|s_{k}\rangle |\varepsilon_{k}\rangle .$$

We now establish:

**Theorem** **1.**
*The set of the unperturbed states $\left\{|s_{k}\rangle \right\}$ of the “control”—of the system S that is being measured or decohered—must be orthogonal.*


**Proof.** From the linearity implied by the unitarity of (ii) and Equation (2.1) we get, for an arbitrary initial state $|\psi_{S}\rangle$ in $H_{S}$ (allowed by the superposition principle, postulate (i));
(2.2)$$|\psi_{S}\rangle |\varepsilon_{0}\rangle =\left(\sum_{k}\alpha_{k}|s_{k}\rangle \right)|\varepsilon_{0}\rangle \Rightarrow \sum_{k}\alpha_{k}|s_{k}\rangle |\varepsilon_{k}\rangle =|\Psi_{SE}\rangle .$$
But, again by (ii), the norm must be preserved,
$$|\sum_{k}\alpha_{k}|s_{k}\rangle {|}^{2}={\left|\sum_{k}\alpha_{k}|s_{k}\rangle |\varepsilon_{k}\rangle \right|}^{2},$$
so that elementary algebra leads to:
(2.3)$$Re\sum_{j,k}\alpha_{j}^{*}\alpha_{k}\langle s_{j}|s_{k}\rangle =Re\sum_{j,k}\alpha_{j}^{*}\alpha_{k}\langle s_{j}|s_{k}\rangle \langle \varepsilon_{j}|\varepsilon_{k}\rangle .$$
This must hold for every $|\psi_{S}\rangle$ in $H_{S}$—for any set of complex $\left\{\alpha_{k}\right\}$. Thus, for any two states in the set $\left\{|s_{k}\rangle \right\}$:
(2.4)$$\langle s_{j}|s_{k}\rangle \left(1-\langle \varepsilon_{j}|\varepsilon_{k}\rangle \right)\;=\;0.$$
This equality immediately implies that $\left\{|s_{k}\rangle \right\}$ must be orthogonal if they are to leave any imprint—deposit any information—in E while remaining intact: It can be satisfied only when $\langle s_{j}|s_{k}\rangle =\delta_{jk}$, unless $\langle \varepsilon_{j}|\varepsilon_{k}\rangle =1$—that is, unless the information transfer has failed, as $|\varepsilon_{j}\rangle =|\varepsilon_{k}\rangle$—the states of E bear no imprint of the states of S.    □

Equation (2.4) establishes postulate (iva)—the orthogonality of the outcome states (i.e., of the “originals” of the copying eigenstates). As we have noted, (iva) is the essence, the “business end” of the collapse axiom (iv). When the outcome states are orthogonal, any value of $\langle \varepsilon_{j}|\varepsilon_{k}\rangle$ is admitted, including $\langle \varepsilon_{j}|\varepsilon_{k}\rangle =0$, which corresponds to a perfect record.

Note that—as long as the state, Equation (2.1) is a direct product before and after the measurement—this conclusion holds for an arbitrary initial state of E, since Equation (2.4) demands orthogonality whenever there is any transfer of information from S to E—that is, whenever $\langle \varepsilon_{j}|\varepsilon_{k}\rangle \neq 1$. It is of course possible that there are subsets of orthogonal states that cannot be distinguished by the environment. We shall consider such degeneracy shortly.

The limitation of copying to distinguishable (orthogonal) outcome states is then a direct consequence of the uncontroversial core postulates (o)–(iii). It can be seen as a resolution of the tension between linearity of quantum theory (superpositions and unitarity of (i) and (ii)) and nonlinearity of the process of proliferation of information—of amplification. This nonlinearity is especially obvious in cloning, as cloning in effect demands “two of the same”. The main difference is that in cloning copies must be perfect. Therefore, scalar products must be the same, $\varsigma_{j,k}=\langle \varepsilon_{j}|\varepsilon_{k}\rangle =\langle s_{j}|s_{k}\rangle$. Consequently, in cloning we have a special case of Equation (2.4): $\varsigma_{j,k}\left(1-\varsigma_{j,k}\right)=0.$ Clearly, there are only two possible solutions; $\varsigma_{j,k}=0$ (which implies orthogonality), or the trivial $\varsigma_{j,k}=1$.

Indeed, we can deduce orthogonality of states that remain unperturbed while leaving small but distinct imprints in E directly from the no-cloning theorem [44,45,46]) that limits copying allowing it for orthogonal sets of states (thus precluding use of entangled quantum states for superluminal communication): As the states of S remain unperturbed by assumption, arbitrarily many imperfect copies can be made. However, each extra imperfect copy brings the collective state of all copies correlated with, say, $|s_{j}\rangle$, closer to orthogonality with the collective state of all of the copies correlated with any other state $|s_{k}\rangle$. Therefore, one could distinguish $|s_{j}\rangle$ from $|s_{k}\rangle$ by a measurement on a collection of sufficiently many copies, and use that information to produce their “clones”. As a consequence, also imperfect copying (any value of $\langle \varepsilon_{j}|\varepsilon_{k}\rangle$ except 1) that preserves the “original” is prohibited.

We now have a useful definition of an *event*. Wheeler [47]—following Bohr—insisted that “No phenomenon is a phenomenon until it is a measured (recorded) phenomenon”. Our contribution is to supply—using information transfer and the dynamics of copying—an operational definition of a “recorded phenomenon”. We have just demonstrated that the ability to record events repeatedly associates them with a set of orthogonal states. This in turn implies discreteness, and the inevitability of jolts, quantum jumps that force the system to choose one of the items on the discrete menu of final (outcome) states.

Events that get recorded repeatedly precipitate quantum jumps. They emerge—as a consequence of the discreteness we have just deduced—from within the quantum measurement setting (as discussed, e.g., by von Neumann, 1932 [4]) where both the state of the measured system and of the apparatus are initially pure, and the final state (while entangled) is also pure. The defining characteristic of an event is a transition from before the measurement (from the old state of the system that was known, but it was not known what will happen when the new measurement is made) to when the outcome of the new measurement can be confirmed by repeated re-measurements.

Appearance of events in a pure state case prompts the question about their probabilities. If we were to proceed logically we would suspend discussion of how the core postulates (o)–(iii) imply the essence of axiom (iv), derive Born’s rule, and only then come back and consider how quantum jumps—the essence of the collapse—emerge in the mixed state case using the relation between pure states and reduced density matrices, the usual tools of decoherence. This course of argument would require a detour before we can come back and complete the discussion that we have already started.

We shall avoid this, but we shall also avoid using probabilities and Born’s rule, as in [35,48]. Some readers may nevertheless prefer to take that detour on their own, “jump” to Section 3, and return to the discussion below after they are convinced that Born’s rule emerges from the symmetries of entanglement in the pure state case. While the reasoning below does not depend on probabilities computed using $p_{k}={\left|\psi_{k}\right|}^{2}$, it employs ideas (such as purification) and mathematical tools (such as trace) that are suggested by decoherence and useful in the “Church of Larger Hilbert Space” approach to mixtures.

We also note that our tasks differ depending on whether mixed states of the control or mixed states of the target (see Figure 1) are the focus of attention. We start below with the simpler case—a target (e.g., an environment) that is in the mixed state. In that case generalization from pure states to mixtures is relatively straightforward, as the challenge is primarily technical [42,49].

Generalization of our discussion to the case when the control—the source of information—is allowed to be in a mixed state must take into account an additional complication: The state of control can change, and yet result in the same copy—a quantum meme or a *qmeme*—of the essential information. This degeneracy is important in considering readout of information from a macroscopic apparatus pointer or any other macroscopic device that is supposed to keep reliable records [43]. Obviously, the detailed microscopic state of such a device is of little consequence—the information of interest is what gets copied. It resides in the corresponding (likely macroscopic) degrees of freedom (e.g., of an apparatus pointer). Many microscopic states may (and usually will) represent the same information. Therefore, degeneracy—the fact that many microstates represent the same record and will result in the same copy of that record—must be considered along with the possibility of mixed states of the control. We shall return to this case of mixed and degenerate control later in this section.

### 2.2. Mixed States of the “Target”

Equations (2.1)–(2.4) are based on idealizations that include purity of the initial state of E. Regardless of whether E designates an environment or an apparatus, this is unlikely to be a good assumption. However, this assumption is also easily bypassed: An unknown state of E can be represented as a pure state of an enlarged system. This is the purification (aka “Church of Larger Hilbert Space”) strategy: Instead of a density matrix $\rho_{E}=\sum_{i}p_{i}|\varepsilon_{i}\rangle \langle \varepsilon_{i}|$ of a mixed state one can deal with a pure entangled state of E and $E^{\prime }$ defined in $H_{E}⊗H_{E^{\prime }}$:(2.5*a*)$$|\varepsilon \varepsilon^{\prime }\rangle =\sum_{i}{\sqrt{p}}_{i}|\varepsilon_{i}\rangle |\varepsilon_{i}^{\prime }\rangle ,$$
so that;
(2.5*b*)$$\rho_{E}=\sum_{i}p_{i}|\varepsilon_{i}\rangle \langle \varepsilon_{i}|={\mathrm{Tr}}_{E^{\prime }}|\varepsilon \varepsilon^{\prime }\rangle \;\langle \varepsilon \varepsilon^{\prime }|\;.$$
Therefore, when the initial state of E is mixed, there is always a pure state in an enlarged Hilbert space. Instead of (2.1) we can then write $|s_{k}\rangle |\varepsilon_{0}\varepsilon^{\prime }\rangle \Rightarrow |s_{k}\rangle |\varepsilon_{k}\varepsilon^{\prime }\rangle$ in obvious notation, and all of the steps that lead to Equations (2.3)–(2.4) can be repeated, so that:(2.6)$$\langle s_{j}|s_{k}\rangle \left(1-\langle \varepsilon_{j}\varepsilon^{\prime }|\varepsilon_{k}\varepsilon^{\prime }\rangle \right)\;=\;0\;\;$$
and forcing one to the same conclusions as Equation (2.4).

Purification relates pure states and density matrices by treating $\rho_{E}=\sum_{i}p_{i}|\varepsilon_{i}\rangle \langle \varepsilon_{i}|$ as a result of a trace. The connection of $\rho_{E}$ with $|\varepsilon \varepsilon^{\prime }\rangle =\sum_{i}{\sqrt{p}}_{i}|\varepsilon_{i}\rangle |\varepsilon_{i}^{\prime }\rangle$ does involve tracing. However, there is no need to regard weights $p_{i}$ as probabilities. They are just coefficients that relate a state of the whole $|\varepsilon \varepsilon^{\prime }\rangle$ and of its part $\rho_{E}$ by a mathematical operation—a trace. Thus, $\rho_{E}$ is a mathematical object that represents a reduction of a pure state that exists in the larger Hilbert space, but does not yet—in absence of Born’s rule—merit statistical interpretation.

Indeed, there is no need to even mention $\rho_{E}$. All of the above discussion can be carried out right from the start with a pure state in a larger Hilbert space. It suffices to assume only that *some* such pure state in the enlarged Hilbert space exists and that lack of purity of E is a result of its entanglement with the rest of the Universe. This does not rely on Born’s rule, but it does assert that ignorance that is reflected in a mixed local state (here, of E) can be regarded as a consequence of entanglement. This assertion is established in the next section, so—as we have already noted—readers can break the order of the presentation, consult the derivation of Born’s rule in the next section, and return here afterwards.

There is also an alternative way to proceed that leads to the same conclusions but does not require purification. Instead, we assume at the outset that we can represent states as density matrices. Unitary evolution preserves scalar products, i.e., Hilbert-Schmidt norm of density operators defined by $\mathrm{Tr}\rho \rho^{\prime }$. Therefore, one is led to:$$\mathrm{Tr}|s_{j}\rangle \;\langle s_{j}|\rho_{E}|s_{k}\rangle \;\langle s_{k}|\rho_{E}=\mathrm{Tr}|s_{j}\rangle \;\langle s_{j}|\rho_{E|j}|s_{k}\rangle \;\langle s_{k}|\rho_{E|k},$$
where $\rho_{E|j}$ and $\rho_{E|k}$ are mixed states of E affected by the two states of S that are unperturbed by copying. This in turn yields;
(2.7)$$|\langle s_{j}|s_{k}\rangle {|}^{2}\left(\mathrm{Tr}\rho_{E}^{2}-\mathrm{Tr}\rho_{E|j}\rho_{E|k}\right)=0,\;$$
which can be satisfied only in the same two cases as before: Either $\langle s_{j}|s_{k}\rangle =0$, or $\mathrm{Tr}\rho_{E}^{2}=\mathrm{Tr}\rho_{E|j}\rho_{E|k}$ which implies (by Schwarz inequality) that $\rho_{E|j}=\rho_{E|k}$ (i.e., there can be no record of nonorthogonal states of S).

This conclusion can be reached even more directly: Obviously, $\rho_{E|j}$ and $\rho_{E|k}$ have the same eigenvalues $p_{m}$ as $\rho_{E}=\sum_{m}p_{m}|\varepsilon_{m}\rangle \;\langle \varepsilon_{m}|$ from which they have unitarily evolved. Consequently, they could differ from each other only in their eigenstates that could contain record of the state of S, e.g.,: $\rho_{E|k}=\sum_{m}p_{m}|\varepsilon_{m|k}\rangle \;\langle \varepsilon_{m|k}|$. However, $\mathrm{Tr}\rho_{E|j}\rho_{E|k}=\sum_{m}{p_{m}}^{2}{\left|\langle \varepsilon_{m|j}|\varepsilon_{m|k}\rangle \right|}^{2}$, coincides with $\mathrm{Tr}{\rho_{E}}^{2}$ iff $|\langle \varepsilon_{m|j}|\varepsilon_{m|k}\rangle {|}^{2}=1$ whenever $p_{m}\neq 0$. It follows that $\rho_{E|j}=\rho_{E|k}$. Therefore, unless $\langle s_{j}|s_{k}\rangle =0$, states $|s_{j}\rangle$ and $|s_{k}\rangle$ cannot leave any imprint that distinguishes them—cannot deposit any record—in E.

In other words, in case of mixed target we can establish our key result using only pure states in an enlarged Hilbert space (purification), or only density matrices. The only reason one might want to invoke Born’s rule is to provide a physically (rather than only mathematically) motivated bridge between these two representations of “impure” states of E. Such a bridge is obviously useful, but it is not essential in arriving at the desired conclusions we reach in this section.

The economy of our assumptions stands in stark contrast with the uncompromising nature of our conclusions: Predictability—the demand that information transfer preserves the state of the system (embodied in postulate (iii))—was, along with the superposition principle (i) and unitarity of quantum evolutions (ii)—key to our derivation of the discreteness of states that can be repeatedly accessed. Discrete terminal states are behind the inevitability of quantum jumps.

We shall see in Section 4 that existence of stable terminal points allows for amplification and for the resulting preponderance of records about the states in which the system persist—in spite of the coupling to the environment—for long time periods. These sojourns of predictable evolution can be occasionally interrupted by a jump into another stable terminal state caused by perturbations that do not commute with the pointer observables monitored by the environment.

### 2.3. Predictability Killed the (Schrödinger’s) Cat

There are several ways to describe our conclusions so far. To restate the obvious, we have established that repeatedly accessible outcome states must be orthogonal. This is the interpretation—independent part of axiom (iv)—all of it except for the literal collapse. The core quantum postulates alone make it impossible to find out preexisting quantum states.

This is enough for the relative state account of quantum jumps—collapse axiom (iv) is not necessary for that. So, a cat suspended between life and death [20] cannot be seen in the records it leaves in the monitoring environment. Repeated records of only one of these two options will be available because only the two stable states (unperturbed by copying) allow for repeatability (postulate (iii))—for predictability (hence the above title).

Another way of stating our conclusion is to note that a set of orthogonal states defines a Hermitian operator when supplemented with real eigenvalues. The above discussion is then a derivation of the Hermitian nature of observables. It justifies the focus on Hermitian operators often invoked in textbook version of measurement axioms [3].

We note that “strict repeatability” (that is, assertion that states $\left\{|s_{k}\rangle \right\}$ cannot change at all in the course of a measurement) is not needed: They can evolve providing that their scalar products remain unaffected. That is,
(2.8)$$\sum_{j,k}\alpha_{j}^{*}\alpha_{k}\langle s_{j}|s_{k}\rangle =\sum_{j,k}\alpha_{j}^{*}\alpha_{k}\langle {\tilde{s}}_{j}|{\tilde{s}}_{k}\rangle \langle \varepsilon_{j}|\varepsilon_{k}\rangle$$
leads to the same conclusions as Equation (2.2) as long as $\langle s_{j}|s_{k}\rangle =\langle {\tilde{s}}_{j}|{\tilde{s}}_{k}\rangle$. Thus, when $|{\tilde{s}}_{j}\rangle$ and $|{\tilde{s}}_{k}\rangle$ are related with their progenitors by a transformation that preserves scalar product (as would, e.g., any reversible evolution) the proof of orthogonality goes through unimpeded. Both unitary and antiunitary transformations are in this class. Other similar generalizations are also possible [50,51].

We can also consider situations when this is not the case—$\langle s_{j}|s_{k}\rangle \neq \langle {\tilde{s}}_{j}|{\tilde{s}}_{k}\rangle$. An extreme example of this arises when the state of the measured system retains no memory of what it was before (e.g., $|s_{j}\rangle \Rightarrow |0\rangle ,\;|s_{k}\rangle \Rightarrow |0\rangle$). For example, photons are usually absorbed by detectors, and coherent states (that are not orthogonal) play the role of the outcomes. Then the apparatus can (and, indeed, by unitarity, has to) “inherit” the distinguishability—the information—previously residing in the system. In that case the need for orthogonality of $|s_{j}\rangle$ and $|s_{k}\rangle$ disappears. Of course such measurements do not fulfill postulate (iii)—they are not repeatable.

We emphasize that Born’s rule was not used above. The values of the scalar product that played a role in the proofs are $\langle s_{j}|s_{k}\rangle =0$ or $\langle s_{j}|s_{k}\rangle =1$, and the key distinction was between the zero and non-zero value of $\langle s_{j}|s_{k}\rangle$. Both “0” and “1” correspond to certainty. For instance, when we have asserted immediately below Equation (2.4) that $\langle \varepsilon_{j}|\varepsilon_{k}\rangle =1$, this implies that these two states of E are certainly identical. We have therefore derived probability for a very special case already. We shall relying on this special case—certainty—in the derivation of probabilities in Section 3.

### 2.4. Records and Branches: Degenerate “Control”

Our discussion so far is based on one key assumption—repeatability of measurement outcomes—which we have usually simplified to mean “nondemolition measurements”, i.e., repeatable accessibility of the same “original” state of the measured system. However, as we have already noted, for microscopic systems this is at best an exception. On the other hand, repeatability is essential for an apparatus A, at the level of measurement *records*. Pointer of an apparatus can be read out many times, and everyone should agree on where does it point—on what is the record. Indeed, this repeatable accessibility is a property of not just apparatus pointers, but a defining property of states that comprise “objective classical reality”. So, while the repeatability postulate (iii) at the level of quantum systems is an idealization of a theorist (e.g., Dirac, [3]), persistence of records stored in A as well as of effectively classical states of macroscopic quantum systems we encounter in our everyday experience is an essential fact of life and, therefore, a key desideratum of a successful theory of objective classical reality. Here, we extend our discussion of repeatability to account for it using our core quantum postulates.

We start by noting that one is almost never interested in the state of the apparatus as a whole: Finding out pure states of an object with Avogadro’s number of atoms (and, hence, with Hilbert space dimension of the order of $10^{10^{23}}$) is impractical and unnecessary. Obviously, there are many microstates of the apparatus that correspond to the same memory state and yield the same readout. We have to modify our above “nondemolition” approach to allow for perturbations of the microscopic states and to account for this degeneracy. Once we have done this, we shall also find it easier to deal with mixed states of A that, for any macroscopic system, are certainly typical.

Consider two pure states $|v_{1}\rangle$ and $|v_{2}\rangle$ that represent the same record “*v*”. We take this to mean that observers or other memory devices $M,\;M^{\prime },...$ will register the same state after interacting with A in either $|v_{1}\rangle$ or $|v_{2}\rangle$:(2.9*v*)$$|v_{1}\rangle |\mu \rangle |\mu^{\prime }\rangle ...\Rightarrow |{\tilde{v}}_{1}\rangle |\mu_{v}\rangle |\mu^{\prime }\rangle ...\Rightarrow |{\tilde{\tilde{v}}}_{1}\rangle |\mu_{v}\rangle |\mu_{v}^{\prime }\rangle ,$$
$$|v_{2}\rangle |\mu \rangle |\mu^{\prime }\rangle ...\Rightarrow |{\tilde{v}}_{2}\rangle |\mu_{v}\rangle |\mu^{\prime }\rangle ...\Rightarrow |{\tilde{\tilde{v}}}_{2}\rangle |\mu_{v}\rangle |\mu_{v}^{\prime }\rangle .$$
Note that evolution of the “original” is allowed (e.g., $|v_{1}\rangle \Rightarrow |{\tilde{v}}_{1}\rangle \Rightarrow |{\tilde{\tilde{v}}}_{1}\rangle$). as long as it does not affect the repeatability of what is read out by $M,\;M^{\prime },...$.

It is straightforward to see that any superposition or any mixed state of $|v_{1}\rangle$ and $|v_{2}\rangle$ will also register the same way—as $|\mu_{v}\rangle$—in the memory M. Registration of a different outcome—different readout *w*—by memory M can be represented as:(2.9*w*)$$|w_{1}\rangle |\mu \rangle |\mu^{\prime }\rangle ...\Rightarrow |{\tilde{w}}_{1}\rangle |\mu_{w}\rangle |\mu^{\prime }\rangle ...\Rightarrow |{\tilde{\tilde{w}}}_{1}\rangle |\mu_{w}\rangle |\mu_{w}^{\prime }\rangle ,$$
$$|w_{2}\rangle |\mu \rangle |\mu^{\prime }\rangle ...\Rightarrow |{\tilde{w}}_{2}\rangle |\mu_{w}\rangle |\mu^{\prime }\rangle ...\Rightarrow |{\tilde{\tilde{w}}}_{2}\rangle |\mu_{w}\rangle |\mu_{w}^{\prime }\rangle .$$
Again, there are many microstates—$|w_{1}\rangle ,|w_{2}\rangle$, etc.—that yield the same readout $|\mu_{w}\rangle$.

The above account offers a model of what happens when an apparatus A is consulted by many observers that can be represented by distinct M’s. They can perturb the microstate but leave the record intact.

We can now repeat the pure state reasoning from above, assuming that the “control”—which was before the measured system, and may now be the apparatus A—is in a pure state. We are led to an analogue of Equation (2.4) that can be satisfied in two different ways: Either the memory devices register the same readout regardless of the underlying microscopic state of the system, as in:(2.10)$$\forall_{k,l}\;\;\langle v_{k}|v_{l}\rangle =\langle {\tilde{v}}_{k}|{\tilde{v}}_{l}\rangle \langle \mu_{v}|\mu_{v}\rangle ,$$
(so that $\langle \mu_{v}|\mu_{v}\rangle =1$, in which case scalar products between the underlying states of A can take any value), or the readouts can differ,
(2.11)$$\forall_{k,l}\;\;\langle v_{k}|w_{l}\rangle =\langle {\tilde{v}}_{k}|{\tilde{w}}_{l}\rangle \langle \mu_{v}|\mu_{w}\rangle ,$$
and $\langle v_{k}|w_{l}\rangle$ have to be orthogonal when they lead to distinct records in $M,\;M^{\prime },...$.

The relation between the states of the control defined by the readout—by the imprints they leave on the state on the “target” M—is reflexive, symmetric and transitive. Hence, it defines equivalence classes: States that leave imprint “*v*” form a class V distinct from states in W that imprint “*w*”. Record states in V (W, etc.) should retain class membership under the evolutions generated by readouts (otherwise they cannot be repeatedly consulted and keep the record). It is natural to represent such equivalence classes of states with orthogonal subspaces in the Hilbert space $H_{A}$. It is also possible to define probabilities as measures on such equivalence classes and regard them as (macroscopic or coarse-grained) “events” (see e.g., Gnedenko, 1968, [52]).

Generalization to when the apparatus is in a mixed state can be carried out using purification strategy as before (this time purifying the “control” A) or by using preservation of the Hilbert-Schmidt product. Thus, unitary evolutions;
(2.12*v*)$$\rho_{V}^{A}\left|\mu \rangle \;\langle \mu \right|⟹{\tilde{\rho }}_{V}^{A}|\mu_{v}\rangle \;\langle \mu_{v}|,$$
(2.12*w*)$$\rho_{W}^{A}\left|\mu \rangle \;\langle \mu \right|⟹{\tilde{\rho }}_{W}^{A}|\mu_{w}\rangle \;\langle \mu_{w}|,$$
where $\rho_{V}^{A}$ ($\rho_{W}^{A}$, etc.) is any density matrix with support restricted to only V (W, etc.) imply equality:(2.13)$$\mathrm{Tr}\rho_{V}^{A}\rho_{W}^{A}{\left|\langle \mu |\mu \rangle \right|}^{2}=\mathrm{Tr}{\tilde{\rho }}_{V}^{A}{\tilde{\rho }}_{W}^{A}{\left|\langle \mu_{v}|\mu_{w}\rangle \right|}^{2}.$$
This is an analogue of the derivation of Equations (2.4)–(2.7) when the mixed state of the control is represented by a density matrix. As before, we conclude that $\mathrm{Tr}\rho_{V}^{A}\rho_{W}^{A}=0$ unless $|\langle \mu_{v}|\mu_{w}\rangle {|}^{2}=1$.

In contrast to Equation (2.6) (where “control” S was pure, but the state of the “target” E was mixed) now the target is the memory M, and its state starts and remains pure. This shifting of “mixedness” from the target (as in Equation (2.7)) to the control may seem somewhat arbitrary, but—in the present setting—it is well justified. The motivation before was the process of decoherence or measurement, and the focus of attention was the system S. Now, the motivation is the readout of the state of the apparatus pointer by observers (but information flow in decoherence and in quantum Darwinism we discuss in Section 4 can be treated in the same manner).

#### Repeatability and Actionable Information

Previously we have modeled the acquisition of information about a system by a (possibly macroscopic) apparatus or by the environment in the course of decoherence. In either case “target” could be expected to be in a mixed state but the “control” was pure. Now we are dealing with an apparatus acting as a macroscopic control. Its microscopic state is in general mixed, and can be influenced by the readout, but we still expect it to retain the record (e.g., of a measurement outcome). This is possible because of degeneracy—many microscopic states represent the same record.[note 3]

This record should be repeatedly accessible and unambiguous. Before, in the discussion following Equations (2.1)–(2.4), repeatability was assured by insisting that the state of the system—of the control—should remain unchanged during the readout. Now, we can no longer count on the preservation of the *state* of the original to establish repeatability. Instead, we demand—as a criterion for repeatable accessibility—that; (i) the copies should contain the same information, and; (ii) that information should suffice to distinguish record V from W.

Above, we have seen how this demand can be implemented when the states of the memories $M,M^{\prime }$, etc. are pure. Relaxing the assumption of pure memory states is possible. One can also allow for decoherence caused by the environment E. Thus, consider sequence of copying operations that, along with decoherence, lead to:(2.14*v*)$$\rho_{V}^{A}\rho_{0}^{M}\rho_{0}^{M^{\prime }}...\rho_{0}^{E}⟹\rho_{V}^{AMM^{\prime }...E},$$
(2.14*w*)$$\rho_{W}^{A}\rho_{0}^{M}\rho_{0}^{M^{\prime }}...\rho_{0}^{E}⟹\rho_{W}^{AMM^{\prime }...E}.$$
Note that we allow the apparatus A that contains the original record, various memories, as well as E to remain correlated. Such a general final state suggests an obvious question: How can we test whether, say, memory M has indeed acquired a copy of the record in A that offers (at least partial) distinguishability of V from W?

To address this question we propose an operational criterion: The information contained in each of the memories should be *actionable*—it should allow one to alter the state of a test system T. Thus, copy in M will be certified as “actionable” when there is a conditional unitary transformation U(T|M) that alters the state of the test system so that:(2.15*v*)$$\rho_{V}^{AMM^{\prime }...E}⊗\rho_{0}^{T}\;\;\overset{U(T|M)}{⟹}\;\;{\tilde{\rho }}_{V}^{AMM^{\prime }...E}⊗\rho_{V}^{T},$$
(2.15*w*)$$\rho_{W}^{AMM^{\prime }...E}⊗\rho_{0}^{T}\;\;\overset{U(T|M)}{⟹}\;\;{\tilde{\rho }}_{W}^{AMM^{\prime }...E}⊗\rho_{W}^{T}.$$
The test of actionability will be successful—the information in M will be declared actionable—when there exists an initial state $\rho_{0}^{T}$ of the test system such that:(2.16*a*)$$\mathrm{Tr}{\left(\rho_{0}^{T}\right)}^{2}>\mathrm{Tr}\rho_{V}^{T}\rho_{W}^{T}.$$
Preservation of Schmidt–Hilbert norm under unitary evolutions implies that—unless $\rho_{V}^{AMM^{\prime }...E}$ and $\rho_{W}^{AMM^{\prime }...E}$ are orthogonal to begin with—the overlap between the two “branches” should increase to compensate for the decrease of overlap in the test system states, Equation (2.16a), so that;
(2.16*b*)$$\mathrm{Tr}\rho_{V}^{AMM^{\prime }...E}\rho_{W}^{AMM^{\prime }...E}<\mathrm{Tr}{\tilde{\rho }}_{V}^{AMM^{\prime }...E}{\tilde{\rho }}_{W}^{AMM^{\prime }...E}.$$
Moreover, as we have assumed that in M, $M^{\prime }...M^{\left(k\right)}...$ there are many copies of the original record in A, this test of actionability can be repeated. However, the overlap of the density matrices of $AMM^{\prime }...E$ on the RHS above cannot increase indefinitely as a result of such multiple iterations of actionability tests, as it is bounded from above by unity. Consequently, we conclude via this *reductio ad absurdum* reasoning that the ability to make multiple copies implies $\mathrm{Tr}\rho_{V}^{AMM^{\prime }...E}\rho_{W}^{AMM^{\prime }...E}=0$. Therefore,
(2.17)$$\mathrm{Tr}\rho_{V}^{A}\rho_{W}^{A}=0$$
is needed to allow for repeatable copying of the original record (or, more generally, the features distinguishing V from W of the original macroscopic state) in A.

We have assumed above that there is no preexisting correlation between the test system and the rest, $AMM^{\prime }...M^{\left(k\right)}...E$. We could have actually assumed a pure state of T: When there is a U(T|M) that alters a mixed state of T, there will certainly be a pure state of T that can be altered.

We also note that (as before) the whole argument can be recast in the language of the “Church of Larger Hilbert Space” [43]. That is, one can carry it out without any appeals to Born’s rule. There is an interesting subtlety in such treatments: The actionable information we have tested for above is local—it resides in a specific $M^{\left(k\right)}$. This need not be always the case: Actionable information may be nonlocal—it may reside in correlations between systems. Such an example is discussed in [43]. The locus of actionable information is assured by the selection of the conditional evolution operator $U(T|M^{\left(k\right)})$ that—above—couples only specific $M^{\left(k\right)}$ to T.

We conclude that only orthogonal projectors (above, of A) can act as “originals” for unlimited numbers of copies. Of course, many of the outcome states of quantum system S inferred from the measurement records in A are not orthogonal. Measurements that result in such outcomes are not repeatable: State of S is perturbed, but its record in A is repeatedly accessible, so there is no contradiction. Repeatability of the records is therefore possible even when the recorded states of S at the roots of the corresponding branches are not orthogonal. Positive operator valued measures (POVM’s, that is generalized measurements with outcomes that do not represent orthogonal states of the measured system, see [5]) arise naturally in this setting [43].

The reasoning behind the conclusions of this subsection parallels the pure states case, Equations (2.1)–(2.4), but the mathematics and, above all, the physical motivation, differ. Before we were dealing with the abstract postulate of repeatability that is found in Dirac [3] and other textbooks, but this idealized version is almost never implemented in the laboratory practice in measurements of microscopic quantum systems. In spite of its idealizations, the abstract Dirac version of repeatability of measurements should not be dismissed too easily: Being able to confirm that a state is what it is known to be is essential to justify the very idea of a state in general, and of a quantum state in particular. The role of the state is, after all, to enable predictions, and the simplest prediction (captured by repeatability, no matter how difficult nondemolition measurements are to implement in practice) is that existence of a state can be confirmed. Indeed, this is how quantum states can give rise to “existence” we have become accustomed to in our quantum Universe.

In a classical Universe repeatability is taken for granted, as an unknown classical state can be found out without endangering its existence. Repeatability in a quantum setting allows one to use fragile quantum states as building blocks of classical reality, as we shall see in more detail in Section 4.

In practice predictability and even repeatability are encountered not in the measured microscopic quantum system S but, rather, in the memory of the measuring apparatus A, and, indeed, in the states of macroscopic systems. Apparatus pointer can be, after all, repeatedly consulted, as can be any effectively classical state. Moreover, our perception of the collapse arises not from the direct evidence of the behavior of some microscopic quantum S, but, rather, from the records of its state inscribed in the memory of a macroscopic (albeit still quantum) A.

We have seen that the same condition of repeatability that led to orthogonality (and, hence, discreteness) in the set of possible outcomes in the pure case of S enforces orthogonality of the subspaces of A (even when the microscopic state of A is allowed to change). Thus, while Equations (2.1)–(2.4) account for quantum jumps in the idealized case of quantum postulates (Dirac, 1958) [3], this subsection shows that discrete quantum jumps can occur as a result of orthogonality of the whole subspaces of the Hilbert space $H_{A}$ corresponding to repeatedly accessible records—to macroscopic pointer subspaces of the measuring device.

### 2.5. Pointer Basis, Information Transfer, and Decoherence

We are now equipped with a set of measurement outcomes or, to put it in a way that ties in with the study of probabilities we shall embark on in Section 3, with a set of possible *events*. Our derivation above did not appeal to decoherence, but decoherence yields einselection (which is, after all, due to the information transfer to the environment). We will now see that einselection based on repeatability and einselection based on decoherence are in effect two views of the same phenomenon.

Popular accounts of decoherence and its role in the emergence of ‘the classical’ often start from the observation that when a quantum system S interacts with some environment E “phase relations in S are lost”. This is, at best, incomplete if not misleading, as it begs the more fundamental question: “Phases between *what*?”. This in turn leads directly to the main issue addressed by einselection: “*What is the preferred basis*?”. This key question is often muddled in the “folklore” accounts of decoherence.

The crux of the matter—the reason why interaction with the environment can impose classicality—is precisely the emergence of the preferred states. The basic criterion that selects preferred pointer states was discovered when the analogy between the role of the environment in decoherence and the role of the apparatus in a nondemolition measurement was recognized: What matters is that there are interactions that transfer information and yet leave some states of the system unaffected [24].

The criterion for selecting such preferred states is persistence of correlation between two systems (e.g., system S and apparatus A). For the preferred pointer states this correlation should persist in spite of immersion of A in the environment. It is obvious that states (of, e.g., A) that are best at retaining correlations (with, e.g., S) also retain identity—i.e., correlation with me, the observer—and resist entanglement with the environment.

Our discussion above confirms that the simple idea of preserving a state while transferring the information about it—also the central idea of einselection—is powerful, and can be analyzed using minimal purely quantum ingredients—core postulates (o)–(iii). It leads to breaking of the unitary symmetry and singles out preferred states of the apparatus pointer (supplied in textbooks by axiom (iv)) without any need to invoke physical (statistical) view of the reduced density matrices (which is central to the decoherence approach to collapse).

This is important, as partial trace (understood as an averaging procedure) and reduced density matrices (understood as probability distributions) employed in decoherence theory rely on Born’s rule (which endows them with physical significance). Our goal in the next section will be to arrive at Born’s rule, axiom (v)—to relate state vectors and probabilities. Obtaining preferred basis and deducing events without invoking density matrices and trace—without relying on Born’s rule—is essential if we are to avoid circularity in its derivation.

To compare derivation of the preferred states in decoherence with their emergence from symmetry breaking imposed by axioms (o)–(iii) we return to Equation (2.2). We also temporarily suspend prohibition on the use of partial trace to compute reduced density matrix of the system:(2.18)$$\rho_{S}=\sum_{j,k}\alpha_{j}\alpha_{k}^{*}\langle \varepsilon_{k}|\varepsilon_{j}\rangle |s_{j}\rangle \langle s_{k}|=Tr_{E}|\Psi_{ES}\rangle \;\langle \Psi_{ES}|.$$
Above we have expressed $\rho_{S}$ in the pointer basis defined by its resilience in spite of the monitoring by E and not in the Schmidt basis. Therefore, until decoherence in that basis is complete, and the environment acquires perfect records of pointer states;
(2.19)$$\langle \varepsilon_{j}|\varepsilon_{k}\rangle =\delta_{jk},$$
the eigenstates of $\rho_{S}$ do not coincide with the pointer states selected for their resilience in spite of the immersion in E.

Resilience—quantified by the ability to retain correlations in spite of the environment, and, hence, by persistence, as in Equations (2.4) and (2.7)—is the essence of the original definition of pointer states and einselection [24,25]. Such pointer states will be in general different from the instantaneous Schmidt states of S—the eigenstates of $\rho_{S}$. They will coincide with the Schmidt states of
(2.20)$$|\Psi_{ES}\rangle =\sum_{k}a_{k}|s_{k}\rangle |\varepsilon_{k}\rangle$$
only when $\left\{|\varepsilon_{k}\rangle \right\}$—their records in E—become orthogonal. We did not need orthogonality of $\left\{|\varepsilon_{k}\rangle \right\}$ to prove orthogonality of pointer states earlier in this section. It will be, however, useful in the next section, as it assures additivity of probabilities of the pointer states.

For pure states this discussion of additivity can be carried out in a setting that is explicitly free of any reference to density matrices or trace, and relies only on correlations (Zurek, 2005) [48]. Born’s rule would be needed to establish the connection between them and to endow reduced density matrix with physical (statistical) interpretation, but—as we have seen—orthogonality of outcomes central for the definition of events can be established without Born’s rule.

So, a piece of decoherence “folklore”—responsible for statements such as “decoherence causes reduced density matrix to be diagonal”—is at best imprecise, and often incorrect. The error is mathematical and obvious: $\rho_{S}$ is Hermitian, so it is always diagonalized by the Schmidt states of S. In addition, what we want in pointer states is preservation of their identity.

Still, “folklore” often assigns classicality to the eigenstates of $\rho_{S}$, and that would make them candidate to the status of events. This was even occasionally endorsed by some of the practitioners of decoherence (Zeh, 1990; [53]; 2007 [54]; Albrecht, 1992 [55], but see Albrecht et al., 2021 [56,57]) and taken for granted by others (see, e.g., [58]). However, by and large it is no longer regarded as viable [31,32,36]: The eigenstates of $\rho_{S}$ are not stable. They depend on the time and on the initial state of S, which disqualifies them as events in the sense needed to develop probability theory, and do not fit the bill as “elements of classical reality”. There are also situations where the eigenstates of $\rho_{S}$ can be (in contrast to pointer states that are local whenever the coupling with the environment is local) very nonlocal [59,60,61].

Nonlocality of pointer states need not necessarily be a problem. The role of decoherence is to predict what happens—what states are “pointer” given the physical context (including Hamiltonians, the nature of the environment, etc.). Thus, testing it in situations when its predictions clash with our classical intuition is of interest (see, e.g., Poyatos, Cirac, and Zoller, 1996 [62]). The problem with the eigenstates of $\rho_{S}$ is—primarily—their dependence on the initial state of the system. This is eliminated by the predictability sieve ([6,28,63]) and the repeatability-based approach (see also Refs. [42,43,49]).

As is often the case with folk wisdom, a grain of truth is nevertheless reflected in such oversimplified “proverbs”: When the environment acquires perfect knowledge of the states it monitors without perturbing them, and $\langle \varepsilon_{j}|\varepsilon_{k}\rangle =\delta_{jk}$, pointer states “become Schmidt”, and end up on the diagonal of $\rho_{S}$. Effective decoherence favors such alignment of Schmidt states with the pointer states. Given that decoherence is—at least in the macroscopic domain—very fast, this can happen essentially instantaneously.

Still, this coincidence should not be used to attempt a redefinition of pointer states as instantaneous eigenstates of $\rho_{S}$—instantaneous Schmidt states. As we have already seen, and as will become even clearer in the rest of this paper, it is essential to distinguish the process that fixes preferred pointer states (that is, dynamics of the information transfer that results in the measurement as well as decoherence, but does not depend on the initial state of the system) from the reasoning that assigns probabilities to these outcomes. These probabilities are determined by the initial state.

### 2.6. Irreversibility of Perceived Events, or “Don’t Blame the 2^nd^ Law—Wavepacket Collapse Is Your Own Fault!”

Irreversibility has been blamed for the collapse of the wavepacket since at least von Neumann (1932) [4]. The causes of irreversibility invoked in this context have typically classical analogues that go back to Boltzmann and the loss of information implicated in the Second Law (Zeh, 2007) [54].

Discrete quantum jumps occur as a consequence of the collapse. They are uniquely quantum, and a central conundrum of quantum physics. They reset the evolution relevant for the future of the observer putting it onto a course consistent with the measurement outcome (and prima facie at odds with the unitarity of quantum evolutions).

We have just seen how the discreteness of quantum jumps follows from the quantum core postulates. We now point out that—in addition to the “usual suspects” traditionally blamed for irreversibility—there is a uniquely quantum reason why events associated with quantum jumps are fundamentally irreversible. It is distinct from the information loss associated with the dynamics that is responsible for the Second Law.

This uniquely quantum source of irreversibility is a result of the information *gain* (rather than its loss). It is noteworthy that quantum physics provides a uniquely quantum key that solves the distinctly quantum conundrum of the wavepacket collapse.

We shall see below that information about the measurement outcome does not preclude reversal of the classical measurement, but makes it impossible to undo evolution that leads to a quantum measurement whenever a superposition of the potential outcomes—hence, the wavepacket collapse—is involved.

#### 2.6.1. Classical Measurement Can Be Reversed Even when Record of the Outcome Is Kept

Let us first examine a measurement carried out by a classical agent/apparatus A on a classical system S. The state s of S (e.g., location of S in phase space) is measured by a classical A that starts in the “ready to measure” state $A_{0}$:(2.21*a*)$$sA_{0}\;\overset{E_{\mathrm{SA}}}{⟹}\;sA_{s}.$$
The question we address is whether the combined state of SA can be restored to the pre-measurement $sA_{0}$ even after the information about the outcome is retained somewhere—e.g., in the memory device D.

The dynamics $E_{\mathrm{SA}}$ responsible for the measurement is assumed to be reversible and, in Equation (2.21*a*), it is classical. Therefore, classical measurement can be undone simply by implementing $E_{\mathrm{SA}}^{-1}$. An example of $E_{\mathrm{SA}}^{-1}$ is (Loschmidt inspired) instantaneous reversal of velocities.

Our main point is that the reversal;
(2.21*a′*)$$sA_{s}\;\overset{E_{\mathrm{SA}}^{-1}}{⟹}sA_{0}.$$
can be accomplished even after the measurement outcome is copied onto the memory device D;
(2.22*a*)$$sA_{s}D_{0}\;\overset{E_{\mathrm{AD}}}{⟹}\;sA_{s}D_{s},$$
so that the pre-measurement state of S is recorded elsewhere (here, in D). Above, $E_{\mathrm{AD}}$ plays the same role as $E_{\mathrm{SA}}$ in Equation (2.21*a*). That is, the states of S and A separately, or the combined state SA will not reveal any evidence of irreversibility. After the reversal;
(2.23*a*)$$sA_{s}D_{s}\;\overset{E_{\mathrm{SA}}^{-1}}{⟹}\;sA_{0}D_{s},$$
the state of SA is identical to the pre-measurement state, even though the recording device still has the copy of the outcome. Starting with a partly known state of the system does not change this conclusion [23].

#### 2.6.2. Quantum Measurement Can’t Be Reversed when the Record of the Outcome Is Kept

Consider now a measurement of a quantum system S by a quantum A:(2.21*b′*)$$\left(\sum_{s}\alpha_{s}|s\rangle \right)|A_{0}\rangle \overset{U_{SA}}{⟹}\;\sum_{s}\alpha_{s}|s\rangle |A_{s}\rangle .$$
The evolution operator $U_{SA}$ is unitary (for example, $U_{SA}=\sum_{s,k}\left|s\rangle \;\langle s|\right|A_{k+s}\rangle \;\langle A_{k}|$ with orthogonal $\left\{|s\rangle \right\}$, $\left\{|A_{k}\rangle \right\}$ would do the job). Therefore, evolution that leads to a measurement is in principle reversible. Reversal implemented by $U_{SA}^{†}$ will restore the pre-measurement state of SA:(2.21*b′*)$$\sum_{s}\alpha_{s}|s\rangle |A_{s}\rangle \overset{U_{SA}^{†}}{⟹}\;\left(\sum_{s}\alpha_{s}|s\rangle \right)|A_{0}\rangle .$$
Let us, however, assume that the measurement outcome is copied before the reversal is attempted:(2.22*b*)$$\left(\sum_{s}\alpha_{s}|s\rangle |A_{s}\rangle \right)|D_{0}\rangle \overset{U_{AD}}{⟹}\sum_{s}\alpha_{s}|s\rangle |A_{s}\rangle |D_{s}\rangle .$$
Here $U_{AD}$ plays the same role and can have the same structure as $U_{SA}$.

Note that unitary evolutions above implement *repeatable* measurement/copying on the states $\left\{|s\rangle \right\}$, $\left\{|A_{s}\rangle \right\}$ of the system and of the apparatus, respectively. That is, these states of S and A remain untouched by the measurement and copying processes. As we have seen, such repeatability implies that the outcome states $\left\{|s\rangle \right\}$ as well as the record states $\left\{|A_{s}\rangle \right\}$ are orthogonal.

When the information about the outcome is copied, the pre-measurement state $\left(\sum_{s}\alpha_{s}|s\rangle \right)|A_{0}\rangle$ of SA pair cannot be restored by $U_{SA}^{†}$. That is:(2.23*b*)$$U_{SA}^{†}\left(\sum_{s}\alpha_{s}|s\rangle |A_{s}\rangle |D_{s}\rangle \right)\;=\;|A_{0}\rangle \left(\sum_{s}\alpha_{s}|s\rangle |D_{s}\rangle \right).$$
The apparatus is restored to the pre-measurement $|A_{0}\rangle$, but the system remains entangled with the memory device. On its own, its state is represented by the mixture:$$\varrho^{S}=\sum_{s}w_{ss}|s\rangle \;\langle s|,$$
where $w_{ss}={\left|\alpha_{s}\right|}^{2}$. Reversing quantum measurement of a state that corresponds to a superposition of the potential outcomes is possible only providing the memory of the outcome is no longer preserved anywhere else in the Universe. Moreover, that means that the information transfer has to be “undone”—scrambling the record makes it inaccessible, but does not get rid of the evidence of the outcome.

We have now demonstrated the difference between the ability to reverse quantum and classical measurement. Information flows do not matter for classical, Newtonian dynamics. However, when information about a quantum measurement outcome is communicated—copied and retained by any other system—the evolution that led to that measurement cannot be reversed.

Quantum irreversibility can result from the information gain rather than just its loss—rather than just an increase of the (von Neumann) entropy. Recording of the outcome of the measurement resets, in effect, initial conditions within the observer’s (branch of) the Universe, resulting in an irreversible, uniquely quantum “wavepacket collapse”. Thus, from the point of view of the measurer, information retention about an outcome of a quantum measurement implies irreversibility. Quantum states are epiontic.

### 2.7. Summary: Events, Irreversibility, and Perceptions

What the observer knows is inseparable from what the observer is: the physical state of the agent’s memory represents the information about the Universe. The reliability of this information depends on the stability of its correlation with external observables. In this very immediate sense, decoherence brings about the apparent collapse of the wavepacket: after a decoherence time scale, only the einselected memory states will exist and retain useful correlations [27,36,64,65]. The observer described by some specific einselected state (including a configuration of memory bits) will be able to access (”recall”) only that state.

Collapse is a consequence of einselection and of the one-to-one correspondence between the physical state of the observer’s memory and of the information encoded in it. Memory is simultaneously a description of the recorded information and part of an identity tag, defining the observer as a physical system. It is as inconsistent to imagine an observer perceiving something other than what is implied by the stable (einselected) records as it is impossible to imagine the same person with a different DNA. Both cases involve information encoded in a state of a system inextricably linked with the physical identity of an individual.

Distinct memory/identity states of the observer (which are also his states of knowledge) cannot be superposed. This censorship is strictly enforced by decoherence and the resulting einselection. Distinct memory states label and inhabit different branches of Everett’s many-worlds universe. The persistence of correlations between the records (data in possession of the observers) and the recorded states of macroscopic systems is all that is needed to recover objective classical reality. In this manner, the distinction between ontology and epistemology—between what is and what is known to be—is dissolved. There can be *no information without representation* [66].Quantum states are *epiontic*.

The discreteness underlying “collapse of the wavepacket” has a well-defined origin—it resolves the conflict between the linearity of the unitary quantum evolutions and the nonlinearity associated with the amplification of information in measurements but also in the monitoring by the environment—in decoherence. Any process that involves (even modest) amplification—that leads to copies, qmemes of an “original state” (which, in view of the demand of repeatability, should survive the copying)—demands orthogonality.

Copying (as any other quantum evolution) is unitary, so it will not result in collapse. However, perception of the collapse will arise as a consequence of the irreversibility induced by the information transfer we have just discussed. This purely quantum irreversibility provides a mechanism for collapse of the wavepacket that was not available (and not needed) in the classical setting—a mechanism that is fundamental, and uniquely quantum. The old question about the origins of irreversibility acquires a new quantum aspect especially apparent in the context of quantum measurements: Thus, while in the classical setting measuring of an evolving state of the system need not alter its evolution, in the quantum setting measurement derails evolution and redirects it onto the track consistent with the record made by the observer. One might say that a measurement re-sets the initial condition of the evolving quantum system [23].

Thus, while the irreversible wavepacket collapse was sometimes blamed on the consciousness of the observer (von Neumann, 1932 [4]; London and Bauer, 1939 [67]; Wigner, 1961 [68]), we have identified a purely physical cause of the collapse: Observer retaining information about the outcome precludes the reversal.

In the next section, we derive Born’s rule. We build on einselection, but in a way ([7,35,48]) that does not rely on axioms (iv) or (v). In particular, the use of reduced density matrices we allowed temporarily shall be prohibited. We shall be able to use them again only after Born’s rule has been derived.

From the point of view of axiom (v) and the rest of this paper the most important conclusion of the present section is that *repeatability requires distinguishability*. In a quantum setting of Hilbert spaces and unitarity of evolutions—postulates (i) and (ii)—this means that *repeatability begets orthogonality*. to assure repeatability—ability to reconfirm what is known—one must focus on mutually exclusive events represented by orthogonal states.

We end this section with a simple purely quantum definition of events in hand: Record made in the measurement resets initial conditions relevant for the subsequent evolution of the branch of the universal state vector tagged by that record. We now take up the question: What is the probability of a particular record—specific new initial condition—given the preexisting superposition of the possible outcome states.

## 3. Born’s Rule from the Symmetries of Entanglement

The first widely accepted definition of probability was codified by Laplace (1820) [69]: When there are *N* possible distinct outcomes and the observer is ignorant of what will happen, all alternatives appear equally likely. Probability observer should then assign to any one outcome is 1/N. Laplace justified this *principle of equal likelihood* using *invariance* encapsulated in his ‘principle of indifference’: Player ignorant of the face value of cards (Figure 2a) will be *indifferent* when they are swapped before he gets the card on top, even when one and only one of the cards is favorable (e.g., a spade he needs to win).

Laplace’s invariance under swaps captures *subjective* symmetry: Equal likelihood is a statement about observers ‘state of mind’ (or, at best, his records), and *not* a measurable property of the real physical state of the system (which is, after all, altered by swaps; see Figure 2b). In the classical setting probabilities defined in this manner are therefore ultimately unphysical. Moreover, indifference, likelihood, and probability are all ill-defined attempts to quantify ignorance. Expressing one undefined concept in terms of another is not a definition.

It is therefore no surprise that equal likelihood is no longer regarded as a sufficient foundation for classical probability, and several other attempts are vying for primacy [70]. Among them, relative frequency approach has perhaps the largest following, although it needs infinite (hence, fictitious) ensembles (von Mises, 1939) [71], and, thus, it is doubtful if it addresses the issue of “subjectivity”.[note 4] Nevertheless, popularity of relative frequency approach made it an obvious starting point in the attempts to derive Born’s rule, especially in the relative states context where “branches” of the universal state vector can be counted. However, attempts to date (Everett [10,11]; DeWitt [14,15]; Graham [72]; Geroch [73]) have been found lacking. This is because counting of many world branches does not suffice. “Maverick” branches that have frequencies of events very different from those predicted by Born’s rule are also a part of the universal state vector. Relative frequencies alone do not imply that an observer is unlikely to be found on such a branch. To get rid of them one would have to assign to them—without physical justification—vanishing probabilities related to their small amplitudes. This amplitude-probability connection goes beyond the relative frequency approach, in effect requiring—in addition to frequencies—another measure of probability. Papers mentioned above introduce it *ad hoc*. This is consistent with Born’s rule, but deriving it on this basis is circular (Stein [74]; Kent [38]; Joos [75]; Weinberg [76]). Indeed, formal attempts based on the “frequency operator” lead to mathematical inconsistencies (Squires [39]).

The problem can be “made to disappear”—coefficients of maverick branches become vanishingly small (along with coefficients of *all* branches)—in the limit of *infinite* and fictitious (and, hence, subjectively assigned) ensembles (Hartle, 1968 [79]; Farhi, Goldstone, and Guttman, 1989 [80]). Such infinite ensembles—one might argue—are always required by the frequentist approach in the classical case, but this is a poor excuse (see Kent, 1990) [38]. As noted above, infinite ensembles are unphysical, subjective, and a weak spot of relative frequencies approach also in a classical setting. Moreover, in quantum mechanics infinite ensembles may pose problems that have to do with the structure of infinite Hilbert spaces (Poulin, 2005 [81]; Caves and Shack, 2005 [82]). It is debatable whether these mathematical problems are fatal, but it is also difficult to disagree with Kent (1990) [38] and Squires (1990) [39] that the need to go to a limit of infinite ensembles to define probability in a finite Universe disqualifies use of relative frequencies in the relative states setting.

The other way of dealing with this issue is to modify physics so that branches with small enough amplitude simply do not count (Geroch, 1984 [73]; Buniy, Hsu, and Zee, 2006 [83]). At least until experimental evidence for the required modifications of quantum theory is found one can regard such attempts primarily as illustration of the seriousness of the problem of the origin of Born’s rule.

Kolmogorov’s approach—probability as a measure (see, e.g., Gnedenko, 1968 [52])—bypasses the question we aim to address: How to relate probabilities to (quantum) states. It only shows that any sensible assignment (non-negative numbers that for a mutually exclusive and exhaustive set of events sum up to 1) will do. Moreover, Kolmogorov assumes additivity of probabilities while quantum theory insists—via superposition principle—on the additivity of state vectors. These two additivity requirements are at odds, as double slit experiment famously demonstrates.

Gleason’s theorem (Gleason, 1957) [84] implements Kolmogorov’s axiomatic approach to probability by looking for an additive measure on Hilbert spaces. It leads to Born’s rule, but provides no physical insight into why the result should be regarded as probability. Clearly, it has not settled the issue: Rather, it is often cited [38,72,79,80] as a motivation to seek a physically transparent derivation of Born’s rule.

We shall now demonstrate how quantum entanglement leads to probabilities based on a symmetry, but—in contrast to subjective equal likelihood based on ignorance—on an *objective* symmetry of *known* quantum states.

### 3.1. Envariance

A pure entangled state of a system S and of another system (which we call “an environment E”, anticipating connections with decoherence) can be always written as:(3.1)$$|\psi_{\mathrm{SE}}\rangle =\sum_{k=1}^{N}a_{k}|s_{k}\rangle |\varepsilon_{k}\rangle \;.$$
Here $a_{k}$ are complex amplitudes while $\left\{|s_{k}\rangle \right\}$ and $\left\{|\varepsilon_{k}\rangle \right\}$ are orthonormal states in the Hilbert spaces $H_{S}$ and $H_{E}$. This *Schmidt decomposition* of a pure entangled $|\psi_{\mathrm{SE}}\rangle$ is a consequence of a theorem of linear algebra that predates quantum theory.

Schmidt decomposition demonstrates that any pure entangled bipartite state is a superposition of *perfectly correlated outcomes* of judiciously chosen measurements on each subsystem: Detecting $|s_{k}\rangle$ on S implies, with certainty, outcome $|\varepsilon_{k}\rangle$ for E, and *vice versa*.

Even readers unfamiliar with Equation (3.1) have likely relied on its consequences: Schmidt basis $\left\{|s_{k}\rangle \right\}$ appears on the diagonal of the reduced density matrix $\rho_{S}=Tr_{E}|\psi_{\mathrm{SE}}\rangle \langle \psi_{\mathrm{SE}}|$. (We have used this fact “in reverse” in the preceding section to “purify” mixed states.) But tracing is tantamount to averaging over states of the traced-out systems with weights given by the squares of their amplitudes. Therefore, physical interpretation of the resulting reduced density matrix (which is central in the usual treatments of decoherence) presumes Born’s rule we aim to derive (see [5] for discussion of how Born’s rule is used to justify physical significance of reduced density matrices). Consequently, we shall avoid employing tools of decoherence or relying on the statistical interpretation of $\rho_{S}$ in this section as this could introduce circularity. Instead, we derive Born’s rule from the symmetries of $|\psi_{\mathrm{SE}}\rangle$.

Symmetries reflect invariance. Rotation of a circle by an arbitrary angle, or of a square by multiples of π/2 are familiar examples. Entangled quantum states exhibit a new kind of symmetry—*entanglement—assisted invariance* or *envariance*: When a state $|\psi_{\mathrm{SE}}\rangle$ of a pair S,E can be transformed by $U_{S}=u_{S}⊗1_{E}$ acting solely on S,
(3.2)$$U_{S}|\psi_{\mathrm{SE}}\rangle =\left(u_{S}⊗1_{E}\right)|\psi_{\mathrm{SE}}\rangle =|\eta_{\mathrm{SE}}\rangle \;,\;$$
but the effect of $U_{S}$ can be undone by acting solely on E with an appropriately chosen $U_{E}=1_{S}⊗u_{E}$:(3.3)$$U_{E}|\eta_{\mathrm{SE}}\rangle =\left(1_{S}⊗u_{E}\right)|\eta_{\mathrm{SE}}\rangle =|\psi_{\mathrm{SE}}\rangle \;,\;$$
then $|\psi_{\mathrm{SE}}\rangle$ is called *envariant* under $U_{S}$ [7,35].

Envariance can be seen on any entangled $|\psi_{\mathrm{SE}}\rangle$. Any unitary operation diagonal in Schmidt basis $\left\{|s_{k}\rangle \right\}$:(3.4*a*)$$u_{S}=\sum_{k=1}^{N}\exp\left(i\phi_{k}\right)|s_{k}\rangle \langle s_{k}|\;,\;$$
is envariant: It can be undone by a *countertransformation*
(3.4*b*)$$u_{E}=\sum_{k=1}^{N}\exp\left(-i\phi_{k}\right)|\varepsilon_{k}\rangle \langle \varepsilon_{k}|,\;$$
acting solely on the environment.

In contrast to familiar symmetries (when a transformation has no effect on a state of an object) envariance is an *assisted symmetry*: The global state of SE *is* transformed by $U_{S}$, but it can be restored by acting on E, physically distinct (e.g., spatially separated) from S. When the global state of SE is envariant under some $U_{S}$, the local state of S alone must be obviously invariant under it. There are analogies between envariance and gauge symmetries, with E assuming the role of a gauge field.

Entangled states might seem an unusual starting point for the study of probabilities. After all, the textbook statement of Born’s rule deals with pure states of individual systems. Nevertheless, already in Schrödinger’s famous “cat” paper [20] the discussion of entangled quantum states leads to the realization that “Best possible knowledge of the whole does not necessarily lead to the same for its parts...”, as well as “The whole is in a definite state, the parts taken individually are not”. Our aim is to recast such essentially negative qualitative statements (which can be read as a realization of the limitations imposed by the tensor structure of quantum states on their predictive powers) into a derivation of Born’s rule, the quantitative tool used for predictions.

Entanglement is also the essence of decoherence responsible for the emergence of “the classical” in a quantum Universe. It is therefore natural to investigate symmetries of entangled quantum states, and explore their implication for how much can be known about parts when the whole is entangled. In addition, as we shall see below, envariance allows one to reassess the role of the environment and sheds new light on the origin of decoherence.

### 3.2. Decoherence as a Result of Envariance

Envariance of entangled states leads to our first conclusion: Phases of Schmidt coefficients are envariant under local (Schmidt) unitaries, Equations (3.4). Therefore, when a composite system is in an entangled state $|\psi_{\mathrm{SE}}\rangle$, the state of S (or E) alone is *invariant under the change of phases* of $a_{k}$. In other words, the *state* of S (understood as a set of all measurable properties of S alone) cannot depend on phases of Schmidt coefficients: It can depend only on their absolute values and on the outcome states—on the set of pairs $\left\{|a_{k}|,|s_{k}\rangle \right\}$. In particular (as we demonstrate below) probabilities cannot depend on these phases.

We have just seen that the loss of phase coherence between Schmidt states—decoherence—is a consequence of envariance: Decoherence is, after all, a selective loss of relevance of phases for the state of S. We stumbled here on its essence while exploring an unfamiliar territory, without the usual backdrop of dynamics and without employing trace and reduced density matrices.

However, decoherence viewed from the vantage point of envariance may look unfamiliar. What other landmarks of decoherence can we find without using trace and reduced density matrices (which rely on Born’s rule, something we do not yet have)? The answer is—all the essential ones [42,48,49]. We have already seen in Section 2 that pointer states (states that retain correlations, are predictable, and, hence, good candidates for classical domain) are singled out directly by repeatability—by the nature of information transfers. So, we already have a set of preferred pointer states and we have seen that when they are aligned with Schmidt basis, phases between them lose relevance for S alone. Indeed, models of decoherence ([7,24,25,27,28,29,32,33]) predict that after a brief (decoherence time) interlude Schmidt basis will typically settle down to coincide with pointer states determined through other criteria (such as predictability in spite of the coupling to the environment). There are exceptions that we have already mentioned in the previous section. They tend to arise when decoherence is incomplete, and/or when the reduced density matrix (that is diagonalized by Schmidt states) is so close to maximally mixed (i.e., some of its eigenvalues are nearly degenerate) that any complete set of states (including pointer states selected for their predictability) can express it while leaving it in a nearly diagonal form (see, e.g., [85]).

This encounter with decoherence on the way to Born’s rule is good omen: Quantum phases must be rendered irrelevant for the additivity of probabilities to replace additivity of complex amplitudes. Of course, one could postulate additivity of probabilities by fiat. This was done by Gleason (1957) [84], but such an assumption is at odds with the overarching additivity principle of quantum mechanics—with the principle of superposition (as is illustrated by the double slit experiment). So, if we set out to understand emergence of the classical domain from the quantum substrate defined by axioms (o)–(iii), additivity of probabilities should be derived (as it is done in Laplace’s approach, see Gnedenko, 1968 [52]) rather than imposed as an axiom (as it happens in Kolmogorov’s measure—theoretic approach, and in Gleason’s theorem).

Assuming decoherence to get $p_{k}={\left|\psi_{k}\right|}^{2}$ (Zurek, 1998 [36]; Deutsch, 1999 [86]; Wallace, 2003 [87]) would mean, at best, starting half way, and raises concerns of circularity [7,32,35,48,88,89,90] as the physical significance of the reduced density matrix—standard tool in the usual treatment of decoherence—is justified using Born’s rule. By contrast, envariant derivation, if successful, can be fundamental, independent of the usual tools of decoherence: It can justify, starting from the basic quantum postulates, the use of the trace and physical significance of reduced density matrices in the study of the quantum-classical transition.

As perceptive analysis by Drezet (2021) [91] shows, envariance has been recently adopted (Wallace, 2010; 2012) [19,92] even in the (modified) decision theory approach that initially dealt with states of a single system, and, therefore, encountered difficulties with circularity of the argument by invoking decoherence before establishing Born’s rule ([35,48,89,90]). We noted this problem already and will discuss it briefly below, in Section 3.7.

### 3.3. Swaps, Counterswaps, and Equiprobability

Envariance of pure states is purely quantum: Classical state of a composite classical system is given by a Cartesian (rather than tensor) product of its constituents. Therefore, to completely know a state of a composite classical system one must know a state of each subsystem. It follows that when one part of a classical composite system is affected by a transformation—a classical analogue of a swap $U_{S}$—state of the whole cannot be restored by acting on some other part. Hence, *pure classical states are never envariant*.

However, a mixed state (of, say, two coins) can mimic envariance: When we only know that a dime and a nickel are ‘same side up’, we can ‘undo’ the effect of the flip of a dime by flipping a nickel. This classical analogue depends on partial ignorance: To emulate envariance, we cannot know individual states of the two coins—just the fact that they are the same side up—just their correlation.

In quantum physics, tensor structure of states for composite systems means that ‘pure correlation’ is possible. We shall now prove that a maximally entangled “even” state with equal absolute values of Schmidt coefficients:(3.5)$$|{\bar{\psi }}_{\mathrm{SE}}\rangle \propto \sum_{k=1}^{N}e^{-i\phi_{k}}|s_{k}\rangle |\varepsilon_{k}\rangle$$
*implies* equal probabilities for any orthonormal set of states $|s_{k}\rangle$ of S and any corresponding set of $|\varepsilon_{k}\rangle$ of E.

Such an *even* state is envariant under a *swap* operation
(3.6*a*)$$u_{S}\left(k⇌l\right)=|s_{k}\rangle \langle s_{l}\left|+\right|s_{l}\rangle \langle s_{k}|\;.\;$$
A swap is a quantum version of the operation that exchanges two cards (Figure 2). It is a unitary that permutes states $|s_{k}\rangle$ and $|s_{l}\rangle$ of the system. A swap $|\mathrm{Heads}\rangle \langle \mathrm{Tails}|+|\mathrm{Tails}\rangle \langle \mathrm{Heads}|$ would flip a coin.

A swap on S is envariant when $|a_{k}\left|=\right|a_{l}|$ because $u_{S}\left(k⇌l\right)$ can be undone by a *counterswap* on E;
(3.6*b*)$$u_{E}\left(k⇌l\right)=e^{i(\phi_{k}-\phi_{l})}|\varepsilon_{l}\rangle \langle \varepsilon_{k}|+e^{-i(\phi_{k}-\phi_{l})}|\varepsilon_{k}\rangle \langle \varepsilon_{l}|,\;$$
as is seen in Figure 2c.

We want to *prove* that probabilities of envariantly swappable outcome states must be equal. But let us proceed with caution: Invariance under a swap is not enough—probability could depend on some other ‘intrinsic’ property of the state. For instance, in a superposition $|g\rangle +|e\rangle$, the ground and excited state can be invariantly swapped, as $|g\rangle +|e\rangle =|e\rangle +|g\rangle$, but *energies* of $|g\rangle$ and $|e\rangle$ are different. Why should probability—like energy—not depend on some intrinsic property[note 5] of the state?

Envariance can be used to prove that this cannot happen—that probabilities of envariantly swappable states are indeed equal. To this end we first define what is meant by “the state” and “the system” more carefully. Quantum role of these concepts is elucidated by three “Facts”—three definitions that recognize what is known about systems and their (mixed) states, but phrase it in a way that does not appeal to the Born-rule dependent tools and concepts (e.g., reduced density matrices):**Fact 1**:Unitary transformations must act on a system to alter its state. That is, when an operator does not act on the Hilbert space $H_{S}$ of S, (e.g., when it has a form $...⊗1_{S}⊗...$) the state of S does not change.**Fact 2**:The state of the system S is all that is needed (and all that is available) to predict measurement results, including probabilities of outcomes.**Fact 3**:The state of a larger composite system that includes S as a subsystem is all that is needed (and all that is available) to determine the state of S.

Note that the states defined this way need not be pure. In addition, note that Facts—while ‘naturally quantum’—are not in conflict with the role of states in classical physics.

Facts are a consequence of quantum theory. They are not, in any sense, additional assumptions. Rather, they clarify operational meanings of concepts (such as “a state”) that will play key role in the derivation of Born’s rule: Facts are the attributes that any sensible notion of a “state” (and, in particular, a “mixed state”) should possess. They also help distinguish purely quantum view based on the core quantum postulates from, e.g., hidden variable theory (where, for example, Fact 2 would not hold).
We can now *prove*:

**Theorem** **2.**
*When Schmidt coefficients satisfy $|a_{k}\left|=\right|a_{l}|$ in an even state $|{\bar{\psi }}_{\mathrm{SE}}\rangle \propto \sum_{k=1}^{N}e^{-i\phi_{k}}|s_{k}\rangle |\varepsilon_{k}\rangle$, Equation (3.5), the local state of S is invariant under a swap $u_{S}\left(k⇌l\right)=|s_{k}\rangle \langle s_{l}\left|+\right|s_{l}\rangle \langle s_{k}|$.*


**Proof.** Swap changes partners in the Schmidt decomposition (and, therefore, alters the global state). However, when the coefficients of the swapped outcomes differ only by a phase, a swap can be undone (without acting on S) by a corresponding counterswap, Equation (3.6b), in E. As the global state of SE is restored, it follows (from Fact 3) that the local state of S must have been also restored. However, (by Fact 1) the state of S could not have been affected by a counterswap acting only on E. So, (by Fact 2) the state of S must be left intact by a swap, in S, of Schmidt states that have the same absolute values of Schmidt coefficients.    □

We conclude that envariance of a pure global state of SE under swaps implies invariance of the corresponding local state of S. We could now follow Laplace, appeal to subjective indifference, apply equal likelihood, and “declare victory”—claim that subjective probabilities must be equal. However, as we have just seen with the example of the superposition $|g\rangle +|e\rangle$ of the eigenstates of energy, invariance of a local state under a swap implies only that the property associated with the swapped states (i.e., energies or probabilities) gets swapped when the states are swapped. Without an appeal to subjective ignorance this does not yet establish that the properties of interest (probabilities or energies) must be equal.

Entanglement (via envariance) allows us to get rid of such subjectivity altogether. The simplest way to prove the desired equality of probabilities is based on perfect correlation between the Schmidt states of S and E. These relative Schmidt states in, e.g., Equations (3.1) or (3.5) are orthonormal, and $|s_{k}\rangle$ are correlated with $|\varepsilon_{k}\rangle$ one-to-one. This implies the same probability for each member of every such Schmidt pair. Moreover (and for the very same reason—perfect correlation) after a swap on S probabilities of swapped states must be the same as probabilities of their two new partners in E. That is, after a swap $u_{S}\left(k⇌l\right)=|s_{k}\rangle \langle s_{l}\left|+\right|s_{l}\rangle \langle s_{k}|$ probability of $|s_{k}\rangle$ must be the same as probability $|\varepsilon_{l}\rangle$, and probability of $|s_{l}\rangle$ must be the same as that of $|\varepsilon_{k}\rangle$.

We now focus on an even state, Equation (3.5). It is envariant under all local unitaries (and, hence, under all swaps). Thus (by Fact 1) the state of E (and, by Fact 2, probabilities it implies) are not affected by swaps in S. So, *swapping Schmidt states of* S *exchanges their probabilities, and when the state is even it also keeps them the same!* This can be true only when the probabilities of envariantly swappable states are equal.

We can now state our conclusion:

**Corollary** **1.**
*When all N coefficients in the Schmidt decomposition have the same absolute values (as in the even states of Equation (3.5)), probability of each Schmidt state must be the same, and, therefore, by normalization, it is $p_{k}=1/N$.*


Readers may regard this as obvious, but (as recognized by Barnum (2003) [93], Schlosshauer and Fine (2005) [94], Drezet (2021) [91] and others) this is the key to Born’s rule. Equation (3.5) is envariant under swaps. This symmetry allowed us to extract physical consequences from quantum mathematics with a very minimal set of ingredients at hand. In the language of the Kolmogorov measure-theoretic axioms we have now established that—when the entangled state is even, Equation (3.5)—positive numbers (the ‘measures’ of probability for events corresponding to individual Schmidt states) must be equal.

Still, this may seem like a lot of work to arrive at something seemingly obvious: The case of unequal coefficients is our next goal. However—as we will see—it can be reduced to the equal coefficient case we have just settled. The symmetry of entanglement inherent in the equal coefficients case provides the crucial link between the quantum state vectors and the experimental consequences. Simple algebra along with the special case of probability—certainty—will lead us directly to Born’s rule[note 6].

We emphasize that in contrast to many other approaches to both classical and quantum probability, our envariant derivation is based not on a subjective assessment of an observer, but on an objective, experimentally verifiable symmetry of entangled states. Observer is forced to conclude that probabilities of local outcomes are equal not because of subjective ignorance, but because of certainty about something else: Certainty about the symmetries of the global state of the composite system implies—via symmetries of entanglement encapsulated in envariance—that local Schmidt states are equiprobable[note 7].

Envariant probability is also a probability of an individual event—there is no need for an ensemble of many events, so that relative frequency of “favorable” events can be used to define probability. Rather, we are decomposing the future possibilities into equiprobable alternatives, and deducing probability from the ratio of the number of favorable alternatives to the total. Above, just one favorable out of *N* equiprobable alternatives leads to $p_{k}=1/N$ for even states. We now extend this approach to general states.

### 3.4. Born’s Rule from Envariance

To illustrate general “finegraining” strategy of reducing cases with unequal coefficients to the previously described equal coefficient case (see (Zurek, 1998) [36] for its density matrix version) we start with an example involving a two-dimensional Hilbert space of the system spanned by states {|0〉,|2〉} and (at least) a three-dimensional Hilbert space of the environment:(3.7*a*)$$|\psi_{\mathrm{SE}}\rangle \;=\;\sqrt{\frac{2}{3}}{|0\rangle }_{S}{|+\rangle }_{E}\;\;+\;\;\sqrt{\frac{1}{3}}{|2\rangle }_{S}{|2\rangle }_{E}.\;$$
System is represented by the leftmost kets, and ${|+\rangle }_{E}={(|0\rangle }_{E}+{|1\rangle }_{E})/\sqrt{2}$ exists in (at least a two-dimensional) subspace of E that is orthogonal to the state ${|2\rangle }_{E}$, so that ${\langle 0|1\rangle }_{E}={\langle 0|2\rangle }_{E}={\langle 1|2\rangle }_{E}={\langle +|2\rangle }_{E}=0$. We already know we can ignore phases in view of their irrelevance for states of subsystems, so we omitted them above. From now on we shall also drop the overall normalization, as the probabilities will only depend on the relative values of the coefficients associated with the alternatives.

To reduce this case to an even state we extend “uneven” $|\psi_{\mathrm{SE}}\rangle$ above to a state $|{\bar{\Psi }}_{\mathrm{SEC}}\rangle$ with equal coefficients by letting E act on an ancilla C. (By Fact 1, since S is not acted upon, probabilities we shall infer for it cannot change.) Transformation into an even state can be accomplished by a generalization of controlled-not acting between E (control) and C (target), so that (in an obvious notation):$$\left|k\rangle \right|0^{\prime }\rangle \Rightarrow |k\rangle |k^{\prime }\rangle ,$$
leading to:(3.8*a*)$$(\sqrt{2}|0\rangle |+\rangle +|2\rangle |2\rangle )|0^{\prime }\rangle ⟹\sqrt{2}|0\rangle \frac{\left|0\rangle \right|0^{\prime }\rangle +|1\rangle |1^{\prime }\rangle }{\sqrt{2}}+\left|2\rangle |2\rangle \right|2^{\prime }\rangle =|{\bar{\Psi }}_{\mathrm{SCE}}\rangle .\;$$
Above, and from now on we skip subscripts: State of S will be always listed first, and state of C will be primed. The cancellation of $\sqrt{2}$ makes it obvious that this is an equal coefficient (“even”) state:(3.9*a*)$$|{\bar{\Psi }}_{\mathrm{SCE}}\rangle \propto |0,0^{\prime }\rangle |0\rangle +|0,1^{\prime }\rangle |1\rangle +|2,2^{\prime }\rangle |2\rangle .\;$$
Note that we have now combined state of S and C and (in the next step) we shall swap states of SC together.

Clearly, for joint states $|0,0^{\prime }\rangle ,|0,1^{\prime }\rangle$, and $|2,2^{\prime }\rangle$ of SC this is a Schmidt decomposition of (SC)E. The three orthonormal product states have coefficients with the same absolute value. THerefore, they can be envariantly swapped. It follows that the probabilities of these Schmidt states—$\left|0\rangle \right|0^{\prime }\rangle ,\;|0\rangle |1^{\prime }\rangle ,$ and $\left|2\rangle \right|2^{\prime }\rangle$—are all equal, so by normalization they are $\frac{1}{3}$. Moreover, and for the same envariant reason, the probability of state |2〉 of the system is $\frac{1}{3}$. As $|0\rangle$ and $|2\rangle$ are the only two outcome states for S; it also follows that probability of |0〉 must be $\frac{2}{3}$. Consequently:(3.10*a*)$$p_{0}=\frac{2}{3};\;\;p_{2}=\frac{1}{3}.\;$$
This is Born’s rule! Probabilities are proportional to the squares of the amplitudes from $|\psi_{\mathrm{SE}}\rangle$, Equation (3.7a).

Note that above we have avoided assuming additivity of probabilities: $p_{0}=\frac{2}{3}$ not because it is a sum of two fine-grained alternatives each with probability $\frac{1}{3}$, but rather because there are only two (mutually exclusive and exhaustive) alternatives for S; $|0\rangle$ and $|2\rangle$, and it was separately established that $p_{2}=\frac{1}{3}$. So, by normalization, $p_{0}=1-\frac{1}{3}$.

Bypassing appeal to additivity of probabilities is essential in interpreting theory with another principle of additivity—quantum superposition principle—which trumps additivity of probabilities or at least classical intuitive ideas about what should be additive (e.g., in the double slit experiment). Here, this conflict is averted: Probabilities of Schmidt states can be added because of the loss of phase coherence that follows directly from envariance, as we have established earlier, and as was discussed in (Zurek, 2005) [48]. We return to this point in the additivity Lemma below.

Consider now a more general case of arbitrary coefficients. For simplicity we focus on entangled state with just two non-zero Schmidt coefficients:(3.7*b*)$$|\psi_{\mathrm{SE}}\rangle =\alpha |0\rangle |\varepsilon_{0}\rangle +\beta |1\rangle |\varepsilon_{1}\rangle \;,\;$$
and assume $\alpha =\sqrt{\frac{\mu }{\mu +\nu }};\;\beta =\sqrt{\frac{\nu }{\mu +\nu }}$, with integer μ, ν.

As before, the strategy is to convert a general entangled state into an even state, and then to apply envariance under swaps. To implement it, we assume E has sufficient dimensionality[note 8] to allow decomposition of $|\varepsilon_{0}\rangle$ and $|\varepsilon_{1}\rangle$ in a different orthonormal basis $\left\{|e_{k}\rangle \right\}$:$$|\varepsilon_{0}\rangle =\sum_{k=1}^{\mu }|e_{k}\rangle /\sqrt{\mu };\;\;\;\;\;|\varepsilon_{1}\rangle =\sum_{k=\mu +1}^{\mu +\nu }|e_{k}\rangle /\sqrt{\nu .}$$

Envariance we need is associated with counterswaps of E that undo swaps of the joint state of the composite system SC. To exhibit it, we let ancilla C interact with E as before, e.g., by employing E as a control to carry out a ‘controlled-not - like’ operation;
$$|e_{k}\rangle |c_{0}\rangle \to |e_{k}\rangle |c_{k}\rangle ,$$
where $|c_{0}\rangle$ is the initial state of C in some suitable orthonormal basis $\left\{|c_{k}\rangle \right\}$. Thus;
(3.8*b*)$$|{\bar{\Psi }}_{\mathrm{SCE}}\rangle \propto \sqrt{\mu }|0\rangle \sum_{k=1}^{\mu }\frac{|c_{k}\rangle |e_{k}\rangle }{\sqrt{\mu }}+\sqrt{\nu }|1\rangle \sum_{k=\mu +1}^{\mu +\nu }\frac{|c_{k}\rangle |e_{k}\rangle }{\sqrt{\nu }}.\;$$
Such CE interaction can happen far from S, so by Fact 1 it cannot influence probabilities in S. $|{\bar{\Psi }}_{\mathrm{SCE}}\rangle$ is envariant under swaps of states $|s,c_{k}\rangle$ (where *s* stands for 0 or 1, as needed) in the composite system SC. This is even more apparent after the obvious cancellations;
(3.9*b*)$$|{\bar{\Psi }}_{\mathrm{SCE}}\rangle \propto \sum_{k=1}^{\mu }|0,c_{k}\rangle |e_{k}\rangle +\sum_{k=\mu +1}^{\mu +\nu }|1,c_{k}\rangle |e_{k}\rangle .\;$$
Hence, $p_{0,k}=p_{1,k}=\frac{1}{\mu +\nu }$. Therefore, it follows that the probabilities of |0〉 and |1〉 are:(3.10*b*)$$p_{0}=\frac{\mu }{\mu +\nu }={\left|\alpha \right|}^{2};\;\;\;p_{1}=\frac{\nu }{\mu +\nu }={\left|\beta \right|}^{2}\;.\;$$
Born’s rule thus emerges here from the most quantum aspects of the theory—entanglement and envariance.

In contrast with other approaches, probabilities in our envariant derivation are a consequence of complementarity, of the incompatibility of the purity of the entangled state of the whole with the purity of the states of parts. Born’s rule appears in a completely quantum setting, without any *a priori* imposition of symptoms of classicality that would violate the spirit of quantum theory.

Envariant derivation (in contrast to Gleason’s successful but unphysical proof and in contrast to unsuccessful “frequentist” attempts in the Everettian “Many-Worlds” setting) does not require additivity as an assumption: The strategy that bypasses appeal to additivity used in the simple case of Equation (3.10a) can be generalized (for details see below and Zurek, 2005 [48]). The assumption of additivity is not needed. In a quantum setting this is an important advance. Additivity of probabilities is a consequence of the envariance of phases of Schmidt coefficients that leads to decoherence. The case of more than two outcomes is straightforward, as is extension by continuity to incommensurate probabilities.

#### 3.4.1. Additivity of Probabilities from Envariance

Kolmogorov’s axiomatic formulation of the probability theory (see Gnedenko, 1968 [52]) as well as the proof of Born’s rule due to Gleason (1957) [84] *assume* additivity of probabilities. This assumption is motivated by the assertion that probability is a *measure*. On the other hand, in the standard approach of Laplace (1820) [69] additivity can be established starting from the definition of probability of a composite event as a fraction of the favorable equiprobable events to the total (see discussion in Gnedenko, 1968 [52]). The key ingredient that makes this derivation of additivity possible is equiprobability

We have already established—using *objective* symmetries (in contrast to Laplace, who had to rely on the subjective ‘state of mind’)—that envariantly swappable events are equiprobable. We can now follow Laplace’s strategy and use equiprobability along with decoherence already justified directly by envariance (see Section 3.2) to prove additivity. This is important, as additivity of probabilities should not be automatically and uncritically adopted in the quantum setting. After all, quantum theory is based on the principle of superposition, our core postulate (i): The principle of superposition asserts supremacy of the additivity of state vectors which is *prima facie* incompatible with additivity of probabilities, as is illustrated by the double slit experiment.

Phases between the record (pointer) states (or, more generally, between any set of Schmidt states) do not influence the outcome of any local measurement that can be carried out on the apparatus (or on decohered records in the memory of the observer). This independence of the local state from the global phases in the Schmidt decomposition invalidates the principle of superposition when the system of interest (or the pointer of the apparatus, or the memory of the observer) are ‘open’, entangled with their environments. Therefore, we can now *establish* (rather than postulate) the probability of a composite event:

**Lemma** **1.**
*Probability of a composite (coarse-grained) event consisting of a subset*

(3.11)
$$\kappa \equiv \left\{k_{1}\vee k_{2}\vee \cdots \vee k_{n_{\kappa }}\right\}\;$$

*of $n_{\kappa }$ of the total N envariantly swappable mutually exclusive exhaustive fine-grained events associated with records corresponding to pointer states of the global state;*

$$|{\bar{\Psi }}_{\mathrm{SAE}}\rangle \;=\;\sum_{k=1}^{N}e^{i\phi_{k}}|s_{k}\rangle |A_{k}\rangle |\varepsilon_{k}\rangle \;=\;\sum_{k=1}^{N}e^{i\phi_{k}}|s_{k},A_{k}\left(s_{k}\right)\rangle |\varepsilon_{k}\rangle ,$$

*is given by:*

(3.12)
$$p\left(\kappa \right)=\frac{n_{\kappa }}{N}\;.\;\left(3.12\right)$$



To prove additivity of probabilities using envariance we consider the state:(3.13)$$|\Upsilon_{\bar{A}\mathrm{ASE}}\rangle \;\propto \;\sum_{\kappa }|A_{\kappa }^{\in }\rangle \sum_{k\in \kappa }|A_{k}\rangle |s_{k}\rangle |\varepsilon_{k}\rangle \;$$
representing both the fine-grained and coarse-grained records. The coarse-graining is implemented by the apparatus $\bar{A}$ with pointer states $|A_{\kappa }^{\in }\rangle$.

We first note that the form of $|\Upsilon_{\bar{A}\mathrm{ASE}}\rangle$ justifies assigning zero probability to $|s_{j}\rangle$’s that do not appear—i.e., appear with zero amplitude—in the initial state of the system. Quite simply, there is no state of the observer with a record of such zero-amplitude Schmidt states of the system in $|\Upsilon_{\bar{A}\mathrm{ASE}}\rangle$, Equation (3.13).

To establish this Lemma we exploit basic implications of envariance: When there are total *N* envariantly swappable outcome states, and they exhaust all of the possible outcomes, each should be assigned probability of 1/N. We also note that when coarse-grained events are defined via $|A_{\kappa }^{\in }\rangle$ as unions of fine-grained events, the conditional probability of the coarse grained event is:(3.14*a*)p(κ|k)=1k∈κ,
(3.14*b*)p(κ|k)=0k∉κ.
To demonstrate the above Lemma we need one more property—the fact that when an event U that is certain (p(U)=1) can be decomposed into two mutually exclusive events,
(3.15)$$U=\kappa \vee \kappa^{⊥},$$
their probabilities must add to unity:(3.16)$$p\left(U\right)=p\left(\kappa \vee \kappa^{⊥}\right)=p\left(\kappa \right)+p\left(\kappa^{⊥}\right)=1\;.$$
This assumption introduces (in a very limited setting) additivity. It is equivalent to the statement that “something will certainly happen”. Note that this very limited version of additivity holds in quantum setting (i.e., it is not challenged by the double slit experiment, and, more generally, by the superposition principle, providing that the events are indeed mutually exclusive).

**Proof.** Proof of the Lemma starts with the observation that probability of any composite event κ of the form of Equation (3.11) can be obtained recursively—by subtracting, one by one, probabilities of all the fine-grained events that belong to $\kappa^{⊥}$, and exploiting the consequences of the implication, Equation (3.14)–(3.16). Thus, as a first step, we have:
$$p(\left\{k_{1}\vee k_{2}\vee \cdots \vee k_{n_{\kappa }}\vee \cdots \vee k_{N-1}\right\}+p\left(k_{N}\right)=1\;.$$
Moreover, for all fine-grained events $p\left(k\right)=\frac{1}{N}$. Hence;
$$p(\left\{k_{1}\vee k_{2}\vee \cdots \vee k_{n_{\kappa }}\vee \cdots \vee k_{N-1}\right\}=1-\frac{1}{N}\;.$$
Furthermore (and this is the next recursive step) the conditional probability of the event $\left\{k_{1}\vee k_{2}\vee \cdots \vee k_{n_{\kappa }}\vee \cdots \vee k_{N-2}\right\}$ given the event $\left\{k_{1}\vee k_{2}\vee \cdots \vee k_{n_{\kappa }}\vee \cdots \vee k_{N-1}\right\}$ is:
$$p\left(\left\{k_{1}\vee k_{2}\vee \cdots \vee k_{N-2}\right\}|\left\{k_{1}\vee k_{2}\vee \cdots \vee k_{N-1}\right\}\right)=1-\frac{1}{N-1},$$
and so the unconditional probability must be:
$$p\left(\left\{k_{1}\vee k_{2}\vee \cdots \vee k_{n_{\kappa }}\vee \cdots \vee k_{N-2}\right\}|U\right)=\left(1-\frac{1}{N}\right)\left(1-\frac{1}{N-1}\right).$$
Repeating this procedure untill only the desired composite event κ remains we have:
(3.17)$$p\left(\left\{k_{1}\vee k_{2}\vee \cdots \vee k_{n_{\kappa }}\right\}\right)=\left(1-\frac{1}{N}\right)\cdots \left(1-\frac{1}{N-(N-n_{\kappa }-1)}\right).$$
After some elementary algebra we finally recover:
$$p\left(\left\{k_{1}\vee k_{2}\vee \cdots \vee k_{n_{\kappa }}\right\}\right)=\frac{n_{\kappa }}{N}\;.$$
Hence, Equation (3.12) holds.    □

**Corollary** **2.**
*Probability of mutually compatible exclusive events κ,λ,μ,⋯ that can be decomposed into unions of envariantly swappable elementary events are additive:*

(3.18)
p(κ∨λ∨μ∨⋯)=p(κ)+p(λ)+p(μ)+⋯



Note that in establishing additivity Lemma we have only considered situations that can be reduced to certainty or impossibility (that is, cases corresponding to the absolute value of the scalar product equal to 1 and 0). This is in keeping with our strategy of deriving probability and, in particular, of arriving at Born’s rule from certainty and symmetries.

#### 3.4.2. Algebra of Records as the Boolean Algebra of Events

Algebra of events (see, e.g., Gnedenko, 1968 [52]) can be then defined by simply identifying events with records inscribed in the coarse-grained pointer states such as $|A_{\kappa }^{\in }\rangle$ in Equation (3.13) of the apparatus $\bar{A}$. Logical product of any two coarse-grained events κ,λ corresponds to the product of the projection operators that act on the memory Hilbert space—on the corresponding records:(3.19*a*)$$\kappa \wedge \lambda \overset{\mathrm{def}}{\;=}P_{\kappa }P_{\lambda }=P_{\kappa \wedge \lambda }\;.\;$$
Logical sum is represented by a projection onto the union of the Hilbert subspaces:(3.19*b*)$$\kappa \vee \lambda \overset{\mathrm{def}}{\;=}P_{\kappa }+P_{\lambda }-P_{\kappa }P_{\lambda }=P_{\kappa \vee \lambda }\;.\;$$
Last not least, a complement of the event κ corresponds to:(3.19*c*)$$\kappa^{⊥}\overset{\mathrm{def}}{\;=}P_{U}-P_{\kappa }=P_{\kappa^{⊥}}\;.\;$$
With this set of definitions it is now fairly straightforward to show:

**Theorem** **3.**
*Events corresponding to the records stored in the memory pointer states define a Boolean algebra.*


**Proof.** To show that the algebra of records is Boolean we need to show that coarse—grained events satisfy any of the (several equivalent, see, e.g., Sikorski, 1964 [95]) sets of axioms that define Boolean algebras:
(a)Commutativity:
(3.20*a, a′*)$$P_{\kappa \vee \lambda }=P_{\lambda \vee \kappa };\;\;\;P_{\kappa \wedge \lambda }=P_{\lambda \wedge \kappa }\;.\;$$(b)Associativity:
(3.20*b, b′*)$$P_{(\kappa \vee \lambda )\vee \mu }=P_{\lambda \vee (\kappa \vee \mu )};\;\;\;P_{(\kappa \wedge \lambda )\wedge \mu }=P_{\lambda \wedge (\kappa \wedge \mu )}\;.$$(c)Absorptivity:
(3.20*c, c′*)$$P_{\kappa \vee (\lambda )\wedge \lambda )}=P_{\kappa };\;\;\;P_{\kappa \vee (\lambda \wedge \kappa )}=P_{\kappa }\;.$$(d)Distributivity:
(3.20*d, d′*)$$P_{\kappa \vee (\lambda \wedge \mu )}=P_{\left(\kappa \vee \lambda )\wedge (\kappa \vee \mu \right)};\;\;\;P_{\kappa \wedge (\lambda \vee \mu )}=P_{\left(\kappa \wedge \lambda )\vee (\kappa \wedge \mu \right)}\;.$$(e)Orthocompletness:
(3.20*e, e′*)$$P_{\kappa \vee (\lambda \wedge \lambda^{⊥})}=P_{\kappa };\;\;\;P_{\kappa \wedge (\lambda \vee \lambda^{⊥})}=P_{\kappa }\;.$$Proofs of (a)–(e) are straightforward manipulations of projection operators. We leave them as an exercise to the interested reader. As an example we give the proof of distributivity: $P_{\kappa \wedge (\lambda \vee \mu )}=P_{\kappa }\left(P_{\lambda }+P_{\mu }-P_{\lambda }P_{\mu }\right)=P_{\kappa }P_{\lambda }+P_{\kappa }P_{\mu }-{\left(P_{\kappa }\right)}^{2}P_{\lambda }P_{\mu }=P_{\kappa \wedge \lambda }+P_{\kappa \wedge \mu }-P_{\kappa \wedge \lambda }P_{\kappa \wedge \mu }=P_{\left(\kappa \wedge \lambda )\vee (\kappa \wedge \mu \right)}\;.$ The other distributivity axiom is demonstrated equally easily.    □

These record projectors commute because records are associated with the orthonormal pointer basis of the memory of the observer or of the apparatus: It is impossible to consult memory cell in any other basis, so the problems with distributivity pointed out by Birkhoff and von Neumann simply do not arise—when records are kept in orthonormal pointer states, there is no need for ‘quantum logic’.

Theorem 3 entitles one to think of the outcomes of measurements—of the records kept in various pointer states—in classical terms. Projectors corresponding to pointer subspaces define overlapping but compatible volumes inside the memory Hilbert space. Algebra of such composite events (defined as coarse grained records) is indeed Boolean. The danger of the loss of additivity (which in quantum systems is intimately tied to the principle of superposition) has been averted: Distributive law of classical logic holds.

### 3.5. Inverting Born’s Rule: Why Is the Amplitude a Square Root of the Frequency of Occurrence?

The strategy we have pursued above to derive Born’s rule for unequal coefficients was to consider two different splits, SE|C and SC|E of the same composite tripartite system SEC. In the beginning we had a bipartite state of SE with unequal coefficients, Equations (3.7). We have finegrained it into an equal coefficient envariantly swappable state by introducing an additional system C, which entangled with SE in such a way that the resulting state of SEC was even, Equations (3.9). In that even state the finegrained probabilities were provably equal. One could then count the number of contributions that included the two alternatives of interest—states $|0\rangle$ and $|1\rangle$ of S. Unequal probabilities were proportional to the number of equiprobable fine-grained contributions in the resulting even states, Equations (3.9).

This reasoning can be reversed: We shall now use the strategy of moving the line dividing the two subsystems of interest in a tripartite composite system to show that the amplitudes must be proportional to the square roots of the frequencies of occurrence of the corresponding events.

We consider the probability of getting a count of *m* 1’s in a measurement by an apparatus A on an ensemble of identically prepared two-state systems:$$|{\overset{˘}{\psi }}_{S}\rangle =\overset{M}{\underset{k=1}{⨂}}(\alpha |0\rangle +\beta |1\rangle {)}_{k}.$$
In course of the (pre-)measurement memory cells of A entangle with S, so, in obvious notation, the resulting state is a product of *M* identical copies:(3.21*a*)$$|{\overset{˘}{\Psi }}_{\mathrm{SA}}\rangle =\overset{M}{\underset{k=1}{⨂}}\left(\alpha |0\rangle \right|a_{0}\rangle +\beta |1\rangle |a_{1}\rangle {)}_{k}=\overset{M}{\underset{k=1}{⨂}}{|\Psi_{\mathrm{SA}}\rangle }_{k}.$$

We shall work with the case of α=β to avoid cumbersome notation. With these simplifications the state vector $|{\overset{˘}{\Psi }}_{\mathrm{SA}}\rangle_{k}=(|0\rangle |a_{0}\rangle +1\rangle |a_{1}{\rangle )}_{k}$ is envariant under a swap $(|0\rangle \langle 1|+{|1\rangle \langle 0|)}_{k}$ acting on *k*’s member of the ensemble, as such a swap can be undone by a counterswap $(|a_{0}\rangle \langle a_{1}\left|+\right|a_{1}\rangle {\langle a_{0}|)}_{k}$ acting on A. This envariance must also hold when the state vector is expanded into a sum:(3.21*b*)$$|{\overset{˘}{\Psi }}_{\mathrm{SA}}\rangle \propto \sum_{m=0}^{M}|{\tilde{s}}_{m}\rangle .$$
In this expression, each unnormalized vector $|{\tilde{s}}_{m}\rangle$ represents all sequences of outcomes and records that have resulted in *m* detections of “1”, that is;
(3.22)$$\begin{array}{ccc} & & |{\tilde{s}}_{0}\rangle =\left|00...0\rangle \right|A_{00...0}\rangle \\ & & |{\tilde{s}}_{1}\rangle =\left|10...0\rangle \right|A_{10...0}\rangle +|01...0\rangle |A_{01...0}\rangle +...+|00...1\rangle |A_{00...1}\rangle \\ & & .....\\ & & |{\tilde{s}}_{m}\rangle =\left|11...10...0\rangle \right|A_{11...10...0}\rangle +...+|00...01...1\rangle |A_{00...01...1}\rangle \\ & & .....\\ & & |{\tilde{s}}_{M}\rangle =\left|11...1\rangle \right|A_{11...1}\rangle . \end{array}$$
The memory state of A contains record of all the outcomes—it is the product of memory states of individual cells, e.g., $|A_{10...0}\rangle =|a_{1}\rangle_{1}|a_{0}\rangle_{2}...{|a_{0}\rangle }_{M}$. All the sequences of the outcomes and the corresponding sequences of the records are orthonormal. Therefore, the sum of the sequences of outcome states and corresponding record states—Equation (3.21b) expressed in terms of Equation (3.22)—constitutes a Schmidt decomposition of $|{\overset{˘}{\Psi }}_{\mathrm{SA}}\rangle$.

The number of distinct outcome sequence states in $|{\tilde{s}}_{m}\rangle$ is M!m!(M−m)!=Mm. Every outcome sequence state is equiprobable since it can be envariantly swapped with any other outcome sequence state. Therefore, the probability of detecting *m* 1’s must be proportional to Mm, the number of equiprobable records with *m* detections of 1.[note 9] Moreover, as we have set α=β, the relative normalization of every such sequence is the same. Consequently, every permutation of the outcomes—every specific sequence of 0’s and 1’s, regardless of the number of 1’s—has a probability of $2^{-M}$. This includes “maverick” states with the unlikely total counts such as $|{\tilde{s}}_{0}\rangle$ and $|{\tilde{s}}_{M}\rangle$.

We now relate probability of a specific count of *m* 1’s to the amplitude of the corresponding state. We prepare to address this question by adding a quantum system that counts 1’s and enters their number in the *register*R. The result is an entangled state of S, A, and R:(3.23*a*)$$|{\overset{˘}{\Upsilon }}_{\mathrm{SAR}}\rangle \propto \sum_{m=0}^{M}|{\tilde{s}}_{m}{\rangle |m\rangle }_{R}.$$
Above, ${|m\rangle }_{R}$ are the orthogonal states of R with distinct total counts.

We shall now use envariance of $|{\overset{˘}{\Upsilon }}_{\mathrm{SAR}}\rangle$ to relate probability of a specific count *m* to the amplitude of the corresponding state ${|m\rangle }_{R}$. To this end we first normalize states $|{\tilde{s}}_{m}\rangle$: Without normalization, amplitudes we are trying to deduce would have no meaning.

Normalizing states of SA is not difficult: Every individual sequence of 0’s and 1’s that corresponds to a possible records has the same norm, and the number of sequences that yield total count of *m* 1’s determines the norm of $|{\tilde{s}}_{m}\rangle$;
〈s˜m|s˜m〉∝Mm.
This is a first step in a purely *mathematical* operation that converts $|{\tilde{s}}_{m}\rangle$ into the corresponding normalized state $|s_{m}\rangle$ that can be later legally used to implement the Schmidt decomposition.

It is now easy to see that states;
(3.24)|sm〉=Mm−12|s˜m〉,
have the same normalization. The state of the whole ensemble is then:(3.23*b*)|Υ˘SAR〉∝∑m=0MMm12|sm〉|m〉R=∑m=0Mγm|sm〉|m〉R.
This is also a Schmidt decomposition, as $|s_{m}\rangle$ and ${|m\rangle }_{R}$ are orthonormal. Given our previous discussion we already know that the probability $p_{m}$ of any specific count *m* is given by the fraction of such sequences. That is:pm=2−MMm.
This follows from counting of the number of envariant (and, hence, equiprobable) permutations of 0’s and 1’s contributing to $|s_{m}\rangle$ and, hence, corresponding to ${|m\rangle }_{R}$.

Indeed, Equation (3.23b) is a coarse-grained version of Equations (3.22) and (3.23a). So, the above expression for $|{\overset{˘}{\Upsilon }}_{\mathrm{SAR}}\rangle$ shows that the amplitude $\gamma_{m}$ of ${|m\rangle }_{R}$—of the state of the register that holds the count of *m* 1’s—is proportional to the square root of the number of equiprobable sequences that lead to that count;
(3.25*a*)|γm|∝Mm=M!m!(M−m)!,
or;
(3.25*b*)$$|\gamma_{m}|={\sqrt{p}}_{m}\;.$$
Equation (3.25) is the main result of our discussion. We have now deduced that absolute values $|\gamma_{m}|$ of the Schmidt coefficients are proportional to the *square roots* of relative frequencies—to the square roots of the cardinalities of subsets of $2^{M}$ equiprobable sequences that yield such ‘total count = *m*’ of composite events. In a sense, our calculation “inverts” the derivation of Born’s rule we have presented before.

As in the earlier derivation of Born’s rule, the key was to express the same tripartite global state $|{\overset{˘}{\Upsilon }}_{\mathrm{SAR}}\rangle$ as two different Schmidt decompositions. Thus,
(3.26*a*)$$\begin{array}{ccc} \;\;\;\;|{\overset{˘}{\Upsilon }}_{S|\mathrm{AR}}\rangle & \propto & |00...0\rangle (|A_{00...0}\rangle {|0\rangle }_{R})\\ & + & |10...0\rangle (|A_{10...0}\rangle {|1\rangle }_{R})+|01...0\rangle (|A_{01...0}\rangle {|1\rangle }_{R})+\\ ... & + & |00...1\rangle (|A_{00...1}\rangle {|1\rangle }_{R})\\ ......\\ & + & |11...1100...00\rangle (|A_{11...1100...00}\rangle {|m\rangle }_{R})+\\ ... & + & |00...0011...11\rangle (|A_{00...0011...11}\rangle {|m\rangle }_{R})\\ ......\\ & + & |11...1\rangle (|A_{11...1}\rangle {|M\rangle }_{R}), \end{array}$$
for the split S|AR of the whole into two subsystems, and;
(3.26*b*)|Υ˘SA|R〉∝∑m=0MMm|sm〉|m〉R=∑m=0Mγm|sm〉|m〉R,
for the alternative SA|R.

The location of the border between the two parts of the whole SAR is the key difference. It redefines “events of interest”. The top $|{\overset{˘}{\Upsilon }}_{S|\mathrm{AR}}\rangle$ treats binary sequences of outcomes as “events of interest”, and, by envariance, assigns equal probabilities $2^{-M}$ to each outcome sequence state. By contrast, in $|{\overset{˘}{\Upsilon }}_{\mathrm{SA}|R}\rangle$ the total count *m* is an “event of interest”, but now its probability can be deduced from $|{\overset{˘}{\Upsilon }}_{S|\mathrm{AR}}\rangle$, as both Schmidt decompositions represent the same state—the same physical situation. This implies (a converse of) Born’s rule: Amplitude of a state ${|m\rangle }_{R}$ of the register R is proportional to the square root of the number of sequences of counts that yield *m*.

Generalization to when α≠β is conceptually simple (although notationally cumbersome). Global state after the requisite adjustment of the relative normalizations is:|Υ˘SAR〉∝∑m=0MMm12αM−mβm|sm〉|m〉R=∑m=0MΓm|sm〉|m〉R.
Coefficients $\Gamma_{m}$ that multiply $|s_{m}{\rangle |m\rangle }_{R}$ combine on equal footing preexisting amplitudes α and β of |0〉 and |1〉 from the initial state, $|{\overset{˘}{\psi }}_{S}\rangle ={⨂}_{k=1}^{M}(\alpha |0\rangle +{\beta |1\rangle )}_{k}$ with the square roots of Newton’s symbols that arise from counting—with the numbers of the corresponding outcome sequences. Once the state representing the whole ensemble is written as $\sum_{m=0}^{M}\Gamma_{m}|s_{m}{\rangle |m\rangle }_{R}$, the origin of the coefficients $\Gamma_{m}$ (or $\gamma_{m}$ before) is irrelevant: Observer presented with a state $\sum_{m=0}^{M}\Gamma_{m}|s_{m}{\rangle |m\rangle }_{R}$ and asked to assess probabilities of outcomes $|s_{m}{\rangle |m\rangle }_{R}$ has no reason to delve into combinatorial origins of $\Gamma_{m}$. For a measurement with outcome states $|s_{m}{\rangle |m\rangle }_{R}$ the origin of the amplitudes $\Gamma_{m}$ that multiply them do not matter. Their absolute values, however, do matter: Observer could implement envariant derivation “from scratch”, starting with whatever coefficients are there in the initial state, and finegraining (as before, Equations (3.8)), to deduce probabilities of various outcomes.

### 3.6. Relative Frequencies from Relative States

We shall now use envariance to deduce relative frequencies from amplitudes. In view of the discussion immediately above the relation between amplitudes and frequencies is already apparent, so this may seem superfluous, but we shall sketch it anyway “for the sake of completeness”, and also because it provides a different—experimentally motivated, one could say—point of view of the alternatives. A much more complete derivation of relative frequencies from envariance is also available in Ref. [48].

We emphasize that we do not need relative frequencies to define probabilities: Probabilities are already in place. They are “single shot”, defined not by counting the number of “favorable events” (as in the relative frequency approach), but, rather, by first establishing equiprobability of a certain class of events, and then by counting the number of equiprobable favorable possibilities. Therefore, the calculation immediately below is, in a sense, only a consistency check.

Consider *M* distinguishable SCE triplets, each already in the fine-grained state;
$$|{\bar{\Psi }}_{\mathrm{SCE}}^{\left(\ell \right)}\rangle ={\left(\sum_{k=1}^{\mu }|0,c_{k}\rangle |e_{k}\rangle +\sum_{k=\mu +1}^{\mu +\nu }|1,c_{k}\rangle |e_{k}\rangle \right)}^{\left(\ell \right)},$$
of Equation (3.9). The state of the whole ensemble is then given by their product;
(3.27)$$|{\overset{˘}{\Omega }}_{\mathrm{SCE}}^{M}\rangle =\overset{M}{\underset{\ell =1}{⨂}}|{\bar{\Psi }}_{\mathrm{SCE}}^{\left(\ell \right)}\rangle =\overset{M}{\underset{\ell =1}{⨂}}(\sum_{k=1}^{\mu }|0,c_{k}\rangle |e_{k}\rangle +\sum_{k=\mu +1}^{\mu +\nu }|1,c_{k}\rangle |e_{k}{\rangle )}^{\left(\ell \right)}.$$
As in the derivation of the inverse of Born’s rule, we can now carry out the product and obtain a sum that will contain ${\left(\mu +\nu \right)}^{M}$ (instead of just $2^{M}$) terms. As in Equation (3.26a), these terms can be now sorted according to the number of 0’s and 1’s they contain. We could even attach a register R, and repeat all the steps we have taken above. We shall bypass these intermediate calculations that are conceptually straightforward but notationally cumbersome. What matters in the end is how many equiprobable terms contain, say, *m* 1’s. The answer is clearly, MmμM−mνm. Therefore, the probability of detecting *m* 1’s in *M* trials is given by:(3.28)pM(m)=MmμM−mνm(μ+ν)M=Mm|α|2(M−m)|β|2m.
We now assume *M* is large, not because envariant derivation requires this—we have already obtained Born’s rule for individual events, M=1—but because the relative frequency approach needs it (von Mises, 1939 [71]; Gnedenko, 1968 [52]). In that limit binomial can be approximated by a Gaussian:(3.29)$$p_{M}\left(m\right)≃\frac{\exp-\frac{1}{2}(\frac{m-{\left|\beta \right|}^{2}M}{\sqrt{M}\left|\alpha \beta \right|}{)}^{2}}{\sqrt{2\pi M}\left|\alpha \beta \right|}.$$

The average number of 1’s is, according to Equation (3.29), $\langle m\rangle ={\left|\beta \right|}^{2}M$, as expected, establishing a link between relative frequency of events in a large number of trials and Born’s rule. This connection between quantum states and relative frequencies does not rest either on circular and *ad hoc* assumptions that relate size of the coefficients in the global state to probabilities (e.g., by asserting that probability corresponding to a small enough amplitude is 0 (Geroch 1984) [73]), or modifications of quantum theory (Weissman, 1999 [96]; Buniy, Hsu, and Zee, 2006 [83]), or on the unphysical infinite limit (Hartle, 1968 [79]; Farhi, Goldstone, and Guttmann, 1989 [80]). Such steps have left past frequentist approaches to Born’s rule (including also these of Everett, DeWitt, and Graham) open to criticism (Stein, 1984 [74]; Kent, 1990 [38]; Squires, 1990 [39]; Joos, 2000 [75]; Auletta, 2000 [97]).

Note that we avoid the problem of two independent measures of probability (number of branches *and* size of the coefficients) that derailed previous relative state attempts. We simply count the number of *envariantly swappable* (and hence *provably equiprobable*) sequences of potential events. This settles the issue of “maverick universes”—atypical branches with numbers of e.g., 0’s very different from the average 〈n〉. They are there (as they should be) but they are very improbable. This is established through a physically motivated envariance under swaps. So, maverick branches did not have to be removed either “by assumption” (DeWitt, 1970 [14]; 1971 [15]; Graham, 1973 [72]; Geroch, 1984 [73]) or by an equally unphysical M=∞. Maverick branches are there, but pose no threat to our envariant derivation.

### 3.7. Envariance—An Overview

Envariance settles a major outstanding quantum foundational problem: The origin of probabilities. Born’s rule can be now established on a basis of a solid and simple physical reasoning, and without assuming additivity of probabilities (in contrast to Gleason, 1957 [84]). We have derived $p_{k}={\left|\psi_{k}\right|}^{2}$ without relying on the tools of decoherence.

There were other attempts to apply Laplacean strategy of invariance under permutations to prove “indifference”. This author (Zurek, 1998) [36] noted that all of the possibilities listed on a diagonal of a unit density matrix (e.g., ∼|0〉〈0|+|1〉〈1|) must be equiprobable, as it is invariant under swaps. This equiprobability—based approach was then extended to the case of unequal coefficients by finegraining, and leads to Born’s rule.

However, a reduced density matrix is not the right starting point for the derivation: A pure state, prepared by the observer in the preceding measurement, is. In addition, to get from such a pure state to a mixed (reduced) density matrix one must “trace”—average over, e.g., the environment. Born’s rule is involved in averaging, which leads to a concern that such a derivation may be circular [7,35,48,88].

One could attempt to deal with a pure state of a single system instead. Deutsch (1999) [86] and his followers (Wallace, 2003 [87]; Saunders, 2004 [98]) pursued this approach in terms of decision theory. The key was again invariance under permutations. It is indeed there for certain pure states (e.g., $|0\rangle +|1\rangle$) but it disappears when the relative phase is involved. That is, $|0\rangle +|1\rangle$ equals the post-swap $|1\rangle +|0\rangle$, but $|0\rangle +e^{i\phi }|1\rangle \neq |1\rangle +e^{i\phi }|0\rangle$, and the difference is *not* the overall phase. Indeed, $|0\rangle +i|1\rangle$ is *orthogonal* to $i|0\rangle +|1\rangle$, so there is no invariance under swaps, and the argument that the swap does not matter because the pre-swap state is in the end recovered is simply wrong. In isolated systems this problem cannot be avoided. (Envariance of course deals with it very naturally, as the phase of Schmidt coefficients is envariant—see Equations (3.1)–(3.4).)

The other problem with decision theory approaches put forward to date is selection of events one of which will happen upon measurement—the choice of the preferred states. These two problems must be settled, either through appeal to decoherence (as in Zurek, 1998 [36], and in Wallace, 2003 [87]), or by ignoring phases essentially *ad hoc* (Deutsch, 1999) [86], which then makes readers suspect that some “Copenhagen” assumptions are involved. (Indeed, Barnum et al. (2000) [99] criticized Deutsch (1999) [86] by interpreting his approach in the “Copenhagen spirit”.) In addition, decoherence—invoked by Wallace, 2003 [87]—employs reduced density matrices, hence, Born’s rule. So, as noted by many, it should be “off limits” in its derivation [32,35,48,89,90].

In more recent papers, advocates of the decision theory approach adopt a strategy (Wallace 2007 [100]; 2010 [92]; 2012 [19]) that in effect relies on envariance. This affinity of the updated decision-theoretic approach with envariant derivation of Born’s rule has been noticed and analyzed [91].

Envariant derivation of Born’s rule we have presented is an extension of the swap strategy in (Zurek, 1998) [36]. However, instead of tracing out the environment, we have incorporated it in the discussion (albeit in the role similar to a gauge field).

Envariance leads to Born’s rule, but also to new appreciation of decoherence. Pointer states can be inferred directly from the dynamics of information transfers as was shown in Section 2 (see also Ref. [48]) and, indeed, in the original definition of pointer states [24]. Not everyone is comfortable with envariance (see, e.g., Herbut, 2007 [101], for a selection of views on envariance). This is understandable—interpretation of quantum theory was always rife with controversies.

#### 3.7.1. Implications and the Scope of Envariance: Why Entanglement? Why Schmidt States?

Envariance is firmly rooted in quantum physics. It is based on the symmetries of entanglement. One may be nevertheless concerned about the scope of envariant approach: $p_{k}={\left|\psi_{k}\right|}^{2}$ for Schmidt states, but how about measurements with other outcomes? The obvious starting point for the derivation of probabilities is not an entangled state of S and E, but a pure state of S. In addition, such a state can be expressed in *any* basis that spans $H_{S}$. So, why entanglement? And why Schmidt states?

Envariance of Schmidt coefficient phases is closely tied to the einselection of pointer states: After decoherence has set in, pointer states diagonalize reduced density matrices nearly as well as the Schmidt states (which diagonalize them exactly). Residual misalignment is not going to be a major problem. At most, it might cause minor violations of the laws obeyed by the classical probability for events defined by the pointer states.

Such violations are intriguing, and perhaps even detectable, but unlikely to matter in the macroscopic setting we usually deal with. To see why, we revisit pointer states—Schmidt states (or einselection—envariance) link in the setting of measurements: Observer O uses an (ultimately quantum) apparatus A, initially in a known state $|A_{0}\rangle$, to entangle with S, which then decoheres as A is immersed in E (Zurek, 1991 [27], 2003 [7]; Joos et al., 2003 [29]; Schlosshauer, 2005 [31]; 2007 [32]). This sequence of interactions leads to:$$|\psi_{S}\rangle |A_{0}\rangle |\varepsilon_{0}\rangle \Rightarrow \left(\sum_{k}a_{k}|s_{k}\rangle |A_{k}\rangle \right)|\varepsilon_{0}\rangle \Rightarrow \sum_{k}a_{k}|s_{k}\rangle |A_{k}\rangle |\varepsilon_{k}\rangle \;.$$
In a properly constructed apparatus pointer states $|A_{k}\rangle$ are unperturbed by E while $|\varepsilon_{k}\rangle$ become orthonormal on a decoherence timescale. So in the end we have a Schmidt decomposition of SA (treated as a single entity) and E.

Apparatus is built to measure a specific observable $\sigma_{S}=\sum_{k}\varsigma_{k}|s_{k}\rangle \langle s_{k}|$. Suppose O knows that S starts in $|\psi_{S}\rangle =\sum_{k}a_{k}|s_{k}\rangle$. The choice of A (of Hamiltonians, etc.) commits observer to a definite set of potential outcomes: Probabilities will refer to $\left\{|A_{k}\rangle \right\}$, or, equivalently, to $\left\{|A_{k}s_{k}\rangle \right\}$ in the Schmidt decomposition.

To answer questions we started with (Why entanglement, Why Schmidt states?), entanglement is a result of interactions that cause measurement and decoherence, and only pointer states *of the apparatus* (e.g., states that are near the diagonal, and can play the role of Schmidt states to a very good approximation after decoherence) can be outcomes.

This emphasis on the role of the apparatus in deciding what happens parallels Bohr’s view captured by “No phenomenon is a phenomenon until it is a recorded phenomenon” (Wheeler, 1983) [47]. In our case A is quantum and symptoms of classicality—e.g., einselection as well as the loss of phase coherence between pointer states—arise as a result of entanglement with E.

Envariant approach applies even when $|s_{k}\rangle$ are not orthogonal: Orthogonality of the record states of the apparatus is assured by their distinguishability. This is because (as noted in Section 2) events agents have direct access to are records in A (rather than states of S). Indeed, as we shall discuss in the next section, we access records in the measuring devices indirectly, by intercepting small fragments of, e.g., photon environment that has helped decohere them and has einselected distinct states of the apparatus pointer. States of A that can leave multiple imprints on the environment (so that we can find out measurement outcomes from the tiny fraction of E) must be distinguishable (hence, orthogonal).

Other simplifying assumptions we invoked can be also relaxed [48]. For example, when E is initially mixed (as will be generally the case), one can ‘purify’ it by adding extra $E^{\prime }$ in the usual manner (see Section 2). Given that we already have a derivation of Born’s rule for pure states, the use of the purification strategy (when it is justified by the physical context) does not require apologies, and does not introduce circularity. Indeed, it is tempting to claim that all probabilities in physics *can* be interpreted envariantly.

Probabilities described by Born’s rule quantify ignorance of the observer O *before* he or she finds out the measurement outcome. Therefore, envariant probabilities admit ignorance interpretation—O is ignorant of the *future* outcome (rather than of an unknown *pre-existing* real state, as was the case classically). Of course, once O’s memory becomes correlated with A, its state registers what O has perceived (say, $|o_{7}\rangle$ that registers $|A_{7}\rangle$). Re-checking of the apparatus will confirm it. Moreover, when many systems are prepared in the same initial state, frequencies of the outcomes will accord with Born’s rule.

Envariant approach uses incompatibility between observables of the whole and its parts. It has been now adopted and discussed by others (Barnum, 2003 [93]; Schlosshauer and Fine, 2005 [94]; Schlosshauer, 2005 [31]; 2006 [37]; 2007 [32]; 2019 [33]; Paris, 2005 [102]; Jordan, 2006 [103]; Steane, 2007 [104]; Bub and Pitovsky, 2007 [105]; Horodecki et al., 2009 [106]; Blaylock, 2010 [107]; Seidewitz, 2011 [108]; Hsu, 2012 [109]; Sebens and Carroll, 2018 [110]; Drezet, 2021 [91]).

In retrospect, it seems surprising that envariace was not noticed earlier and used to derive probability or to provide insights into decoherence and einselection: Entangling interactions are key to measurements and decoherence, so entanglement symmetries would seem relevant. However, entanglement was viewed as a paradox, as something that needs to be explained, and not used in an explanation. This attitude is, fortunately, changing.

#### 3.7.2. Towards the Experimental Verification of Envariance

Purifications, use of ancillas, fine-graining, and other steps in the derivation need not be carried out in the laboratory each time probabilities are computed using the state vector: Once established, Born’s rule is a *law*. It follows from the tensor product and the geometry of the Hilbert spaces for composite quantum systems. We used assumptions about C,E, etc., to demonstrate $p_{k}={\left|\psi_{k}\right|}^{2}$, but this rule must be obeyed even when no one is there to immediately verify compliance. So, even when there is no ancilla C at hand, or when E is initially mixed or too small for fine-graining, one could (at some later time, using purification, extra environments and ancillas) verify that bookkeeping implicit in assigning probabilities to $|\psi_{S}\rangle$ or pre-entangled $|\psi_{SE}\rangle$ abides by the symmetries of entanglement.

The obvious next question is how to verify envariance directly. Testing whether the global state is recovered following a swap and a counterswap using tools that favor dealing locally with individual systems is the essence of the experimental difficulties. The few tests of envariance carried out to date approach these challenges differently. The first (and, to date, most precise) test uses pairs of entangled photons to perform the requisite transformations (Vermeyden et al., 2015) [111]. The final global state is acquired by measurements on the individual photons, and characterized using quantum tomography.

Envariance was indeed confirmed with impressive accuracy. Aware of the dangers of circularity Vermeyden et al. used Bhattacharyya coefficient to analyze the experimental data. They have also tested (and constrained) the theory of Son (2014) [112] which allows for powers other than the standard square, $p_{k}={\left|\psi_{k}\right|}^{2}$, in the relation between probability and amplitude. Entangled quantum states were found to be (99.66 ± 0.04)% envariant as measured using the quantum fidelity, and (99.963 ± 0.005)% as measured using a modified Bhattacharyya coefficient. According to the authors, the systematic deviations are due to the less-than-maximal entanglement in their photon pairs.

The experiment of Harris et al. (2016) [113] verified envariance by showing that pure quantum states consisting of two maximally entangled degrees of freedom are left unaltered by the action of successive ’swapping’ operations, each of which is carried out on a different (entangled) degree of freedom. Moreover, it tested and confirmed the perfect correlation used in the derivation of Born’s rule. That is, it showed that Schmidt partners—states that belong to different Hilbert spaces but are linked to one another via tensor product in a Schmidt decomposition—are detected together upon measurement. Bhattacharyya coefficients were again used in testing and the accuracies of well over 99% were attained.

The advent of quantum computers has allowed theorists to act as experimenters. A pioneering example is the test of envariance using 5 qubits of IBM’s Quantum Experience carried out by Deffner (2016) [114]. In addition to testing envariance on entangled pairs, he has also investigated larger GHZ-like states with up to five entangled qubits. The accuracy to which envariance holds decreased with the increasing number of qubits from ∼95% for pairs to ∼75% for quintuples. Final measurement was again done on individual spins. Clearly, quantum computers are at this point no match for serious laboratory experiments, but as they improve, even small quantum devices may be useful as tests of fundamental physics.

While present day quantum computers are imperfect, the progress is, of recent, relatively rapid. It seems therefore likely that much more accurate implementations of strategies that test this fundamental symmetry as well as more advanced tests (needed to verify “finegraining” in the case of unequal absolute values of the coefficients) should be within reach soon. A possible circuit that can be used to verify envariance is illustrated in Figure 3A.

In the meantime, it may be useful to consider other experimental settings and other designs that require laboratory tests but that allow one to verify that (in the wake of a swap and a counterswap) nothing happens that the global state is restored. A possible design of such a test that employs Hong-Ou-Mandel (HOM) interferometry (Hong, Ou, and Mandel, 1987 [115]; Milonni and Eberley, 2010 [116]) is shown in Figure 3B.

## 4. Quantum Darwinism

Objective existence in classical physics was an attribute of the state of a system. An unknown classical state could be measured and found out by many, and yet retain its identity—remain unchanged—even when observers were initially ignorant of what it is.

In contrast to classical states, quantum states are fragile—they cannot be, in general, found out without getting perturbed by the measurement. Objective existence in a quantum world emerges—we shall see—as a consequence of correlations between a system and its environment. It is not (as in classical physics) “sole responsibility” of the system.

Quantum states can, in effect, exist objectively—retain their identity and result in compatible records of independent measurements by many observers—providing observers measure only observables that commute with the preexisting state of the system. In the aftermath of decoherence, this means restriction to its pointer observable. In that case, observers will agree about the outcomes—their measurements will not invalidate one another and will not be erased by decoherence. Therefore, a consensus about the state based on independent measurements—the essence of objectivity—can be established. Such a consensus is the only operational requirement for the “objective existence of classical reality” in our quantum Universe. However, why should observers measure only pointer observables?

Quantum Darwinism provides a simple and natural explanation of this restriction, and, hence, of objective existence—bulwark of classicality—for the einselected states. Quantum Darwinism recognizes that the information we acquire about the Universe comes to us indirectly, through the evidence systems of interest deposited in their environments, and that the only states capable of depositing multiple copies—many quantum memes or qmemes—are the einselected pointer states. Observers access directly only the record made in a fraction of the environment—an imprint of the original state of S on a fragment F of E. There are usually multiple copies of that original (e.g., of this text) that are disseminated throughout E (e.g., by the photon environment—by the light scattered by a printed page or emitted by a computer screen). Observers can find out states of various systems indirectly and agree about their findings because correlations of S with E (which we quantify below using mutual information) allow E to be a *witness* to the pointer state of the system.

In this section we define mutual information, and use it to characterize the information that can be gained about S from E. Objectivity arises because of redundancy—the same information can be obtained independently by many observers from many fragments of E. So, in a sense, objectivity of an observable is quantified by the redundancy of its records—the number of its copies—in E.

The multiplicity of records of the pointer observable of S in E accounts for all the symptoms of the “wavepacket collapse” that are accessible via localized measurements of the environment: Observers who have detected a fragment of E that bears an imprint of the pointer observable of the system will—in the future—encounter only states of the rest of the environment fragments that convey message consistent with the pointer state they have initially seen, and will be able to communicate only with others who have recorded the same pointer state (and are therefore on the same “branch”). Moreover, an observer who decides to verify the state of S (either by measuring it directly, or by intercepting additional fragments of E) will obtain data which confirm that S is in a pointer state that was revealed by the initial measurement on the fragment E that was measured first.

The system itself is untouched—it is not measured directly. Observers acquire their information indirectly, from the qmemes in the environment that has “measured” the system while decohering it. What we find out about our quantum Universe as a consequence of decoherence (that restricts stable states of macroscopic systems to the einselected pointer states) and of quantum Darwinism (selective proliferation of the information about these pointer states). What we see looks classical—it is our familiar classical world: Environment communicates information about pointer states that were selected by decoherence.

We perceive our reality as classical because we are immersed in the information bearing halos of macroscopic systems—because our world consists of extantons, composite entities that combine the source of information (the pointer observable of the macroscopic object that resides in the extanton core) with the means of its delivery (information-bearing halo of, e.g., photons). We only pay attention to the message (the state of the extanton core) and take for granted—ignore—its means of delivery (extanton halo laden with data about the pointer state of the core).

Fragility of individual quantum states is no longer a problem. Observers will generally destroy the evidence (e.g., absorb photons in their retina) while acquiring information. However, there are now many copies of the same information—all imprinted with the data about the underlying state of the system. Therefore, even though evidence of the state of the system may be in part erased, consensus about it will emerge in the end, even as observers measure different fragments of the environment in ways that obliterate carriers of that information.

Last not least, even when observers do not know what are the pointer states of the system, the environment does, and will let them know: Consensus between the evidence carried by different fragments of E emerges as these measurements contain redundant information only about the pointer states[note 10].

Quantum Darwinism can be developed starting from the same assumptions as decohrence theory. Nevertheless, results of the two previous sections are essential when one aims to arrive at a consistent and comprehensive *quantum theory of the classical*. Derivation of the pointer states via repeatability in Section 2 allows us to anticipate preferred states capable of leaving multiple records in E with minimal assumptions that tie directly to the narrative of quantum Darwinism. Only states that are monitored by E without getting perturbed can survive long enough to deposit copious qmemes, their information—theoretic progeny, in the environment: The no-cloning theorem is not an obstacle when “cloning” involves not an unknown quantum state, but an einselected observable. The copies are then messages with the information about the pointer states. The environment becomes an *amplification channel*—a quantum communication channel that carries multiple qmemes of the classical information about the ‘events’ corresponding to the pointer states of S.

The inevitable price of the amplification of the preferred observable is the destruction of the information about the complementary observables and about the initial superposition of the pointer states of S. A single copy of that state is diluted in all of E, so quantum information can be obtained only through global measurements that are inaccessible to local observers. Thus, a quantum environment can transmit only classical information about the pointer observable of S. Or, more precisely, the ability to spawn and disseminate qmemes endows pointer states with all the prerogatives of objective classical existence. Such preferred observable is “fittest” in the Darwinian sense—the original pointer state has survived evolutionary pressures of its environment and has spawned copious qmemes, information - theoretic offspring, advertising its objective existence.

To make these ideas rigorous we shall calculate the number of copies of S in E. To do that we will compute entropies of S, E, and various fragments F of E (see Figure 4) using reduced density matrices and relevant probabilities. It is therefore fortunate that, in Section 3, we have already derived Born’s rule from the symmetries of entanglement. This gives us the right to employ the usual tools of decoherence—trace and reduced density matrices interpreted as statistical entities—to compute entropy.

### 4.1. Mutual Information, Redundancy, and Discord

Quantities that play key role in quantum Darwinism are often expressed in terms of the von Neumann entropy:(4.1)H(ρ)=−Trρlgρ,
The density matrix ρ can describe the state of just one, or of a collection (ensemble) of many quantum systems.

As was done (since at least Laplace) classically, probabilities underlying quantum von Neumann entropy can be regarded as a measure of ignorance[note 11]. However, the density matrix ρ provides more than just its eigenvalues. It is an operator—it has eigenstates. One may be tempted to add that ρ also determines what one is ignorant of: This is generally not the case. Observer can be interested in an observable whose eigenstates do not diagonalize ρ. Indeed, as Section 2 demonstrated, even the einselected pointer states do not always diagonalize $\rho_{S}$. Thus, states that are predictable (because of their stability) and therefore useful may not coincide with instantaneous eigenstates of the reduced density matrix.

What is the ignorance of someone interested in an observable with states $\left\{|\pi_{k}\rangle \right\}$ that differ from the eigenstates of ρ? The corresponding entropy is then the Shannon entropy given by:(4.2)$$H\left(p_{k}\right)=-\sum_{k}p_{k}\mathrm{lg}p_{k}\;,$$
where probabilities of $\left\{|\pi_{k}\rangle \right\}$ are:(4.3*a*)$$p_{k}=Tr\langle \pi_{k}|\rho |\pi_{k}\rangle \;.$$
As noted above, $\left\{|\pi_{k}\rangle \right\}$ may be pointer states (indeed, we shall adopt this notation for the pointer states in this section). They will be (almost) as good in diagonalizing the reduced density matrix of the system as its eigenstates after decoherence. As was noted in Section 3, it is only then that one can associate the usual interpretation of probabilities with pointer states. Otherwise additivity of probabilities may be in danger, as consistent histories approach (Griffiths, 1984 [118]; 2002 [119]; Gell-Mann and Hartle, 1990 [120], 1993 [121]; Omnès 1992 [122]; Griffiths and Omnès, 1999 [123]) makes especially clear.

Note that above—in Equations (4.1) and (4.2)—we have used the same *H* to denote both von Neumann and Shannon entropy. Sometimes different letters (e.g., *S* and *H*) are used for this purpose. We will not do that because, to begin with, S is used to denote the system. Moreover, immediately below we shall consider entropies that are in a sense partly von Neumann and partly Shannon. Last not least, throughout most of this section our focus will be on von Neumann entropy and on the corresponding mutual information.

#### 4.1.1. Mutual Information

Mutual information will help us find out how much a fragment of the environment knows about the system, and what does it know. It is the difference between entropy of two systems treated separately and jointly:(4.4)I(S:A)=H(S)+H(A)−H(S,A).
Mutual information I(S:A) quantifies of how much S and A know about one another.

For classical systems the above definition of I(S:A) is equivalent to the definition of mutual information that employs conditional entropy (e.g., H(S|A)). Conditional entropy quantifies the ignorance about S left after the state of A is found out:(4.5*a*)H(S,A)=H(A)+H(S|A).
A similar formula reverses the roles of the two parties:(4.5*b*)H(S,A)=H(S)+H(A|S).

In classical settings, when states can be characterized by probability distributions, one can simply substitute either of the Equations (4.5) for H(S,A) in Equation (4.4) and obtain an equivalent expression for mutual information. In quantum physics “knowing” is not as innocent—it requires performing a measurement that in general alters the joint density matrix into an outcome—dependent *conditional density matrix* describing the state of the system given the measurement outcome—e.g., given the state $|A_{k}\rangle$ of the apparatus A:(4.6)$$\rho_{S|A_{k}\rangle }=\langle A_{k}|\rho_{SA}|A_{k}\rangle /p_{k}\;.$$
Above, in accord with Equation (4.3a),
(4.3*b*)$$p_{k}=\mathrm{Tr}\langle A_{k}|\rho_{SA}|A_{k}\rangle \;.$$
Given the outcome $|A_{k}\rangle$ the conditional entropy is:(4.7)$$H\left(S|A_{k}\rangle \right)=-\mathrm{Tr}\rho_{S|A_{k}\rangle }\mathrm{lg}\rho_{S|A_{k}\rangle }\;,$$
which leads to the average conditional entropy:(4.8)$$H\left(S|\left\{|A_{k}\rangle \right\}\right)=\sum_{k}p_{k}H\left(S|A_{k}\rangle \right)\;.$$
This much information about S one expects will be still missing after a measurement of an observable with the eigenstates $\left\{|A_{k}\rangle \right\}$ on A. Note that, as in the discussion of probabilities and envariance, this estimate of the expected missing information is relevant for a “bystander”—someone who knows what was measured, but does not yet know the result. (Observer who knows the outcome would use Equation (4.7) instead). An average over all the outcomes, Equation (4.8), gives the expected remaining ignorance about S as long as one does not know the outcome.

Once the observer perceives the outcome, the relevant state of S (and the corresponding relevant entropy) will be given by Equations (4.6) and (4.7). This is the infamous “collapse”—the range of possibilities is reduced to a single actuality. We note that even for a bystander—observer’s friend who knows that the measurement has already happened, but who has not yet found out the outcome[note 12] the joint state of SA is usually affected, as the reconstituted density matrix $\sum_{k}p_{k}|A_{k}\rangle \;\langle A_{k}|\rho_{S|A_{k}\rangle }$ differs in general from the pre-measurement $\rho_{SA}$. In particular, the entropy of the reconstituted mixed state is usually larger than the entropy of the pre-measurement $\rho_{SA}$.

This increase of entropy is characteristically quantum. It was pointed out already by von Neumann (1932) [4]. Decoherence explains it as an inevitable consequence of the correlations with the environment E that “monitors” A. From the point of view of the bystander, correlations of A with the environment or with the observer have a similar effect—they can invalidate some of the information the bystander had about SA, and hence increase entropy.

The entropy in Equation (4.8) can be viewed as a half von Neumann—half Shannon: It involves (quantum) conditional density matrices as well as (effectively classical) probabilities of outcomes. Given $\rho_{SA}$, one can also address a more specific question, e.g., how much information about a specific observable of S (characterized by its eigenstates $\left\{|s_{j}\rangle \right\}$) will be still missing after a specific observable with the eigenstates $\left\{|A_{k}\rangle \right\}$ of A is measured. This can be answered by using $\rho_{SA}$ to compute the joint probability distribution:(4.9)$$p\left(s_{j},A_{k}\right)=\langle s_{j},A_{k}|\rho_{SA}|s_{j},A_{k}\rangle \;.$$
These joint probabilities are in effect classical. They can be used to calculate Shannon joint entropy for any two observables (one in S, the other in A), as well as entropy of each of these observables separately, and to obtain the corresponding (Shannon) mutual information:(4.10)$$I\left(\left\{|s_{j}\rangle \right\}:\left\{|A_{k}\rangle \right\}\right)=H\left(\left\{|s_{j}\rangle \right\}\right)+H\left(\left\{|A_{k}\rangle \right\}\right)-H\left(\left\{|s_{j}\rangle \right\},\left\{|A_{k}\rangle \right\}\right)\;.$$

We shall find uses for all of these variants of mutual information. The von Neumann entropy based I(S:A), Equation (4.4), answers the question “how much the systems know about each other”, while the Shannon version immediately above quantifies mutual information between two specific observables. Shannon version is (by definition) basis dependent. It is straightforward to see that, for the same underlying joint density matrix:(4.11)$$I\left(S:A\right)\geq I\left(\left\{|s_{j}\rangle \right\}:\left\{|A_{k}\rangle \right\}\right)\;.$$
Equality is attained only for a special choice of the two measured observables, and only when the eigenstates of $\rho_{SA}$ are direct products $|s_{k}\rangle |A_{k}\rangle$ of the orthogonal sets of states $\left\{|s_{k}\rangle \right\}$ and $\left\{|A_{k}\rangle \right\}$ of S and A. In that case correlations between S and A can be regarded as completely classical.

With the help of Equation (4.8) one can define “half way” (Shannon—von Neumann) mutual informations that presume a specific measurement on one of the two systems (e.g., A), but make no such commitment for the other one. For instance,
(4.12*a*)$$J\left(S:\left\{|A_{k}\rangle \right\}\right)=H\left(S\right)-H\left(S|\left\{|A_{k}\rangle \right\}\right)\;$$
would be one way to express mutual information defined “asymmetrically” in this way. A corresponding formula;
(4.12*b*)$$J\left(A:\left\{|s_{k}\rangle \right\}\right)=H\left(A\right)-H\left(A|\left\{|s_{k}\rangle \right\}\right)\;$$
is relevant when S is measured in the basis $\left\{|s_{k}\rangle \right\}$.

#### 4.1.2. Quantum Discord

Quantum discord is the difference between the mutual information defined using symmetric von Neumann formula, Equation (4.4), and one of the half-way Shannon—von Neumann versions [124,125,126]. For example:(4.13*a*)$$D\left(S;\left\{|A_{k}\rangle \right\}\right)=I\left(S:A\right)-J\left(S:\left\{|A_{k}\rangle \right\}\right)\;.$$
Discord is a measure of how much information about the two systems is inaccessible locally—how much of the globally accessible mutual information is lost when one attempts to find out the state of SA starting with a local measurement on A with outcomes $\left\{|A_{k}\rangle \right\}$.

Discord is asymmetric and basis-dependent, as information gain about S depends on what gets measured on A. The least discord (corresponding to optimal $\left\{|A_{k}\rangle \right\}$):(4.13*b*)$$D\left(S;A\right)=\min_{\left\{|A_{k}\rangle \right\}}\left\{D\left(S;\left\{|A_{k}\rangle \right\}\right)\right\}=0\;\;$$
disappears iff $\rho_{SA}$ commutes with $A=\sum_{k}\alpha_{k}|A_{k}\rangle \;\langle A_{k}|$:(4.14*a*)$$\left[\rho_{SA},A\right]=0.\;\;$$
When that happens, quantum correlation is classically accessible from A. This can be assured iff;
(4.14*b*)$$\left[\rho_{SA},\rho_{A}\right]=0.\;$$
Decoherence of A that einselects preferred pointer basis $\left\{|A_{k}\rangle \right\}$ will evolve $\rho_{SA}$ to where Equations (4.14) are satisfied to a good approximation [126].

When the composite system is classical, so that its state can be found out without disturbing it and can be—prior to measurements—characterized by a probability distribution that is independent of the measuring process, the symmetric *I* (defined using joint entropy, Equation (4.4)) and the asymmetric *J* (defined using conditional entropy, Equation (4.12)) coincide. The proof (see, e.g., Cover and Thomas, 1991) [127] relies on Bayes’ rule relating conditional and joint probabilities. Thus, non-vanishing discord signifies breakdown of Bayes’ rule in quantum physics.

We emphasize that the asymmetric “half way” (Shannon—von Neumann) mutual information *J* is indeed asymmetric—it depends on whether measurements are carried out on A or S. In the classical case asymmetric-looking definition of *J* results, courtesy of Bayes’ rule, in a symmetric mutual information, as J(S:A)=I(S:A)=J(A:S). In the classical case J(S:A)=J(A:S), so this does not matter, but in the quantum case, in general, J(S:A)≠J(A:S).)

As a consequence of asymmetry between the system that is measured and its partner whose state is inferred indirectly, based on the outcome of that measurement, it is possible to have correlations that are classically accessible only “from one end” [128]. For instance:$$\rho_{SA}=\frac{1}{2}\left(|↑\rangle \;\langle ↑|\right|A_{↑}\rangle \;\langle A_{↑}\left|+\right|↗\rangle \;\langle ↗\left|\right|A_{↗}\rangle \;\langle A_{↗}|)$$
will be classically accessible through a measurement on A with orthogonal record states $\left\{|A_{↑}\rangle ,|A_{↗}\rangle \right\}$ (i.e., when $\langle A_{↑}|A_{↗}\rangle =0$), but classically inaccessible to any measurement on S when 〈↑|↗〉≠0. Indeed, the original motivation for introducing discord was the observation that decoherence of the apparatus makes the correlations accessible from A [124].

Minimization used in Equation (4.13b) raises the obvious question: Could one do better if one used positive operator valued measures (POVM’s) rather than Hermitian observables with orthogonal eigenstates? The answer is, unsurprisingly, “Yes”. In the case of POVM’s $\left\{\pi_{k}\right\}$ the asymmetric mutual information coincides with the familiar Holevo quantity χ (Holevo, 1973 [129]):(4.12*c*)$$\chi \left(\rho_{A}\right)=\max_{\left\{\pi_{k}\right\}}(H\left(\rho_{A}\right)-\sum_{k}p_{k}H\left(A|\pi_{k}\right)),\;$$
so that the minimum discord can be now expressed as:(4.13*c*)$$D\left(S;A\right)=I\left(S:A\right)-\chi \left(\rho_{A}\right).\;$$
Holevo quantity bounds the capacity of quantum channels to carry classical information. This role of Holevo χ fits naturally into the discussion of quantum Darwinism where fragments F of the environment play a role of quantum channels transmitting information about S that is being decohered by E. In particular, Zwolak and Zurek (2013) [130] point out that χ and D—classical and quantum information transmitted by this channel—are complementary, while Touil et al., (2022) [131] show that the Holevo bound is a reasonable estimate of the information about S that can be accessed by optimal measurement of F.

Quantum discord has become an active area of research following indications that the “quantumness” it defines may play a fundamental role in operation of quantum thermodynamic demons ([128], also Brodutch and Terno, 2010 [132]), in defining completely positive maps (Rodrígues-Rosario et al., 2008) [133], and, especially, in quantum information processing (Datta, Shaji, and Caves, 2008) [134] as well as quantum communication (Piani, Horodecki and Horodecki, 2008 [135]; Luo and Sun, 2010 [136]; Gu et al., 2012 [137]; Dakic et al., 2012 [138]). Our brief introduction to discord is clearly incomplete (see, however, Modi et al., 2012 [139] and Bera et al., 2018 [140] for reviews), but it will suffice for our purpose.

Questions we shall analyze using mutual information and discord will concern (i) redundancy of information (e.g., how many copies of the record does the environment have about S), and; (ii) what is this information about (that is, what observables of the system are recorded in the environment with large redundancy).

Objectivity appears as a consequence of large redundancy. In the limit of large redundancy the precise value of redundancy has as little physical significance as the precise number of atoms on thermodynamic properties of a large system. So, it will be often enough to show that redundancy is large (rather than to calculate exactly what it is). Discord will turn out to be a measure of the unattainable quantum information that cannot be extracted by local measurements from the fragments of the environment.

One might be concerned that having different measures—different mutual informations—could be a problem, as this could lead to contradictory answers, but in practice this never becomes a serious issue for two related reasons: There is usually a well-defined pointer observable that obviously minimizes discord, so various possible definitions of mutual information tend to agree where it matters. Moreover, the effect we are investigating—quantum Darwinism—is not subtle: We shall see thet there are usually many copies of pointer states of S in E, so (as is discussed by Touil et al. (2022) [131] and Zwolak (2022) [141]) the discrepancy between redundancies computed using different methodologies—differences between the numbers of copies defined through different measures—is irrelevant.

#### 4.1.3. Evidence and Its Redundancy

We study a quantum system S interacting with a composite quantum environment $E=E_{1}⊗E_{2}⊗\cdots ⊗E_{N}$. The question we consider concerns the information one can obtain about S from a fragment F of the environment E consisting of several of its subsystems (see Figure 4). To be more specific, we partition E into non-overlapping fragments. Redundancy of the record is then defined as the number of disjoint fragments each of which can supply sufficiently complete information (i.e., all but the *information deficit* δ<1) about S;
(4.15)$$I\left(S:F_{\delta }\right)\geq \left(1-\delta \right)H_{S}\;\;.$$
Small information deficit, δ≪1, implies that nearly all the classically available information can be obtained from $F_{\delta }$. This will not always be the case, and δ≪1 is not a condition for the effectively classical behavior or even for an agreement between different observers[note 13].

We now define redundancy as the number of fragments that can independently supply almost all—all but δ—of the missing information $H_{S}$ about the system:(4.16)$$R_{\delta }={\frac{1}{f}}_{\delta }\;.\;$$
Definition of redundancy can be illustrated using partial information plots that show the dependence of the mutual information on the size of the intercepted fraction of the environment: Redundancy is the length of the plateau of I(S:F) measured in the units set by the support of the initial, rising, portion of the graph—the part starting at I(S:F)=0 and ending when $I\left(S:F_{\delta }\right)=\left(1-\delta \right)H_{S}$ (see Figure 5). Thus, to estimate redundancy we will need to determine how much information about S can one get from a typical fragment F that contains a fraction;
$$f_{\delta }=\frac{{}^{♯}\;F_{\delta }}{{}^{♯}\;E}=\frac{number\;of\;subsystems\;in\;F_{\delta }}{number\;of\;subsystems\;in\;E}$$
of E. For this we need the dependence of the mutual information I(S,F) on $f={}^{♯}\;F/{}^{♯}\;E$.

Examples of partial information plots (or “PIPs”) of the von Neumann mutual information for a pure composite system consisting of S and E are shown in Figure 5. The first observation is that these plots are asymmetric around $f=\frac{1}{2}$. This can be demonstrated[note 14] using elementary properties of the von Neumann mutual information and assuming; (i) F is typical, and; (ii) the whole of SE pure [143].

There is a striking difference between the character of PIPs for random pure states in the whole joint Hilbert space $H_{SE}=H_{S}⊗H_{E}$ and states resulting from decoherence—like evolution: For a random state very little information obtains from fragments with $f<\frac{1}{2}$. By contrast, for PIPs that result from decoherence already a small fragment F of E (f≪1) will often supply nearly all the information that can be obtained from less than almost all SE.

The character of such decoherence—generated PIPs suggest dividing information into (i) easily accessible *classical information* $H_{S}$ that can be inferred (up to the information deficit δ) from local measurements—from any sufficiently large fragment $F_{\delta }$ that is still small compared to half of E, and (ii) *quantum information* that is locally inaccessible, but is present, at least in principle, in the global observables of the whole SE.

This shape of PIPs is a result of einselection: When there is a preferred observable in S that is monitored but not perturbed by the environment, the information about it is recorded over and over again by different subsystems of E. A combined state of S and E resulting from decoherence will have a *branching* structure;
(4.17)$$|\Psi_{SE}\rangle =\sum_{k}e^{i\phi_{k}}\sqrt{p_{k}}|\pi_{k}\rangle |\varepsilon_{k}^{\left(1\right)}\rangle |\varepsilon_{k}^{\left(2\right)}\rangle \cdots |\varepsilon_{k}^{\left(l\right)}\rangle \cdots \;$$
with the pointer states of the system $|\pi_{k}\rangle$ at the base of each branch. Subsystems of E are correlated with these pointer states, but, individually, will often contain only poor (far from distinguishable) records of $|\pi_{k}\rangle$. Nevertheless, even when records contained in individual subsystems are insufficient—$\langle \varepsilon_{j}^{\left(l\right)}|\varepsilon_{k}^{\left(l\right)}\rangle$ is nowhere near $\delta_{jk}$—sufficiently long fragments F of branches labelled by distinct pointer states $|\pi_{k}\rangle$ will be approximately orthogonal. As a result, nearly all of the easily accessible classical information can be often recovered from small fragments—a fraction of the environment.

#### 4.1.4. Mutual Information, Pure Decoherence, and Branching States

In quantum Darwinism, fragment F plays a role of an apparatus or of a communication channel designed to access the same pointer observable that can survive intact in spite of the immersion of S in E. Decoherence singles out preferred observables of S. They are determined by the dynamics of decoherence, so—in presence of fixed interaction Hamiltonians—they remain unchanged even as more and more copies of the information about the system are deposited in E. This is fortunate, as calculating mutual information is in general difficult. However, when decoherence is the only significant process—when we are dealing with *pure decoherence* that results in perfect branching states—calculations simplify [131,141,144,145,146]).

Pure decoherence is defined by the system-environment Hamiltonian that commutes with $\left\{|\pi_{k}\rangle \right\}$, pointer states of S, and does not directly correlate subsystems $E_{l}$ of the environment:(4.18*a*)$$H_{SE}=\sum_{k}\sum_{l}\varsigma_{kl}|\pi_{k}\rangle \;\langle \pi_{k}|{\hat{\gamma }}_{E_{l}}.$$
Above, Hermitian operators ${\hat{\gamma }}_{E_{l}}$ act on individual subsystems of E. The resulting evolution operator $U_{SE}\left(t\right)$ factors:(4.18*b*)$$U_{SE}\left(t\right)=e^{-iH_{SE}t/\hbar }=U_{SF}\left(t\right)U_{SE_{/F}}\left(t\right).$$
This independence of the evolution of a fragment of the environment F from its remainder $E_{/F}$ will greatly simplify our calculations. Addition of a self-Hamiltonian $H_{S}$ that commutes with $H_{SE}$ would not affect our discussion. Similarly, one could add self-Hamiltonians of the environment subsystems. As long as $U_{SE}\left(t\right)$ factors as above, our conclusions will be valid.

We now consider evolution of $\rho_{SF}$ starting from an initially uncorrelated state;
$$\rho_{SE}\left(0\right)=\rho_{S}\left(0\right)\rho_{E1}\left(0\right)\rho_{E2}\left(0\right)...$$
Given our assumptions that include pure decoherence, the evolved state is given by;
(4.19)$$\rho_{SF}\left(t\right)=U_{SF}\left(t\right)\rho_{SdE_{/F}}\left(t\right)\rho_{F}\left(0\right)U_{SF}{\left(t\right)}^{†}$$
where;
$$\rho_{SdE_{/F}}\left(t\right)={\mathrm{Tr}}_{E_{/F}}U_{SE_{/F}}\left(t\right)\rho_{S}\left(0\right)\rho_{E_{/F}}\left(0\right)U_{SE_{/F}}{\left(t\right)}^{†}$$
represents the state of S decohered by $E_{/F}$, the remainder of the environment. The structure of Equation (4.19) implies that the joint entropy of the system S and the environment fragment F is given by:$$H_{SF}\left(t\right)=H_{SdE_{/F}}\left(t\right)+H_{F}\left(0\right).$$
This identity is valid for branching states, Equation (4.17), resulting from pure decoherence. It allows us to write (Zurek, 2007 [49]; Zwolak, Quan, and Zurek, 2009 [144]; 2010 [145]):(4.20)$$I\left(S:F\right)=\overset{local/classical}{\left(H_{F}-H_{F}\left(0\right)\right)}+\overset{global/quantum}{\left(H_{S}-H_{SdE_{/F}}\right)}.\;$$
The mutual information is given by the sum of two contributions, which (as indicated above) can often be regarded as, respectively, classical and quantum.

The first contribution, $H_{F}-H_{F}\left(0\right)$, is the increase of the entropy of the fragment F. In our case of pure decoherence it is all due to the correlation with S—due to the information F acquires about S. This information about S can be accessed indirectly, by measuring F, and is available locally (hence, it is within reach of local observers). In the PIP representing decoherence in Figure 5 increase of $H_{F}$ is responsible for the rapid rise of I(S:F) that starts at f=0 and for its leveling off at the classical plateau at $H_{S}$ that can happen already at f≪1. This information about S is easily accessible because it has been widely disseminated—many independent fragments of E share it.

The second term, $H_{S}-H_{SdE_{/F}}$, turns out (Zwolak, Quan, and Zurek, 2010) [145] to be the discord in the pointer basis of S. It is usually negligible when f<1, but can become significant when f→1 (F→E). Consequently, it represents information that can be obtained only via global measurements—measurements that involve nearly all SE. When decoherence by the environment E is solely responsible for $H_{S}$, one can rewrite $H_{S}-H_{SdE_{/F}}$ as $H_{SdE}-H_{SdE_{/F}}$—as the difference between the decoherence caused by all of E and its remainder, $E_{/F}$. As long as the remainder $E_{/F}$ keeps S decohered, $H_{S}-H_{SdE_{/F}}=0$. Only when, with the increase of *f*, $E_{/F}$ becomes too small to effectively decohere S, $H_{SdE_{/F}}$ begins to be substantially less that $H_{S}$, and eventually disappears. In that limit of a vanishing remainder $H_{S}-H_{SdE_{/F}}\to H_{S}$, and I(S:F) approaches $2H_{S}$ for f→1.

To sum up, the initial climb of I(S:F) to the classical plateau at $H_{S}$ is due to $H_{F}-H_{F}\left(0\right)$. The final climb from that classical plateau to the quantum peak $I\left(S:F\right)=2H_{S}$ happens when the remainder of the environment $E_{/F}$ is not enough to keep S decohered, so that $H_{SdE_{/F}}$ falls below $H_{S}$. This quantum value of $I\left(S:F\right)=2H_{S}$ can be reached at f=1, and only when SE, as a whole, is pure (so that $H_{SE}=0$, and $H_{S}=H_{E}$). This additional quantum information can be revealed only by measurements that have global entangled states of SE as outcomes.

In addition to the need for global measurements there is another reason that suggests that $H_{S}-H_{SdE_{/F}}$ represents quantum information. It is best illustrated by contrasting the behavior of $H_{S}-H_{SdE_{/F}}$ for f→1 when the system starts in a mixture of pointer states (so that its entropy is $H_{S}$ already before it couples to E, and cannot increase any more due to decoherence by E) with the alternative—when S starts in a corresponding pure “Schrödinger cat” state (so its entropy is due to decoherence by E, $H_{S}=H_{SdE}$).

In the case of initially pure E and mixed S the entropy $H_{SdE_{/F}}$ remains unchanged—the system was pre-decohered—and I(S:F) levels off at the classical plateau even as f→1—it can never exceed $H_{S}$. However, when S is initially pure, $H_{SdE_{/F}}$ disappears as F→E and $E_{/F}$ becomes too small to be an effective decoherer. In that case the additional information that can be in principle recovered from SE concerns phases between the pointer states (or, to put it differently, observables complementary to the pointer observable).

In the “opposite” case—when E is initially mixed but S is initially pure—the first term in Equation (4.20) vanishes, and $H_{F}\left(0\right)$ is already as large as $H_{F}$ can get, so that classical plateau disappears—the mutual information remains negligible until *f* is nearly 1. However, even now $H_{S}-H_{SdE_{/F}}$ eventually attains $H_{S}=H_{SdE}$, as $H_{SdE_{/F}}$ vanishes when $E_{/F}$ shrinks with F→E. So, at the end of the PIP there is still a quantum peak, but now it rises not above the classical plateau at $I\left(S:F\right)=H_{S}$, but, rather from the “sea level”, I(S:F)=0, so that at f=1 mutual information reaches the peak value of only $H_{S}$.

#### 4.1.5. Surplus Decoherence and Redundant Decoherence

In realistic situations, observers can intercept only a fraction of the environment. Thus, f<1 (indeed, usually f≪1). It is therefore often natural to assume that the remainder of the environment suffices to keep S decohered. This implies that $\rho_{SF}$ has eigenstates that are products of pointer states of S with some states of F (and not, e.g., entangled states of SF; indeed, in this case $\rho_{SF}$ has vanishing discord—it is classically accessible from S). This will be true providing that there is *surplus decoherence*, so that one does not need all of E to keep S decohered—the remainder $E_{/F}$ of the environment is enough.

The essence of surplus decoherence is easily traced (and closely tied) to the branching structure of the states of SE we have already recognized as a consequence of branching states, Equation (4.17), and of pure decoherence, Equation (4.18). Surplus decoherence implies that states of $E_{/F}$, the remainder of E, are nearly orthogonal (so they can constitute nearly perfect records of pointer states of S). This guarantees that the same states that are selected by decoherence and diagonalize decohered $\rho_{S}$ also help diagonalize $\rho_{SF}$ (i.e., $[\rho_{S},\rho_{SF}]=0$, Equation (4.14)—the joint state $\rho_{SF}$ is classically accessible from S, and its discord disappears in the pointer basis). This assumption breaks down when the states of the remainder $E_{/F}$ correlated with the pointer states are no longer orthogonal, so that the eigenstates of $\rho_{SF}$ are entangled or discordant states of SF, and $[\rho_{S},\rho_{SF}]\neq 0$. For an initially pure SE this will always eventually happen as f→1, providing that S started in a superposition of pointer states.

Quantum Darwinism recognizes that situations when there are many copies of S inscribed in E are commonplace in our Universe. A single accurate copy in the environment suffices to decohere S. Thus, when there are many copies, one can expect not just “surplus decoherence” but a situation when S is decohered “many times over”. This situation (which often turns out to be generic in our Universe) defines *redundant decoherence*.

Redundant decoherence may sound like an oxymoron—once coherence is lost from S, one might say, there is no way to decohere S even more. Indeed, redundant decoherence will have no additional effect on S; $\rho_{S}$ will remain diagonal in the pointer basis, and $H_{S}$ will no longer increase. However, it will turn out to be useful (e.g., in discussions of irreversibility) to appeal to the redundancy of decoherence $R_{\delta_{D}}$. Redundancy of decoherence is defined—by analogy with the redundancy of information about S in E, Equations (4.15) and (4.16)—by enquiring what typical fraction $f_{\delta_{D}}$ of the environment suffices to decohere the system to the extent given by the *decoherence deficit*$\delta_{D}$:(4.21)$$H_{SdF_{\delta_{D}}}=\left(1-\delta_{D}\right)H_{S}=\left(1-\delta_{D}\right)H_{SdE}.$$
A very small fragment of E, $f_{\delta_{D}}\ll 1$, will often suffice. The redundancy of decoherence is then defined by:(4.22)$$R_{\delta_{D}}=\frac{1}{\delta_{D}},$$
and is at least as large (and can be much larger) than the previously defined redundancy of information about S:(4.23)$$R_{\delta_{D}}\geq R_{\delta }.$$
The equality, $R_{\delta_{D}}=R_{\delta }$, can be attained only when the environment is initially pure and its subsystems do not interact (so that all of the entropy in its fragments is due to the information it acquires about S). Mixed environment with interacting subsystems can still decohere S very effectively, but it is generally more difficult to retrieve information about S, so in this case one can have $R_{\delta_{D}}>R_{\delta }$ (or even $R_{\delta_{D}}\gg R_{\delta }$).

#### 4.1.6. Information Gained by Pure and Mixed Environments

Further simplifications of Equation (4.20) are often possible. Thus, when the environment is initially pure, we get:$$I\left(S:F\right)=H_{F}+\left(H_{S}-H_{SdE_{/F}}\right).$$
Moreover, when $H_{S}=H_{SdE_{/F}}$ (due to surplus decoherence):(4.24*a*)$$I\left(S:F\right)=H_{F}\;$$
follows. This simple expression for mutual information is valid for pure decoherence in an initially pure environment for *f* starting at 0 until F gets to be so large that the decoherence by the (shrinking) remainder $E_{/F}$ of the environment is no longer effective and $H_{S}\neq H_{SdE_{/F}}$.

The opposite case of a completely mixed E yields:(4.25)$$I\left(S:F\right)=\left(H_{S}-H_{SdE_{/F}}\right).$$
When S is initially pre-decohered in the pointer basis this implies I(S:F)=0. However—and this may seem surprising—for an initially pure S mutual information will still rise as f→1, but now (with completely mixed E)—it will reach only $H_{S}$ (and not $2H_{S}$ as was the case for pure SE). This means that quantum phase information is still “out there”, and, at least in principle could be recovered (see Ref. [147]), in spite of the completely mixed E.

One might be surprised that completely mixed environments can be effective decoherers. After all, decoherence is caused by the environment “finding out” about the system, and in a completely mixed environment there does not seem to be any place left to accommodate the data. The right way of thinking about this relies on the “Church of Larger Hilbert Space” view of the mixtures, and is very much in tune with the envariant derivation of probabilities and Born’s rule in the preceding section. Mixed environment can be regarded as one half of an entangled pair (so that probabilities are due to the symmetries of entangled state involving E and its purifying “doppelganger” $E^{\prime }$ we have explored using Schmidt decomposition in the envariant derivation of Born’s rule). That entangled pair will acquire information about S. Thus, even when the environment E is completely mixed, it will decohere the system as if it was entangled with and “purified” by $E^{\prime }$.

The last case (that will be relevant for some of the examples we are about to consider) assumes initially pure S and E, as well as redundant (or at least surplus) decoherence. In that case:(4.24*b*)$$I\left(S:F\right)=H_{F}=H_{SdF}.$$
This last equality follows from the Schmidt decomposition of SE with F and $SE_{/F}$ as subsystems. Under pure decoherence the reduced density matrix $\rho_{F}$ does not depend on whether the system is coupled to the remainder of the environment, $E_{/F}$. Thus, $\rho_{F}$ would be the same if it evolved only in contact with isolated S. However, in that case the system would be decohered only by F, and $H_{F}=H_{S}$ (by Schmidt decomposition), which establishes $H_{F}=H_{SdF}$.

This last equality allows one to use standard tools of decoherence to compute $H_{F}$. Equality $I\left(S:F\right)=H_{SdF}$ follows from Equation (4.20) under the assumption of surplus decoherence, $H_{S}-H_{SdE_{/F}}\approx 0$.

#### 4.1.7. Environment as a Communication Channel

We conclude this section by noting that (with the few simplifications that were already justified) the way in which the information about S is transmitted by the fragment F of E is analogous to the transfer of the (classical) information about the pointer states through a quantum channel. The joint state of SF has the form:(4.26)$$\rho_{SF}=Tr_{E_{/F}}\rho_{SE}≃\sum_{k}p_{k}|\pi_{k}\rangle \;\langle \pi_{k}|\rho_{F_{k}}\;.$$
Effectively classical pointer states $|\pi_{k}\rangle \;\langle \pi_{k}|$ are encoded in the quantum states $\rho_{F_{k}}$ of the channel F. The theorem (due to Holevo, 1973 [129] and Schumacher, 1995 [148]) shows that the capacity of a quantum channel to carry classical information is bounded from above by:(4.27*a*)$$\chi \left(F:\left\{\pi_{k}\right\}\right)=H_{F}-\sum_{k}p_{k}H\left(F|\pi_{k}\right)\;.$$
When $\pi_{k}=|\pi_{k}\rangle \;\langle \pi_{k}|$ are orthogonal projectors, χ coincides with the asymmetric mutual information $J(F|\left\{|\pi_{k}\rangle \right\})$, Equation (4.12). Generalization to where $\pi_{k}$ are POVM’s is possible. Quantum discord can be then bounded from below by:(4.27*b*)$$D\left(F:\left\{\pi_{k}\right\}\right)=I\left(F:S\right)-\chi \left(F:\left\{\pi_{k}\right\}\right)\;.$$
Discord is the mutual information that cannot be communicated classically.

One can rewrite the definition of quantum discord as:(4.27*c*)$$I\left(F:S\right)=J\left(F:\left\{\pi_{k}\right\}\right)+D\left(F:\left\{\pi_{k}\right\}\right)\;.$$
This conservation law (Zwolak and Zurek, 2013) [130] provides a new view of complementarity: The left hand side is fixed and basis—independent, while both terms on the right hand side depend on $\left\{\pi_{k}\right\}$. The first term represents information about the observable ς that can be gained by the measurements on F. This information is maximized for the pointer observable Π. Information inscribed in F about any other observable ς will be less. One can show [130] that the information about ς that can be obtained from F is:(4.28)χ(ς:F)=H(ς)−H(ς|Π).
Above, H(ς|Π) is the conditional entropy, the information about ς still missing when Π is known. For instance, observables complementary to Π cannot be found out from F.

We can now understand why only the pointer observable (and, possibly, observables closely aligned with it) can be found out by intercepting a fraction of the environment. Redundant imprinting of an observable ς is possible only when there is a fragment F that is large enough so that the information about ς can be extracted from it:(4.29*a*)χ(ς:F)≥(1−δ)H(ς).
Using the earlier expression for χ(ς:F) one obtains an inequality H(ς|Π)≤δH(ς). In other words, redundant information about observables that are so poorly correlated with the pointer observable that the conditional entropy satisfies the inequality:(4.29*b*)H(ς|Π)>δH(ς),
cannot be obtained from the environment fragments [48,130,149]. We shall return to this subject below.

#### 4.1.8. Quantum Darwinism and Amplification Channels

The usual focus of the communication theory is to optimize the channel capacity. Thus, in quantum communication theory one would consider messages (such as our $\pi_{k}$ above) sent with the probabilities $p_{k}$, one at a time. Capacity is defined in the limit of many uses of the channel. That is, in this setting one would be dealing with an ensemble described by a density matrix of the form:(4.30*a*)$$\varrho_{SF}^{N}={\left(\sum_{k}p_{k}|\pi_{k}\rangle \;\langle \pi_{k}|\rho_{F_{k}}\right)}^{⊗N}.$$
The theorems are established in the limit of N→∞.

By contrast, in quantum Darwinism we are dealing with a state that can be approximately expressed as:(4.30*b*)$$\rho_{SF}≃\sum_{k}p_{k}|\pi_{k}\rangle \;\langle \pi_{k}|\overset{R_{\delta }}{\underset{l}{⨂}}\rho_{F_{k}}^{\left(l\right)}\;.\;\left(4.30b\right)$$
That is, the same message $|\pi_{k}\rangle \;\langle \pi_{k}|$ is inscribed over and over, $∼R_{\delta }$ times, in the environment as a whole. This is how its multiple copies can reach many observers.

This is also why an observer who consults one fragment of E and infers the state of the system from it will get data consistent with the first finding from the consecutive fragments. In view of this structure of the states of the environment the nomenclature “amplification channel” is well justified: Shared quantum information becomes effectively classical, since quantum discord cannot be shared (Streltsov and Zurek, 2013) [150]. Quantum Darwinism implements amplification that provides means of such sharing, shedding new light on the ubiquity of amplification in the transition from quantum to classical.

We shall now illustrate these insights in models of quantum Darwinism. We shall also investigate situations where some or even all of the simplifying assumptions employed above break down. We also note that the above discussion focused on the mutual information defined via von Neumann entropy, and thus, that it prepared us to answer the question about the amount of information that was deposited in, and can be extracted from F. This largely bypasses the question: What is this information about? One can anticipate that the answer is “pointer states”. We have already produced evidence of this. We shall confirm and quantify it (by enquiring how much information one can extract from F about other observables) while discussing quantum Darwinism in specific models.

### 4.2. Quantum Darwinism in Action

Dissemination of information throughout the environment has not been analyzed until recently. Given the complexity of this process, it is no surprise that the number of results to date is still rather modest, but they have already led to new insights into the nature of the quantum-to-classical transition. The models discussed here show that; (i) *dynamics responsible for decoherence is capable of imprinting multiple copies of the information about the system in the environment*. Whether that environment can serve as a useful witness depends on the memory space it has available to store this information, and whether the information is stored unscrambled and unperturbed and is accessible to observers.

Quantum Darwinism will always lead to decoherence, but the reverse is not true: There are situations where the environment cannot store any information about S (e.g., because E is completely mixed to begin with) or where the information it stores becomes scrambled by the dynamics (so that it is effectively inaccessible to local observers).

So, redundancy of records that is so central to quantum Darwinism is not necessarily implied by decoherence. Moreover; (ii) *redundancy can keep on increasing long after decoherence has completely decohered the system*: Copies of the einselected pointer observable can continue to be added—imprinted on E. As we have already noted, redundancy of decoherence is at least as large as $R_{\delta }$, and can be much larger: It is possible to have hugely redundant decoherence while $R_{\delta }$ remains negligible. However, typically, both redundancies will continue to increase as the system and the environment continue to interact.

Last not least; (iii) *only the einselected pointer states can be redundantly recorded in E.* While multiple copies of the information about the preferred pointer observable are disseminated throughout E, only one copy of the complementary information is (at best) shared by all the subsystems of the environment, making it effectively inaccessible.

Using imperfect analogies with classical devices, one can say that the information flow from S to E acts as an amplifier for the pointer observable, and, simultaneously, as a shredder for the complementary observables, dispersing slivers of a single copy of the phase information in the correlations with many subsystems of the environment. All of these fragments would have to be brought together and coherently reassembled to recover preexisting state of S. By contrast, many copies of the information about the pointer observable are readily available from the fragments of E.

In addition to these general characteristics of quantum Darwinism we shall see that realistic models—e.g., photon scattering—can lead to huge redundancies, and that environment that is partially mixed can still serve as an effective communication channel, allowing many observers independent access to the information about the preferred observable of the system.

#### 4.2.1. C-Nots and Qubits

The simplest model of quantum Darwinism is a rather contrived arrangement of many (*N*) target qubits that constitute subsystems of the environment interacting via a *controlled not* (“c-not”) with a single control qubit S. As time goes on, consecutive target qubits become imprinted with the state of the control S:$$\left(a|0\rangle +b|1\rangle \right)⊗|0_{\varepsilon_{1}}\rangle ⊗|0_{\varepsilon_{2}}\rangle \cdots ⊗|0_{\varepsilon_{N}}\rangle ⟹$$
$$⟹\left(a|0\rangle ⊗|0_{\varepsilon_{1}}\rangle +b|1\rangle ⊗|1_{\varepsilon_{1}}\rangle \right)⊗|0_{\varepsilon_{2}}\rangle \cdots ⊗|0_{\varepsilon_{N}}\rangle ⟹$$
$$⟹a|0\rangle ⊗|0_{\varepsilon_{1}}\rangle ⊗\cdots ⊗|0_{\varepsilon_{N}}\rangle +b|1\rangle ⊗|1_{\varepsilon_{1}}\rangle \cdots ⊗|1_{\varepsilon_{N}}\rangle .$$
It is evident that this pure decoherence dynamics is creating branching states with multiple records of the logical (as well as pointer) states $\left\{|0\rangle ,|1\rangle \right\}$ of the system in the environment. Mutual entropy between S and a subsystem $E_{k}$ can be easily computed. As the *k*’th c-not is carried out, $I(S:E_{k})$ increases from 0 to:$$I\left(S:E_{k}\right)=H_{S}+H_{E_{k}}-H_{S,E_{k}}=-{\left|a\right|}^{2}{\mathrm{lg}|a|}^{2}-{\left|b\right|}^{2}\mathrm{lg}{\left|b\right|}^{2}\;.$$
Thus, each $E_{k}$ is a sufficiently large fragment of E to supply complete information about the pointer observable of S.

The very first c-not causes complete decoherence of S in its pointer basis $\left\{|0\rangle ,|1\rangle \right\}$. This illustrates points (i)–(iii) above—the relation between the (surplus and redundant) decoherence and quantum Darwinism, the continued increase of redundancy well after coherence between pointer states is lost, and the special role of the pointer observable.

As each environment qubit is a perfect copy of the pointer states of S, redundancy *R* in this simple example is eventually given by the number of fragments that have complete information about S—that is, in this case, by the number of environment qubits, R=N. There is no need to define redundancy in a more sophisticated manner, using δ, when each environment subsystem contains perfect copy of the pointer state: It will arise only in the more realistic cases when the analogues of c-not’s are imperfect. We also note that decoherence is redundant, as each environment qubit suffices to decohere the system, so the redundancy of decoherence is also given by *N*.

Partial information plots in our example would be trivial: I(S:F) jumps from 0 to the “classical” value given by $H_{S}=-{\left|a\right|}^{2}{\mathrm{lg}|a|}^{2}-{\left|b\right|}^{2}\mathrm{lg}{\left|b\right|}^{2}$ at f=1/N, continues along the plateau at that level until f=1−1/N, and eventually jumps up again to the quantum peak at twice the level of the classical plateau as the last qubit is included: The whole SE is still in a pure state, so when F=E, $H_{S,F}=H_{S,E}=0$. However, this much information exists only in global entangled states, and is therefore accessible only through global measurements.

Preferred pointer basis of the control S is of course its logical basis $\left\{|0\rangle ,|1\rangle \right\}$. These pointer states are selected by the “construction” of c-not’s. They remain untouched by copying into consecutive environment subsystems $E_{k}$. After decoherence takes place;
$$I\left(S:F\right)=J(\left\{|0\rangle ,|1\rangle \right\}:F)$$
for any fragment of the environment when there is at least one subsystems of E correlated with S left outside of F, which suffices to decohere S. When this is the case, minimum quantum discord disappears:$$D\left(F:S\right)=I\left(S:F\right)-J(\left\{|0\rangle ,|1\rangle \right\}:F)=0\;,$$
and one can ascribe probabilities to correlated states of S and F in the pointer basis of S that are singled out by the c-not “dynamics”. Discord would reappear only if all of E got included, as then $I\left(S:F\right)=2H_{S}$, twice the information of the classical plateau of the PIP. Thus, all of E is needed to detect coherence in S: When a single environment qubit is missing, it is impossible to tell if the initial state was a superposition or a mixture of $|0\rangle$ and $|1\rangle$ of S.

As soon as decoherence sets in, $H_{S}=H_{S,F}$ for any fragment F that leaves enough of the rest of the environment $E_{/F}$ to einselect pointer states in S. Consequently;
$$I\left(S:F\right)=H_{F}\;,$$
and $H_{F}=H_{SdF}$, illustrating Equations (4.20) and (4.24).

#### 4.2.2. Central Spin Decohered by Noninteracting Spins

A generalization of a model with c-not gates and qubits is a model with a central spin system interacting with the environment of many other spins. In effect, perfect c-not’s discussed above become imperfect when a collection of spins interacts with the central spin system via Ising Hamiltonian:(4.31*a*)$$H=\sigma^{z}\sum_{i}d_{i}\sigma_{i}^{z}\;.$$
Above $\sigma^{z}$ and $\sigma_{i}^{z}$ act on the spins of the system and on the subsystems of the environment.

Several different versions of such models were studied as examples of quantum Darwinism ([131,143,144,145,146,149,151,152,153,154,155,156,157]). In this section, we focus on the steady state situation when the evolution results, at long times, in a PIP that is largely time-independent.

Hamiltonian of Equation (4.31a) provides an example of pure decoherence. It can imprint many copies of the preferred observable $\sigma^{z}$ onto the environment. Given the example of c-not’s and qubits, this is no surprise.

Copies of the pointer states of S are, of course, no longer perfect: A single subsystem of the environment is typically no longer perfectly correlated with the system. It is therefore usually impossible for a single subsystem of E to supply all the information about S. Nevertheless, when the environment is sufficiently large, asymptotic form of I(S:F) has—as a function of the size of the fragment F—the same character we have already encountered with c-not’s: A steep rise (where in accord with Equations (4.20) and (4.24), every bit of information stored in F reveals new information about S) followed by a plateau (where the information only confirms what is already known). This is clearly seen in Figure 6A: Only when the environment is too small to convincingly decohere the system, PIPs do not have a plateau.

In a central spin model with a large, initially pure and receptive environment (so that there is a pronounced classical plateau) and for a small information deficit δ, mutual information of a fragment F with ${}^{♯}\;F$ finite-dimensional subsystems is approximately [152]:(4.32)$$I\left(S:F\right)=H_{S}-\frac{1}{2}\left(e^{H_{S}}-1\right)\left(d_{E}^{-{}^{♯}\;F}-d_{E}^{-({}^{♯}\;E-{}^{♯}\;F)}\right)\;.$$
Above, $d_{E}$ is the size of the Hilbert space of the effective memory of a single environment subsystem (e.g., $d_{E}=2$ for a spin $\frac{1}{2}$ that can use all its memory to store information about S). This is a good approximation only as I(S:F) is close to the plateau: For *f* near 0 or near 1 mutual information is approximately linear in *f*, see Figure 6B, although actual dependence on *f* turns out to be more complicated in exactly solvable models (see, e.g., Touil et al., 2022) [131].

When δ≪1 and $f<\frac{1}{2}$, we can use Equation (4.32) to estimate redundancy. To this end we retain dominant terms and set $I\left(S:F\right)=\left(1-\delta \right)H_{S}$ to get:$$\left(1-\delta \right)H_{S}\approx H_{S}-\frac{1}{2}\left(e^{H_{S}}-1\right)d_{E}^{-{}^{♯}\;F_{\delta }}.$$
A simple formula for ${}^{♯}\;F_{\delta }$, the number of subsystems that reduce information deficit to δ follows when $H_{S}\gg 1$;
(4.33*a*)$${}^{♯}\;F_{\delta }\approx \log_{d_{E}}\frac{e^{H_{S}}-1}{2\delta H_{S}}\approx \frac{H_{S}-\ln2\delta H_{S}}{\ln d_{E}}\approx \frac{H_{S}}{\ln d_{E}}\;.$$
This last approximate answer shows that in the central spin model redundancy is close to what one might guess: It is given by the number of environment fragments that have enough subsystems—$H_{S}/\ln d_{E}$—to store information about S. In other words, there are approximately;
(4.33*b*)$$R_{\delta }=\frac{{}^{♯}\;E}{{}^{♯}\;F_{\delta }}\approx \frac{{}^{♯}\;E\ln d_{E}}{H_{S}}$$
fragments of E that “know” the state of the system. Note that the information deficit δ does not appear in this approximate answer: We have dropped the subdominant lnδ in Equation (4.33a).

In the discussion above $\ln d_{E}$ enters as a measure of the memory capacity of a subsystem of E. This and the universality of re-scaling in Figure 6B suggest a conjecture: The environment will fill in the space available to store information with qmems of S.

We conclude that, when only a part of the Hilbert space of the environment subsystem is available to record the state of the system S, one should be able to use just this “accessible memory” of the subsystem instead of its maximal information storage capacity $h_{m}=\ln d_{E}$ in the estimates of redundancy. We shall now corroborate this conjecture.

#### 4.2.3. Quantum Darwinism in a Hazy Environment

Inaccessibility of the memory of E can have different causes [144,145,153,158]. The most obvious one is the possibility that the environment starts in a partly mixed state, so that some of its Hilbert space is already “taken up” by the information that is of no interest to observers[note 15]. It is then tempting to use the available memory of the environmental subsystem, given by the maximal information storage capacity $h_{m}$ less *h*, the preexisting entropy, instead of its maximal memory $h_{m}=\ln d_{E}$ in Equation (4.33b). When this substitution is made, the estimated redundancy is:(4.33*c*)$$R_{\delta }^{h}\approx \frac{{}^{♯}\;E}{{}^{♯}\;F_{\delta }}\approx {}^{♯}\;E\frac{\left(h_{m}-h\right)}{H_{S}}\;,$$
or;
(4.33*d*)$$R_{\delta }^{h}\approx (1-\frac{h}{h_{m}})R_{\delta }\;.$$
That is, redundancy in a partly mixed environment, $R_{\delta }^{h}$, decreases compared to the redundancy in the pure environment $R_{\delta }$ in proportion to the memory that is still available to accept the information about the system.

In the manageable case of a single qubit system, $H_{S}=\ln2$, the substitution of $h_{m}-h$ for $\ln d_{E}$ works [145], although the expression we have used above for, e.g., $R_{\delta }$ has to be modified, as the derivation of Equation (4.33a) assumes $H_{S}\gg 1$. A rather large environment (hundreds of qubits, with symmetries of the initial state and of the interaction Hamiltonian exploited to keep the size of the memory down) was used to explore a range of values of *h*. This was necessary because the principal effect of a mixed environment is to lower the slope of the initial part of PIP’s by $\left(1-h/h_{m}\right)$, so now it takes more subsystems of E to get to the “plateau”.

The results for a central spin $\frac{1}{2}$ in the environment of spin $\frac{1}{2}$ subsystems confirmed that the redundancy decreased by approximately $1-h/h_{m}$ (Zwolak, Quan, and Zurek, 2010) [145]. This change in $R_{\delta }^{h}$ was due to the change in the slope of the early part of PIP. Equation (33c,d) became a more accurate approximation when the environment was more mixed, i.e., when *h* was, to begin with, closer to $h_{m}$. Of course, when $h=h_{m}$, no information about S can be recovered from the environment, as $H_{F}\left(t\right)$ cannot increase when it starts at a maximum. Nevertheless, as already noted, mixed environments are still very effective in decohering the system. Thus, even as the classical contribution $H_{F}\left(t\right)-H_{F}\left(0\right)=0$, disappears, the quantum contribution, $H_{S}-H_{SdE_{/}F}$, to I(S:F) remains similar to the case when E was initially pure.

As we have seen before with c-not’s and qubits, the system decoheres as soon as a single copy of its state is imprinted with a reasonable accuracy in E, and—when the environment is initially pure—a few imperfect imprints establish the initial rising part of the PIP. However, as new subsystems of E become correlated with S, the size of the plateau increases and its elongation (when plotted as a function of ${}^{♯}\;F$) occurs without any real change to the early part of the PIP (see inset in Figure 6A). Thus, the number of copies of the information E has about S can grow long after the system was decohered. This increase of the number of copies leads to the corresponding increase of the redundancy of decoherence. Moreover, redundancy of decoherence $R_{\delta_{D}}$ exceeds the redundancy of the information available to the observers $R_{\delta }$, as even mixed environments retain their undiminished ability to decohere.

#### 4.2.4. Quantum Darwinism and Pointer States

What information is redundantly acquired by E and can be recovered by observers from its fragments? In systems with discrete observables such as spins one can prove that it concerns pointer states of the system. The proof was first given in the idealized case of perfect environmental records, and then extended to the case of imperfect records (Ollivier, Poulin, and Zurek, 2004 [149]; 2005 [151]).

A natural way to characterize such correlations is to use the mutual information between an observable σ of S and a measurement e on E: Shannon mutual information I(σ:e) measures the ability to predict the outcome of measurement of σ on S after a measurement e. For a given density matrix of S⊗E, the measurements results are characterized by a joint probability distribution
$$p\left(\sigma_{i},e_{j}\right)=\mathrm{Tr}\left\{\left(\sigma_{i}⊗e_{j}\right)\rho^{SE}\right\},$$
where $\sigma_{i}$ and $e_{j}$ are the spectral projectors of observables σ and e. By definition, the mutual information is the difference between the initially missing information about σ and the remaining uncertainty about it when e is known. Shannon mutual information is defined using Shannon entropies of subsystems (e.g., $H\left(\sigma \right)=-\sum_{i}p\left(\sigma_{i}\right)\log p\left(\sigma_{i}\right)$) and the joint entropy $H\left(\sigma ,e\right)=-\sum_{i,j}p\left(\sigma_{i},e_{j}\right)\log p\left(\sigma_{i},e_{j}\right)$:I(σ:e)=H(σ)+H(e)−H(σ,e).

The information about observable σ of S that can be optimally extracted from ν environmental subsystems is
$$I_{\nu }\left(\sigma \right)=\max_{\left\{e\in M_{\nu }\right\}}I\left(\sigma :e\right)$$
where $M_{\nu }$ is the set of all measurements on those ν subsystems. In general, $I_{\nu }\left(\sigma \right)$ will depend on which particular ν subsystems are considered. For simplicity, we will assume that any *typical* ν environmental subsystems yield roughly the same information. This may appear to be a strong assumption, but relaxing it does not affect our conclusions. By setting $\nu ={}^{♯}\;E=N$ to the total number of subsystems of E, we get the information content of the entire environment. The condition;
$$I_{N}\left(\sigma \right)\approx H\left(\sigma \right)$$
expresses the *completeness* prerequisite for objectivity: All (or nearly all) missing information about σ of S must be in principle obtainable from all of E.

As a consequence of the basis ambiguity, information about many observables σ can be deduced by a suitable (generally, global) measurement on the entire environment [24,25]. Therefore, completeness, while a prerequisite for objectivity, is not a very selective criterion (see Figure 7a for evidence). To claim objectivity, it is not sufficient to have a complete imprint of the candidate property of S in the environment. There must be many copies of this imprint that can be accessed independently by many observers: *information must be redundant*.

To quantify redundancy, we count the number of copies of the information about σ present in E:$$R_{\delta }\left(\sigma \right)={}^{♯}\;E/\nu_{\delta }\left(\sigma \right)=N/\nu_{\delta }\left(\sigma \right).$$
Above $\nu_{\delta }\left(\sigma \right)$ is the smallest number of typical environmental subsystems that contain almost all the information about σ (i.e., $I_{\nu }\left(\sigma \right)\geq \left(1-\delta \right)I_{N}\left(\sigma \right)$).

The key question now is: What is the structure of the set O of observables that are **completely**, $I_{N}\left(\sigma \right)\approx H\left(\sigma \right)$, and **redundantly**, $R_{\delta }\left(\sigma \right)\gg 1$ with δ≪1, imprinted on the environment? The answer is provided by the theorem:

**Theorem** **4.***The set O is characterized by a unique observable Π, called by definition the***maximally refined observable***, as the information $I_{\nu }\left(\sigma \right)$ about any observable σ in O obtainable from a fraction of E is equivalent to the information about σ that can be extracted from its correlations with the maximally refined observable Π:*(4.34)$$I_{\nu }\left(\sigma \right)=I\left(\sigma :\Pi \right)$$*for $\nu_{\delta }\left(\Pi \right)\leq m\ll N$*.

**Proof.** Let $\sigma^{\left(1\right)}$ and $\sigma^{\left(2\right)}$ be two observables in O for δ=0. Since $\sigma^{\left(1\right)}$ and $\sigma^{\left(2\right)}$ can be inferred from two disjoint fragments of E, they must commute. Similarly, let $e^{\left(1\right)}$ (resp. $e^{\left(2\right)}$) be a measurement acting on a fragment of E that reveals all the information about $\sigma^{\left(1\right)}$ (resp. $\sigma^{\left(2\right)}$) while causing minimum disturbance to $\rho^{SE}$. Then, $e^{\left(1\right)}$ and $e^{\left(2\right)}$ commute, and can thus be measured *simultaneously*. This combined measurement gives complete information about $\sigma^{\left(1\right)}$ *and* $\sigma^{\left(2\right)}$. Hence, for any pair of observables in O, it is possible to find a more refined observable which is also in O. The maximally refined observable Π is then obtained by pairing successively all the observables in O. By construction Π satisfies equality $I_{\nu }\left(\sigma \right)=I\left(\sigma :\Pi \right)$ for any σ in O.    □

Theorem 4 can be extended to nearly perfect records for assumptions satisfied by usual models of decoherence (Ollivier, Poulin, and Zurek, 2005 [151]). The proof is based on the recognition that only the already familiar pointer observable can have a redundant and robust imprint on E. This Theorem can be understood as *a consequence of the ability of the pointer states to persist while immersed in the environment*. This resilience allows the information about the pointer observables to proliferate, very much in the spirit of the “survival of the fittest”.

Note that the above Theorem does not guarantee the existence of a *non trivial* observable Π: when the system does not properly correlate with E, the set O will only contain the identity operator.

Two important consequences of this theorem follow: (*i* ) An observer who probes only a fraction of the environment is able to find out the state of the system as if he measured Π on S; (*ii* ) Information about any other observable ς of S will be inevitably limited by the available correlations existing between ς and Π. In essence, our theorem proves the uniqueness of redundant information, and therefore the selectivity of its proliferation.

We can illustrate this preeminence of the pointer observable in our simple model: a single central spin $\frac{1}{2}$ interacting with a collection of *N* such spins, Equation (4.31a). As seen in Figure 7a environment as a whole contains information about any observable of S. Preferred role of the pointer observable becomes apparent only when one seeks observables that are recorded *redundantly* in E. Figure 7b shows that only the pointer observable $\Pi =\sigma_{z}$ (and observables that are nearly co-diagonal with it) are recorded redundantly, illustrating the theorem quoted above. The “ridge of redundancy” is strikingly sharp.

Further confirmation and extension of the theorem quoted above is the relation (derived under the assumption of surplus decoherence; Zwolak and Zurek, 2013 [130]) between the available information (characterized by the Holevo quantity χ, measure of the capacity of the quantum information channel for classical information) about an arbitrary observable ς and the pointer observable Π of the system, see Equation (28), and its consequences, Equations (29a) and (b).

Comparison of Figure 7a,b also shows that redundancy of $\sigma_{z}$ increases long after the environment as a whole is strongly entangled with S. This is seen in a steady rise of the redundancy $R_{\delta }$ with the action. Thus, as anticipated, redundancy can continue to increase long after the system has decohered.

The origin of the consensus between different observers is the central lesson that follows from our considerations. In everyday situations, observers have no choice in the observables of systems of interest they will measure. This is because they rely on the “second hand” information they obtain from the same environment that is responsible for decoherence. In addition, the environment that selects a certain pointer observable will record redundantly only the information about that observable.

Information about complementary observables is in principle still “out there”, but one would have to intercept essentially all of the environment (to be more precise, all but its $1-\frac{1}{R_{\delta_{D}}}\geq 1-\frac{1}{R_{\delta }}$ fraction) and measure it in the right way (that is, using an observable with entangled eigenstates) to have any hope of acquiring that information. By contrast, to find out about the pointer observables, a small fraction of the environment $∼\frac{1}{R_{\delta }}$ is enough.

As we shall see, redundancies for, e.g., the photon environment are astronomical. It is therefore no surprise that we rely on the information that can be obtained with little effort from a small fraction of E. In addition, it is also no surprise that the complementary information is inaccessible. Therefore (and as is established through a sequence of results in Girolami et al., 2022 [142] the evidence of quantumness is unavailable from the fragments of E. Moreover, observers that find out about their Universe in this way will agree about the outcomes. This is how objective classical reality emerges from a quantum substrate in our quantum Universe.

#### 4.2.5. Redundancy vs. Relaxation in the Central Spin Model

We have seen defining characteristics of quantum Darwinism in the central spin model. Thus; (i) decoherence begets redundancy which; (ii) can continue to increase long after decoherence saturated entropy of the system at $H_{S}$. Moreover, (iii) both decoherence and quantum Darwinism single out the same pointer observable. There are multiple copies of the pointer observable in the environment, and only the information about the pointer states is amplified by the decoherence process. This happens at the expense of the information about the complementary observables (i.e., information about the phases between the pointer states).

Our model is illustrated in Figure 8. It differs from the central spin models we have investigated as now the spins of the environment interact and can exchange information. The effect of the interactions between the environmental subsystems on partial information plots and on the redundancy is seen in Figure 9. The coupling of the central spin to the environmental spins is stronger than their couplings to each other. As a result, partial information plots quickly assume the form characteristic of quantum Darwinism caused by pure decoherence, and redundancy rises to values of the order of the size of the environment.

Initially, pure decoherence is a reasonable approximation. However, as time goes on, correlations between spins of the environment gradually build up. Interactions between the spins of E mean that the state of a fragment of the environment begins to entangle with—begins to acquire information about—the rest of the environment. As a consequence, the information individual spins had about the system becomes delocalized as it is shared with the rest of E. The structure of the states of SE is no longer branching—it is no longer possible to represent them by Equation (4.17). More general entangled states of the Hilbert space of SE are explored. Eventually, interactions may take the systems closer to equilibrium (although we do not expect complete equilibration in our model—the central spin decoheres, but its pointer states remain unaffected by the environment). Consequently, over time, PIPs are expected to change character from the steep rise followed by a long plateau characteristic of pure decoherence (red graph in Figure 5) to the approximate step function shape (green graph in Figure 5) that is characteristic of a collection of spins in equilibrium (see Page, 1993 [159]).

All of the models investigated so far were “pure decoherence”—subsystems of the environment interacted only with the system S via an interaction that left the pointer basis untouched. This assured validity of Equation (4.20), so that the information gained by the environment about the pointer states of S remained localized, available from the fragments of the environment.

The idealization of pure decoherence is often well-motivated. Photon environment, for example, consists of subsystems (photons) that interact with various systems of interest but do not interact with one another. There are, however, other environments—such as air—that contribute to or even dominate decoherence, but consist of interacting subsystems (air molecules). Thus, while the pointer basis is still untouched by decoherence, information about it will no longer be preserved in the individual subsystems of the environment—it will become delocalized, and, hence, impossible to extract from the local fragments of the environment consisting of its natural subsystems.

To investigate what happens when subsystems of E interact and exchange information we relax the assumption of pure decoherence and consider a model that adds to the central spin model of Equation (4.31a) interactions between the environmental spins:(4.31*b*)$$\;\;\;\;\;\;\;\;H=\sigma^{z}\sum_{i}d_{i}\sigma_{i}^{z}+\sum_{j,k}m_{jk}\sigma_{j}^{z}\sigma_{k}^{z}.$$
As a consequence, the information about the system of interest is still present in the correlations with the environment, but it will gradually become encrypted in non-local states of E that are inaccessible to local observers—i.e., that do not provide information about S via measurements that access small ($f<\frac{1}{2}$) fragments of E consisting of its subsystems.

The timescale over which pure decoherence is a good approximation depends on the strength of the couplings. In our case, the coupling between the system and the spins of the environment is significantly stronger than the couplings between the spins of E. As a result, states of SE acquire initially an approximately branching structure and have PIP’s that allow significant redundancy to develop. However, over time, interactions within E take the system closer to equilibrium, and PIPs change. More detailed discussion of this can be found in the caption of Figure 9, and, especially, in Riedel, Zurek, and Zwolak, 2012 [153].

Our simple model with weakly interacting environment subsystems illustrates why environments where the subsystems (photons) are in effect non-interacting are used by observers to gather information rather than environments (such as air) that may be more effective in causing decoherence but scramble information acquired in the process because their subsystems (air molecules) interact with each other.

#### 4.2.6. Quantum Darwinism in Quantum Brownian Motion

Evolution of a single harmonic oscillator (the system) coupled through its coordinate with a collection of many harmonic oscillators (the environment) is a well known exactly solvable model (Feynman and Vernon, 1963 [160]; Dekker, 1977 [161]; Caldeira and Leggett, 1983 [162]; Unruh and Zurek, 1989 [163]; Hu, Paz, and Zhang, 1992 [164], Paz, Habib, and Zurek, 1993 [165]; Tegmark and Wheeler, 2001 [166]; Bacciagaluppi, 2004 [167]).

This is not a pure decoherence model. While the environment oscillators do not interact, the self-Hamiltonian of S does not commute with the interaction Hamiltonian. Therefore, when the oscillations are underdamped preferred states selected by their predictability are Gaussian minimum uncertainty wavepackets (Zurek, Habib, and Paz, 1993 [63]; Tegmark and Shapiro, 1994 [168]; Gallis, 1996 [169]). This is in contrast to spin models (including the model we have just discussed) where exact and orthogonal pointer states can be often identified. So, while decoherence in this model is well understood, quantum Darwinism—where the focus is not on S, but on its relation to a fragment F of E—presents novel challenges.

Here, we summarize results (Blume-Kohout and Zurek, 2008 [170]; Paz and Roncaglia, 2009 [171]) obtained under the assumption that fragments of the environment are “typical” subsets of its oscillators—that is, subsets of oscillators with the same spectral density as the whole E.

The QBM Hamiltonian:$$H=H_{S}+\frac{1}{2}\sum_{\omega }\left(\frac{q_{\omega }^{2}}{m_{\omega }}+m_{\omega }\omega^{2}y_{\omega }^{2}\right)+x_{S}\sum_{\omega }C_{\omega }y_{\omega }$$
describes a collection of the environment oscillators coupled to the harmonic oscillator system with:$$H_{S}=\left(\frac{p_{S}^{2}}{m_{S}}+m_{S}\Omega_{0}^{2}x_{S}^{2}\right)/2,$$
and the environmental coordinates $y_{\omega }$ and $q_{\omega }$ describe a single band (oscillator) $E_{\omega }$. As usual, the bath is defined by its spectral density, $I\left(\omega \right)=\sum_{n}\delta \left(\omega -\omega_{n}\right)\frac{C_{n}^{2}}{2m_{n}\omega_{n}},$ that quantifies the coupling between S and each band of E. An *ohmic* bath with a sharp cutoff Λ: $I\left(\omega \right)=\frac{2m_{S}\gamma_{0}}{\pi }\omega$ for ω∈[0…Λ] was adopted: A sharp cutoff (rather than the usual smooth rolloff) simplifies numerics. Each coupling is a differential element, $dC_{\omega }^{2}=\frac{4m_{S}m_{\omega }\gamma_{0}}{\pi }\omega^{2}d\omega$ for ω∈[0…Λ]. For numerics, whole range of frequencies [0…Λ] was discretized by dividing it into into discrete bands of width Δω, which approximates the exact model well up to a time $\tau_{rec}∼\frac{2\pi }{\Delta \omega }$.

The system was initialized in a squeezed coherent state, and E in its ground state. QBM’s linear dynamics preserve the Gaussian nature of the state, which can be described by its mean and variance:$$\vec{z}=\left(\begin{array}{c} \langle x\rangle \\ \langle p\rangle \end{array}\right)\;;\;\;\Delta =\left(\begin{array}{cc} \Delta x^{2} & \Delta xp\\ \Delta xp & \Delta p^{2} \end{array}\right).$$
Its entropy is a function of $a^{2}={\left(\frac{\hbar }{2}\right)}^{-2}\det\left(\Delta \right)$, its squared symplectic area. Thus;
(4.35)$$H\left(a\right)=\frac{1}{2}\left((a+1)\ln(a+1)-(a-1)\ln(a-1)\right)-\ln2\approx \ln\left(\frac{e}{2}a\right),$$
where *e* is Euler’s constant, and the approximation is excellent for a>2. For multi-mode states, numerics yield H(ρ) exactly as a sum over Δ’s symplectic eigenvalues (Serafini et al., 2004) [172]. The theory proposed in Ref. [170] approximates a collection of oscillators as a single mode with a single $a^{2}$.

Mutual information illustrated in partial information plots (Figure 10) shows that I(S:F)—the information about S stored in F—rises rapidly as the fraction of the environment *f* included in F increases from zero, then flattens for larger fragments. Most—all but ∼1 nat—of $H_{S}$ is recorded redundantly in E. When S is macroscopic, this *non-redundant information* is dwarfed by the total amount of information available from even small fractions of E.

Calculations simplify in the macroscopic limit where the mass of the system is large compared to masses of the environment oscillators. This regime (of obvious interest to the quantum—classical transition) allows for analytic treatment based on the Born-Oppenheimer approximation: Massive system follows its classical trajectory, largely unaffected by E. The environment will, however, decohere a system that starts in a superposition of such trajectories. In the process, E that starts in the vacuum will become imprinted with the information about the location of S.

The basic observation is that the area of the 1−σ contours in phase space determines entropy. As a result of decoherence, the squared symplectic area corresponding to the state of the system will increase by $\delta a_{S}^{2}$. This is caused by the entanglement with the environment, so the entropies and symplectic areas of environment fragments increase as well. When F contains a randomly selected fraction *f* of E, $\rho_{F}$’s squared area is $a_{F}^{2}=1+f\delta a_{S}^{2}$, and that of $\rho_{SF}$ is $a_{SF}^{2}=1+\left(1-f\right)\delta a_{S}^{2}$. Applying Equation (4.35) (where $\delta a_{S}^{2}\gg 1$) yields:(4.36)$$I\left(S:F\right)\approx H_{S}+\frac{1}{2}\ln\left(\frac{f}{1-f}\right).$$
This “universal” I(S:F) (Blume-Kohout and Zurek, 2008 [170]; Roncaglia and Paz, 2009 [171]) is valid for significantly delocalized initial states of S (which implies large $H_{S}$). It is a good approximation everywhere except very near f=0 and f=1 (where it would predict singular behavior). It has a classical plateau at $H_{S}$ which rises as decoherence increases entropy of the system.

In contrast to PIP’s we have seen before (e.g, in the central spin model), adding more oscillators to the environment does not simply extend the plateau: The shape of I(S:F) is only a function of *f* and so it is invariant under enlargement of E. This is because the couplings of individual environment oscillators are adjusted so that the damping constant of the system oscillator remains the same. Therefore, increasing the number of oscillators in the environment does not really increase the number of the copies of the state of the system in E—in a sense it only improves the accuracy with which the environment with a continuum distribution of frequencies (e.g., a field) is modeled using discrete means.

When the above equation for I(S:F) is solved for $f_{\delta }$ one arrives at the estimate for the redundancy:(4.37)$$R_{\delta }\approx e^{2\delta H_{S}}\approx s^{2\delta }.$$
The last equality above follows because an *s*-squeezed state decoheres to a mixed state with $H_{S}\approx \ln s$. This simple last formula for $R_{\delta }$ holds where it matters—after decoherence but before relaxation begins to force the system to spiral down towards its ground state.

As trajectories decay, plateau flattens compared to what Equation (4.36) would predict. This will initially increase redundancy $R_{\delta }$ above the values attained after decoherence (see Figure 11). Eventually, as the whole SE equilibrates, the system will spiral down to occupy a mixture of low-lying number eigenstates, and $R_{\delta }$ will decrease.

Quantum Brownian motion model confirms that decoherence leads to quantum Darwinism. However, details of quantum Darwinism in QBM setting are different from what we have becomes accustomed to in the models involving discrete Hilbert spaces of S and of subsystems of E. Partial information plots with the shape that is independent of the size of E and a scaling of redundancy with the information deficit δ that is not logarithmic as before are clear manifestations of such differences.

Buildup of redundancy still takes longer than the initial destruction of quantum coherence. Nevertheless, various time-dependent processes (such as the increase of redundancy caused by dissipation) remain to be investigated in detail. Moreover, localized states favored by einselection are redundantly recorded by E. So, quantum Darwinism in QBM confirms many of the features of decoherence we have anticipated earlier. On the other hand, we have found an interesting tradeoff between redundancy and information deficit δ, Equation (4.37). It suggests that, in situations where QBM is applicable, objectivity (as measured by redundancy) may come at the price of accuracy.

We also note that surplus decoherence we have described before can be found in the case of QBM. This follows, in effect, from the fact that $a_{SF}^{2}=1+\left(1-f\right)\delta a_{S}^{2}$ approaches $a_{S}^{2}=1+\delta a_{S}^{2}$ for small *f*. Consequently, it is evident that $H_{S}-H_{S,E_{/F}}\approx 0$, and the mutual information I(S:F) is given by $H_{F}$. This is not obvious from the simple scale-invariant Equation (4.37) above.

To sum up, we note that while broadly defined tenets of quantum Darwinism—multiple records of S in E, buildup of redundancy to large values, etc.—are satisfied in QBM, there are also interesting differences. Thus, QBM—in contrast to the spin models—does not have an obvious version in which decoherence is pure (as there is no perfect pointer observable that commutes with the whole Hamiltonian of SE). However, when the mass of S is large and the initial state is delocalized in position, at least early on pure decoherence is a good (Born-Oppenheimer—like) approximation.

Eventually the collection of oscillators begins to relax, and the information about the system flows from their individual states to correlations between them. This is because, even though the oscillators of the environment do not directly interact with each other, they do interact indirectly via the system oscillator: There is no perfect pointer observable for our harmonic oscillator S. The obvious consequence of this is the damping suffered by S. Less obvious is its role in coupling of the environment oscillators: Even though they do not interact directly (as did spins of the environment in Equation (4.31b)), they exchange information about one another indirectly, via S, which in time creates entanglements that make it more difficult to extract information about S from the fragments of E defined by collections of the (original) oscillators.

#### 4.2.7. Huge Redundancy in Scattered Photons

The two decoherence models discussed above—central spin model and quantum Brownian motion—are the two standard workhorses of decoherence. They were the early focus of quantum Darwinism primarily because one could analyze them using many of the tools developed to study decoherence. We have thus seen quantum Darwinism in action, and we have already confirmed in these idealized models that the expectations about the shape of partial information plots resulting from decoherence and about the buildup of redundancy are satisfied—with variations—in both cases. This is reassuring. However, while redundancies appeared in both cases, they were modest ($R_{\delta }∼10$), in part as a result of the limited size of the environment. Moreover, neither model is an accurate representation of how we find out about our world.

In our Universe vast majority of data acquired by human observers comes via the photon environment. A fraction *f*, usually corresponding to a very small fragment $F_{f}$ of the photon environment scattered or emitted by the “system of interest” is intercepted by our eyes. This is how we find out about what we have grown accustomed to regard as “objective classical reality”.

It is fair to expect that the photon environment should have significant redundancies—we use up only a tiny fraction of photon evidence, and others who look at the same systems generally agree about their states. To investigate quantum Darwinism in (photon) scattering processes we turn to the model of decoherence discussed by Joos and Zeh (1985) [174] that was since updated (Gallis and Fleming, 1990 [175]; Hornberger and Sipe, 2003 [176]; Dodd and Halliwell, 2003 [177]; Halliwell, 2007 [178]) and applied by others, for example to calculate decoherence in the fullerene experiments (Hornberger et al., 2003 [179]; Hackermüller et al., 2004 [180]). The book by Schlosshauer (2007) [32] provides a good overview.

In contrast to Joos and Zeh (1985) [174] who focused on decoherence caused by the isotropic black-body radiation we are interested in the information content of the scattered photons. We shall therefore primarily consider a distant point source that illuminates an object—a dielectric sphere of radius *r*—that is initially in a non-local superposition (see Figure 12), although we shall also discuss the isotropic case as a counterpoint. We shall assume thermal distribution of energies (and, hence, wavelengths) of the incoming radiation and (at least in the results we shall focus on below) we will assume that photons come as a plane wave from a single direction (which approximates illumination by a distant localized light source such as the Sun or a light bulb). Generalizations to other models of illumination have been considered by Riedel and Zurek, 2011 [158].

The scattering process is responsible for the decoherence of S and for the imprinting of information about the location of the scatterer S in the photon environment. As the initial state of S we take a nonlocal “Schrödinger cat” superposition of two locations:$$|\psi_{S}\left(\vec{x}\right)\rangle =\left(|{\vec{x}}_{1}\rangle +|{\vec{x}}_{2}\rangle \right)/\sqrt{2}\;.$$
We ignore the self-Hamiltonian of the sphere so that its pointer states are localized in space. One can justify this approximation by pointing to the large mass of the sphere. Large mass also enables our other approximation—we shall ignore the momentum imparted to the sphere by the photons. Note that—under these assumptions—scattering of photons from S results in pure decoherence.

Scattering takes the initial pure state density matrix $\rho_{S}^{0}=|\psi_{S}\left(\vec{x}\right)\rangle \;\langle \psi_{S}\left(\vec{x}\right)|$ into a mixture with the off-diagonal terms:$$|\langle {\vec{x}}_{1}|\rho_{S}|{\vec{x}}_{2}\rangle {|}^{2}=\gamma^{N}|\langle {\vec{x}}_{1}|\rho_{S}^{0}|{\vec{x}}_{2}\rangle {|}^{2}=\Gamma {\left|\langle {\vec{x}}_{1}|\rho_{S}^{0}|{\vec{x}}_{2}\rangle \right|}^{2}\;.$$
Above, γ is the decoherence factor corresponding to scattering by a single photon. It is given by:(4.38)$$\gamma =|\mathrm{Tr}\left(S_{{\vec{x}}_{1}}\rho_{S}^{0}S_{{\vec{x}}_{2}}^{†}\right){|}^{2}\;,$$
where $S_{{\vec{x}}_{i}}$ is the scattering matrix acting on the photon when the dielectric sphere S is located at ${\vec{x}}_{i}$. Scattering by *N* photons results in the decoherence factor $\Gamma =\gamma^{N}$. The entropy of the decohered system turns out to be:(4.39)$$H_{S}=\ln2-\sum_{n=1}^{\infty }\frac{\Gamma^{n}}{2n(2n-1)}=\ln2-\sqrt{\Gamma }\mathrm{arctanh}\sqrt{\Gamma }-\ln\sqrt{1-\Gamma }\;.$$

To compute the decoherence factor Γ we use the classical cross section of a dielectric sphere in the dipole approximation (where the wavelength of photons is much larger than the size of the sphere and the photons are not sufficiently energetic to individually resolve the superposition). Under these assumptions the decoherence factor Γ due to blackbody radiation can be obtained explicitly, and has a form $\Gamma =\exp(-t/\tau_{D})$ where $\tau_{D}$ is the decoherence time. Its inverse, the decoherence *rate*, is given by:(4.40)$$\frac{1}{\tau_{D}}=C_{\Gamma }\left(3+11\cos^{2}\theta \right)\frac{I{\tilde{a}}^{6}\Delta x^{2}k_{B}^{5}T^{5}}{c^{6}\hbar^{6}},$$
where $C_{\Gamma }=161,280\;\zeta \left(9\right)/\pi^{3}\approx 5210$ is a numerical constant, *I* is the *irradiance* (that is, radiative power per unit area) while $\tilde{a}\equiv r{\left[\left(\epsilon -1\right)/\left(\epsilon -2\right)\right]}^{1/3}$ is the effective radius of the sphere that takes into account its permittivity ϵ, and θ is the angle between the direction of incoming plane wave $\hat{n}$ and $\vec{\Delta x}$ (see Figure 12).

The decoherence rate does not increase for arbitrarily large Δx with the square of the separation, as Equation (4.40) would indicate. Rather, this expression is only valid in the Δx≪λ limit we are considering here. For Δx≫λ, the decoherence rate saturates [175];
(4.41)$$\frac{1}{{\tilde{\tau }}_{D}}={\tilde{C}}_{\Gamma }\frac{I{\tilde{a}}^{6}k_{B}^{3}T^{3}}{c^{4}\hbar^{4}},$$
with ${\tilde{C}}_{\Gamma }=57,600\;\zeta \left(7\right)/\pi^{3}\approx 1873$. In the intermediate region, where Δx∼λ, decoherence time $\tau_{D}$ has a complicated dependence on both Δx and θ. The results discussed here are valid for *all*Δx providing that the correct $\tau_{D}$ is used.

To obtain the mutual information we use the (pure decoherence) identity $I\left(S:F\right)=\left(H_{F}-H_{F}\left(0\right)\right)+\left(H_{S}-H_{SdE_{/F}}\right)$, Equation (4.20). The result is:(4.42)$$I\left(S:F_{f}\right)=\ln2+\sum_{n=1}^{\infty }\frac{\Gamma^{(1-f)n}-\Gamma^{fn}-\Gamma^{n}}{2n(2n-1)}.$$
Figure 13 shows the plot of this mutual information as a function of the fraction of the environment *f* for several times. For large *t* (small Γ) the sum is dominated by the lowest power of Γ. Thus, for $f<\frac{1}{2}$ we have:(4.43)$$I\left(S:F_{f}\right)=\ln2-\frac{1}{2}\Gamma^{f}.$$
This allows us to estimate redundancy for δ<0.5 as;
(4.44)$$R_{\delta }\approx \frac{1}{\ln(2\delta \ln2)}{\frac{t}{\tau }}_{D}.$$
As in the case with the central spin (but not with the quantum Brownian motion) redundancy depends only weakly—logarithmically—on the information deficit δ.

What is even more important, redundancy continues to increase linearly with time at a rate given by the inverse of the decoherence time $\tau_{D}$. This is different than in either central spin or quantum Brownian motion models. This difference is due to the nature of these models: There the subsystems of the environment continued to interact with the decohered system, so that the information they acquired about S sloshed back and forth between S and correlations with E. As a result, a steady state was reached, and redundancy saturated at modest values. In the case of photons the transfer of information is unidirectional—they scatter and go off to infinity (or fall into the eye of the observer). Even a tiny speck of dust can decohere quickly in a photon environment at very modest cosmic microwave background temperatures as was noted by Joos and Zeh, 1985 [174].

Redundancy of illuminated objects can quickly become enormous [158,181]. For instance, a 1μm speck of dust in a superposition of Δx∼1μm on the surface of Earth illuminated by sunlight ($T=5250^{\circ }$K) would produce $R_{\delta }∼10^{8}$ records of its state—its location—in just 1 microsecond. As a consequence, huge redundancies that grow linearly with time are inevitable.

By contrast with the case of the point source (or, more generally, with the case where illumination comes from more than one direction (Riedel, and Zurek, 2011) [158]), isotropic illumination by black body radiation (Figure 13b) does not result in the buildup of redundancy. This is understandable, as a maximum entropy of blackbody radiation that fills in all the space near the system has no more room to store the information about the location of S—its initial entropy $H_{F}\left(0\right)$ is already at a maximum. As a result, $H_{F}-H_{F}\left(0\right)=0$, and the classical, locally accessible contribution to mutual information disappears. This is in spite of the fact that quantum decoherence caused by blackbody radiation is very effective.

Rapid increase and large values of redundancy signify objectivity—many ($∼10^{8}$ observers after only 1 microsecond!) could in principle obtain the same information and will agree about the state of the systems. Thus, quantum Darwinism resulting from the photon environment is very effective, and accounts for the emergence of the “objective classical reality” in our quantum Universe.

The flip side of the huge redundancies is irreversibility—the difficulty of undoing redundant decoherence. Restoring the state of our (modest) ersatz Schrödinger cat to the preexisting superposition would require control and manipulation all of the fragments of the environment. In our example this means $∼10^{14}$ fragments that, after just one second, independently “know” the location of S. Recovery of phase coherence—reversal of decoherence—requires intercepting all of E, including the very last fraction that has a record of S and that corresponds to the“quantum rise” (at f→1) in the plots of the mutual information. Moreover, such manipulations of ES would involve global observables.

Measurement is *de facto* irreversible not just because observer in possession of the record of its outcome cannot reverse evolution that led to the wavepacket collapse (as discussed in Section 2), and not because “observers choose to ignore the environment”, but because they cannot get hold of and control all of ∼$10^{14}$ (or more) fragments of E with the record of the outcome—and that would be a precondition for the attempted reversal. This is especially obvious with photons: As soon as a minute fraction (∼10${}^{-14}$ of the photons that scattered) escape within a second the irreversible “reduction of the state vector” is a *fait accompli*. Moreover, escaped photons are gone for good—reversibility of the dynamics of this process is trumped by relativistic causality.

### 4.3. Experimental Tests of Quantum Darwinism

Experimental study of quantum Darwinism faces the problem familiar already from the tests of decoherence: Both decoherence and quantum Darwinism are so efficient that in everyday life their consequences are taken for granted and even in the laboratory it is difficult to find situations where the effect of the environment can be adjusted at will and quantified.

In the study of decoherence this problem was bypassed by using carefully tuned microsystems with controllable coupling to the environment (Brune et al., 1996 [182]; Hackermüller et al., 2004 [180]; Haroche and Raimond, 2006 [183]). Similar strategy was adopted in the experimental studies of quantum Darwinism, although the branching states that contain information about S imprinted on E have to contend with decoherence of the composite SE coupled to the more distant degrees of freedom (that cause additional decoherence of SE as a whole): As E with the record of the pointer states of S grows, the differences between the growing branches seeded by the pointer states increase, and become more susceptible to decoherence.

The strategy of finding mesoscopic systems where the process that is key to quantum Darwinism—proliferation of multiple copies of information about S—can be controlled and studied has nevertheless led to several experiments. Thus, the groups of Mauro Paternostro (Ciampini et al., 2018) [184] and Jian-Wei Pan (2019) [185] carried out what amounts to logical gates to imprint information about a system initiated in a superposition of pointer states on a collection of photons. The goal was confirmation of the key idea—that a small fraction of E suffices to gain almost all information about S, and that enlarging that fraction confirms what was already found out. Results are consistent with this expectation. Similar strategy (and similarly positive results) was obtained in an emulation of quantum Darwinism on an IBM 5-qubit quantum computer (Chisholm et al., 2021) [186].

The group of Fedor Jelezko (Unden et al., 2019) [187] used nitrogen vacancy (NV) center as a system, relying on its natural interaction with four surrounding C${}^{13}$ nuclei in the diamond, as illustrated in Figure 14. The density matrix of SE was reconstructed using quantum tomography. The resulting partial information plots demonstrate that the gain of information upon the measurement of the first fragment of E is largest. As anticipated, information gains from the additional fragments are show diminishing returns.

The results of all of these experiments are consistent with what was expected, with one exception: If SE as a whole was isolated from the other environments, there should have been a corresponding “uptick” in mutual information as ${}^{♯}\;F\to {}^{♯}\;E$. There was no convincing evidence of that signature of the overall purity. This is hardly surprising: As noted earlier, in addition to deliberate decoherence resulting in quantum Darwinism, the system-environment composite interacts with other degrees of freedom. Phase coherence between the branches corresponding to the amplified pointer states of SE is then suppressed by that spurious decoherence.

### 4.4. Summary: Environment as an Amplification Channel

Decoherence has made it clear that quantum states are far more fragile than their classical counterparts. A state that is stable (and, therefore, can aspire to classicality) is selected—einselected—with the environment having decisive say in the matter. However, even this dramatic change of view (that limits usefulness of the idealization of “isolated systems”—mainstay of classical physics—in explaining how our quantum Universe works) turns out to underestimate the role of the environment.

Quantum Darwinism demonstrates that preferred states are not only selected for their stability (ability to survive the “hostile environment”) but are communicated by the very same environment that also serves as a communication channel. Therefore, the environment acts both as a censor (for some states) and as an advertising agent that disseminates many copies of the information about the pointer states while suppressing complementary information about their superpositions.

Regarding the environment as a communication channel is more than a figure of speech: A quantum communication channel can be regarded (see, e.g., Wilde, 2013 [188]; Preskill, 2020 [189]) as a correlated state of an input and an output. Quantum channel used to transmit classical information can be represented by a state:$$\rho_{SF}=Tr_{E_{/F}}\rho_{SE}≃\sum_{k}p_{k}|\pi_{k}\rangle \;\langle \pi_{k}|\rho_{F_{k}}\;,$$
where $|\pi_{k}\rangle$ are effectively classical (i.e., orthogonal) input states—messages that are to be communicated. They are encoded in the output density matrices $\rho_{F_{k}}$ that are the records of $|\pi_{k}\rangle$, and that are to be eventually measured. This was our Equation (4.26).

In the idealized situation when both S and E start pure, the initial state of the whole is also pure:$$\rho_{SE}=|\Psi_{SE}\rangle \;\langle \Psi_{SE}|$$
where (see Equation (4.17));
$$|\Psi_{SE}\rangle =\sum_{k}e^{i\phi_{k}}\sqrt{p_{k}}|\pi_{k}\rangle |\varepsilon_{k}^{\left(1\right)}\rangle |\varepsilon_{k}^{\left(2\right)}\rangle \cdots |\varepsilon_{k}^{\left(l\right)}\rangle \cdots$$
Tracing over a part of the environment yields a state of the form of $\rho_{SF}$ above, where $\rho_{F_{k}}$ represent states of the fragments correlated with the effectively classical pointer states $|\pi_{k}\rangle$. Moreover, “roots” of the branches $|\pi_{k}\rangle \;\langle \pi_{k}|$ that appear in $\rho_{SF}$ are largely independent of what part of the environment is traced over for most of the range of the possible sizes of the the $E_{/F}$: This is an excellent approximation for $f_{\delta }<f<1-f_{\delta }$, that is, all along the plateau of the partial information plots. Thus, the same preferred branches of SF are singled out regardless of what part of the environment is detected, and what part is out of reach at least as long as $E_{/F}$ suffices to decohere the rest.

This transformaton of the communicated message from quantum to classical is implied by one more feature of $\rho_{SE}$ that can be (see Equation (4.30)) approximated as:$$\rho_{SE}≃\sum_{k}p_{k}|\pi_{k}\rangle \;\langle \pi_{k}|\overset{R_{\delta }}{\underset{l}{⨂}}\rho_{F_{k}}^{\left(l\right)}\;,$$
when the off-diagonal terms of $\rho_{SE}$ (obviously still present in $|\Psi_{SE}\rangle \;\langle \Psi_{SE}|$) are out of reach, as would be the case for an observer who cannot acquire all of SE and measure it in the correct basis. Thus, Equation (4.30) (reproduced above) provides a justification of the existence of branches with many ($R_{\delta }$) copies of the same message, suggesting a natural extrapolation “by induction” of the objective existence of the “root” $|\pi_{k}\rangle$ of the branch[note 16].

Independence of the communicated message from $E_{/F}$, the remainder of the environment that is traced over, is key: All the fragments of the environment know about the same observable of the system. Thus, omitting some part of the environment is perfectly justified—up to the information of the order of the information deficit δ it does not alter what is known about the system of interest.

Indeed, it is usually an excellent approximation to assert that, if we were to look at a macroscopic S—if we intercepted even a very small fraction of E, $f\ll f_{\delta }$, just a few of the photons that bounced off of S—the information we would get would be consistent with S occupying a single as yet unknown pointer state. Our “looking” is usually insufficient to identify such a single state, but if we continued with more precise measurements, we would eventually conclude based on all the data gathered from E (and confirmed by direct measurement of S, if desired) that whatever pure state we inferred from the incomplete partial information was in the end gradually revealed by the data gathered along the way.

In other words, in many situations our acquisition of information will correspond to the initial rising part of the partial information plot. Our confidence about the existence of a definite state at the end of the information acquisition process is based on cases when the plateau is reached, or even simply on the fact that the additional, higher resolution information is consistent with the information acquired earlier.

The step from the epistemic (“I have evidence of $|\pi_{17}\rangle$”.) to ontic (“The system is in the state $|\pi_{17}\rangle$”.) is then an extrapolation justified by the nature of $\rho_{SE}$: Observers who detected evidence consistent with $|\pi_{17}\rangle$ will continue to detect data consistent with $|\pi_{17}\rangle$ when they intercept additional fragments of E. So, while the other branches may be in principle present, observers will perceive only data consistent with the branch to which they got attached by the very first measurement. Other observers that have independently “looked at” S will agree.[note 17]

Objective existence of classical reality turns out to be an enormously simplifying, exceedingly accurate, and, therefore, very useful approximation. Thus, when agents’ success depends on acting in response to perceived “objective reality”, they can do that with confidence based on the indirect data obtained from E.

There is a sense in which our strategy explores and vindicates the artificial division of the Universe between quantum and classical introduced by Heisenberg—what John Bell (1990) [193] called a “shifty split”—in measurements. “Shifty” in the description of the “split” was not meant to be a compliment. The split happened somewhere along the von Neumann chain that connected the quantum system with the observer. It divided a quantum part of the chain (where superpositions were allowed) from the classical part (where a single actuality existed). What quantum Darwinism shows is that “shifty” can be regarded as a statement of invariance, and, thus, upgraded from a statement of contempt to a recognition of a symmetry.

Decoherence introduced the environment into the picture of quantum measurement. The original von Neumann’s chain has fanned out: Parts of the chain separate from the links that connect S via the apparatus A with the observer, and go sideways, into E. They disseminate the same information—$R_{\delta }$ copies of it—as the chain connecting S with the observer splits into sub-chains. Indeed, there may be—and often are—other observers benefiting from the information communicated by these sub-chains.

Quantum Darwinism recognizes and quantifies this fanning out of the von Neumann chain. Shifty split could be placed anywhere along the plateau of the partial information plot. The information about S that can be extracted from the chain is invariant under the shift of that split. In addition, every branch of the chain is firmly attached to the effectively classical pointer state singled out by decoherence.

## 5. Quantum Darwinism and Objective Existence: Photohalos and Extantons

We reviewed research aimed at understanding how the classical world we perceive emerges from the counterintuitive laws of quantum mechanics. It is time to take stock. Do we now understand why we perceive our undeniably quantum Universe as classical?

Quantum Darwinism holds the key to this last question. Deducing discreteness that sets the stage for the wavepacket collapse from repeatability and unitarity reveals quantum origin of quantum jumps, defines “events”, and justifies the emphasis on the Hermitian nature of observables, but this is a derivation of one of the *quantum* textbook axioms. The origin of Born’s rule in the symmetries of entanglement matters in interpreting *quantum* probabilities, but—as important as axiom (v) is for experimental predictions—its derivation from the core quantum postulates (o)–(iii) does not directly account for the familiar everyday reality.

Quantum Darwinism, by contrast, explains why our world appears classical, and why XIX century physics (physics still relevant for our everyday routine) seemed at the time like the whole story. It shows how the information is channeled from the “objects of interest’’ to us, observers, leading to consensus about what exists—to the idealization of objective classical reality.

Our everyday world comprises “systems of interest’’ endowed with the rest mass—the focus of XIX century physics—and environments that often play the role of communication channels. The quantumness of the massive “systems of interest’’ is suppressed by decoherence that is continually uploading qmemes of their states into the environments that broadcast that information. Thus, every object of interest is ensconced in an expanding halo of qmemes—information-carrying fragments of E. We perceive our world by intercepting fragments of these information-laden halos. We extract data about the systems of interest but tend to ignore the role of the halos—the means of its transmission—in the emergence of the familiar classical reality.

### 5.1. Anatomy of an Extanton

The combination of the macroscopic, massive core with the information—bearing halo defines an *extanton* (as in “extant’’). Extantons are responsible for how we perceive our Universe. The tandem—the macroscopic decohered system along with its information-laden halo—fulfills Bell’s desideratum for “beables’’ (Bell, 1975) [40][note 18]. As long as an observer relies on the halo of photons (or, possibly, other decohering environments) for the information about S, only the pointer states of the extanton core can be found out.

Extantons exist as beables should: The state of the environment is of course perturbed by agent’s measurements (e.g., photons are absorbed by our eyes). However, redundancy means that this does not alter the information about the core still available from the rest of the halo in many redundant ($R_{\delta }\gg 1$) copies.

The essence of the extanton—the only attribute classical physics cares about—is the state of its core. Information about it is available in multiple copies, and, hence, virtually unaffected by the perturbations of the halo. Indeed, as the environment continues to monitor S, qmemes of its pointer state in E multiply, and the halo continues to expand while the pointer states stay put (or evolve as a classical system would).

Extantons are not “elementary’’, but neither are atoms of even elementary particles such as protons or mesons (which are made out of quarks), not to mention planets or stars. Yet protons, mesons, atoms, molecules, planets, stars, etc., are all useful in representing and accounting for various phenomena in our Universe.

Extantons account for the classical world of our everyday experience. Quantum Darwinism explains how they come about. We can reduce extantons to their constituents, but—as other higher-level entities such as composite elementary particles, atoms, molecules, viruses, stars—extantons enable an approximate, but simple and useful description of the classical realm, of the familiar world emerging from within the quantum substrate.

Part of that simplification that led to the Newtonian view of the world is our habit of ignoring halos, carriers of the information, and focusing on the extanton cores. These cores are the “objects of interest’’. Many are subject to Newtonian dynamics. It was convenient to ignore the means by which the information about the cores is delivered, and just focus on the physics of the cores.

Inseparable bond of the core with the halo is responsible for what we perceive. However, conditioned by Newtonian physics we tend to ignore the presence and the role of the halos, and just use the data they deliver to find out about the cores. In the end, only these cores count as elements of our everyday reality. Yet, it is the halos that bear responsibility for the suppression of quantum superpositions and for our perception of the familiar—robust and objective—classical world.

The difficulty with the quantum-classical correspondence arises when the role of the halo is overlooked and the extanton core is treated as if it were isolated, and, therefore, subject to unitary evolution. Core is indeed quantum (as is everything else in our Universe) but it is by no means isolated. Therefore, one cannot expect the core of an extanton to follow a unitary quantum evolution. Nevertheless, whenever Newtonian dynamics is a good approximation)its classical evolution can be approximately reversible.

#### 5.1.1. Extantons and “The Classical”

No one has ever worried about interpretation of classical physics. This is because classical states were thought to be real—to exist independently of what was known about them. There was no need to be concerned about the effect of information acquisition. Even though measurement could perturb a system, that perturbation could always be made as small as required.

The main reason for the interpretational discomfort with quantum theory is our faith in the underlying objective reality. It is based on our everyday experience that leads us to believe the world we inhabit exists independently of the information we (or other agents) have.

We owe this confidence in objective classical reality to the fact that we are immersed in the extanton halos and inundated with the information about their cores. As a result of this overload with “free” data we have grown up believing that we can examine systems and determine their states without perturbing them—classical measurement would just update the record (change the state of the apparatus or of the observer’s memory) without ‘backreaction’. In the classical setting information about a system is obtained, but its state or its evolution is untouched.

One might call this a *myth of immaculate perception*. It asserts a unidirectional information flow (hence, flow of influence, from the measured system to the observer) that reveals the state of the system but leaves it unaffected. This seems to defy the spirit of Newton’s principle that action elicits reaction.

Quantum Darwinism accounts for the origin of this myth. All macroscopic objects telegraph their pointer states via their decohering environments. They are enveloped by information halos—by the environment with multiple records of their decoherence-resistant states. Our everyday world does not consist of isolated systems. Rather it is defined by the extantons consisting of a macroscopic object enveloped in and heralded by its halo, imprinted on the environment.

There is a great variety of extantons. They all consist of the core and the halo that “knows’’ the state of the core, and may be persuaded to share that information with observers. Planets are extanton cores and so are the grains of photographic emulsion. Any object one can see is likely an extanton core, its state communicated to us by photons—by its photohalo. Some extantons (like planets) follow approximately reversible dynamics. Others (like grains of photographic emulsion or Brownian particles) are heavily damped or embedded in an immobilizing medium competing with the photohalo in monitoring of the core (and, hence, in decohering it). Every apparatus pointer is an extanton core.

Observers intercept fractions of the halo to gain information about the core. Direct measurement of the core is in principle also possible, but only measurements of the pointer observable would lead to predictable results: Other outcomes—superpositions of pointer states—are quickly invalidated by decoherence. Thus, except for laboratory settings (where decoherence can be kept at bay by a near-perfect isolation) the only observables with predictive value are the pointer observables readily available from the halo.

Photohalo is the main channel through which observers find out the state of the core. The information available from the photohalo is usually limited to the data that are in effect macroscopic (therefore, easily imprinted on the halo, such as the location of the dielectric sphere discussed earlier).

#### 5.1.2. Photohalos, Photoextantons, and Information Detached from Existence

Photons play a preeminent role as information carriers. Every object we know is bathed in a radiation environment. Each is surrounded by an information halo, its photons disseminating qmemes with the speed of light. We eavesdrop on these photohalos, intercepting small fractions that nevertheless reveal the state of the core.

Photoextantons are the family of extantons that advertise the state of their core via their photohalos. One can consider extantons where photohalo is the only relevant environment. There are at least two good reasons for this. To begin with, our eyesight is responsible for most of the information we obtain, so it is of interest to consider photoextantons. Perhaps more importantly, there is a sense in which photoextantons come close to the ideal—one might say Platonic ideal—of the separation of existence and information detached from existence.

Information (in the form of the photohalo) has—apart from the dramatic consequences of decoherence—only a negligible effect on the evolution of the cores. Photohalo detaches from the core and runs off seemingly without any dynamical consequences while the core can evolve in approximate accord with Newton’s laws. Questions such as “Does the moon exist when no one’s looking?’’ are motivated by the illusion that one can apply quantum theory to the isolated cores of extantons and recover classical reality.

Quantum states are *epiontic*. Quantum physics has eliminated separation of existence and information. Nevertheless, extanton structure restores this separation (with suitable caveats): The core exists (it persists in a pointer state). And the information about it is continually detached and propagates as the photohalo.

Everyday practice of quantum theory has inherited from classical physics the habit of dealing with isolated systems. Yet, we can never encounter isolated macroscopic extanton cores. Trying to understand cores of extantons as if they were isolated is the cause of the “measurement problem’’. Decoherence and einselection were the first steps towards its resolution, towards the understanding why our Universe looks classical to us. Quantum Darwinism provides the complete answer, and extantons are its embodiment.

Properties of systems that reside in the extanton cores and give rise to the qmemes in E are untouched by the measurements on the halo, yet they reveal the state of the core. This is guaranteed by the theorems we have already discussed that also imply (depending on specific assumptions) uniqueness or near uniqueness of the states that can be repeatedly imprinted on the halo, and, therefore, deduced from the environment fragments: The optimal strategy for the agent is to find out what the halo redundantly advertises—that is, pointer states of the extanton core—and then use that information to deduce other observables of interest.

#### 5.1.3. Photohalos and the Quantum Origins of Irreversibility

Information in classical, Newtonian physics was immaterial—it was about physics, but it was not a part of physics. Its acquisition did not influence states or the *reversible* dynamics of classical systems. Trajectories in phase space were unaffected by the information transfer. In particular, classical dynamics involved in the measurement process could be reversed even when the outcome of the measurement was recorded. By contrast (and as we have seen in Section 2) copying of a quantum measurement outcome precludes its reversal.

How could an effectively classical realm of our fundamentally quantum Universe follow so respectfully laws that are at odds with the epiontic nature of quantum states? What detaches information about the states from these states so that they can follow classical reversible dynamics, oblivious to what is known about them?

Platonic ideal of the separation of existence and the detached information about what exists would be fulfilled if the information in the photohalo would have no effect on the energy or momentum of the core, but would still decohere it efficiently enough to assure effective classicality of its state. Photohalos exert no (easily) noticeable influence on trajectories of the pointer states of the core. However, cores can interact indirectly with other cores (e.g., via gravitational forces) or scatter elastically (billiard balls) or inelastically. Consequences of such interactions on the motion or the state of the core or on its properties (e.g., merger of the cores in inelastic scattering) will be reflected in the photohalo.

When core is sufficiently massive, emission or scattering of photons has negligible effect on its momentum (although photohalo will still decohere its quantum state very efficiently). How heavy the core should be to make such an approximation accurate depends on the energy of the emitted or scattered photons. Planets are clearly sufficiently massive, and so are billiard balls or even dust grains [21,23,174]. Depending on detailed criteria, one might even classify fullerenes decohered by radiation they emit [180] as photoextantons.

Irreversibility associated with the wavepacket collapse is a natural consequence of the photohalo. As soon as a minute fraction of the photohalo escapes, the ‘reduction of the state vector’ becomes irreversible, since the escaped photons are gone for good, And, as we have seen in Section 2 (and unlike in the classical realm) presence of even a single record of the measurement outcome—of the pointer state—anywhere in the Universe precludes reversal of the evolution that would bring back superpositions. Thus, the “in principle” reversibility of the equations responsible for inscribing, e.g., $10^{14}$ copies of the location of the dielectric sphere on the sunlight per second we have discussed earlier is trumped by relativistic causality: Reversal becomes impossible in principle as soon as even one such copy runs off to infinity with the speed of light, never to be turned back[note 19].

The information about the classical states (of the cores) becomes detached: Separation of the states of the cores from the information about them is how extantons account for the emergence of the objective classical reality. This mechanism is also the uniquely quantum origin of irreversibility in our Universe.

Photohalo may not be the only environment. However, in our Universe interactions depend on distances. Therefore, localized pointer states imprinted on the photohalo are left intact also by the other environments that may be contributing to decoherence but—like air—are not as useful as witnesses.

### 5.2. Quantum Darwinism and the Existential Interpretation

States in classical physics were “real”: Their *objective existence* was established operationally—they could be found out by an initially ignorant observer without getting perturbed in the measurement process. Hence, they existed independently of what was known about them.

Information was, by contrast, “not real”. This was suggested by the immunity of classical states to measurements, and by the fact that, in Newtonian dynamics, information about an evolving system was of no consequence for its evolution. Information was what the observer knew subjectively, a mere shadow of the real state, irrelevant for physics.

This dismissive view of information ran into problems when the classical Universe of Newton confronted thermodynamics. Clash of these two paradigms led to Maxwell’s demon, and is echoed in the discussions of the origins of the arrow of time.

The specter of information was and still is haunting physics. The seemingly unphysical record state was beginning to play a role reserved for the real state!

Quantum states are epiontic. They combine information and existence and (like photons or electrons that can be wave-like or particle-like) they can reveal one or the other aspect of their nature depending on circumstances.

We have just seen how, in the quantum setting, in extantons, existence and information about existence intertwine. The state known to observers is defined and made objective by what is known about it—by the information observers can access. “It from bit” comes to mind (Wheeler, 1990) [196].

The main new ingredient is the dramatic upgrade of the role of the environment. It has information—multiple records of S—and is willing to reveal it. It acquires information about the system while causing decoherence and einselection, but—and this is the upgrade—it acts as a communication channel.

In classical Newtonian settings information might have been dismissed as unphysical as it had no significance for dynamics. But *information is physical* (Landauer, 1991) [197]. Moreover, *there is no information without representation*—information must reside somewhere (e.g., in the photohalos). And the presence and availability of such evidence (objective because there are plenty of records of the pointer state) has its legal consequences.

The role of E in quantum Darwinism is not that of an innocent bystander, who simply reports what has happened. Rather, the environment is an accomplice in the “crime” of selecting and transforming fragile epiontic quantum states into robust, objectively existing classical states. Objective existence has its price: Environment—induced decoherence invalidates quantum principle of superposition, leading to einselection—to censoring the Hilbert space. Information transfer associated with decoherence selects preferred pointer states, and banishes their superpositions.

Moreover, testimony offered by the environment is biased—it depends on how (through what observable) E monitors S. Fragments of the environment—qmemes carried by the extanton halo—can reveal information only about the very same pointer states E has helped einselect.

Operational criterion for objective existence is the ability to find out a state without disturbing it. According to this operational definition, pointer states exist in more or less the same way their classical counterparts did: They can be found out without getting perturbed by anyone who examines one of the multiple copies of the record of S “on display” in the environment.

### 5.3. From Quantum Core Postulates to Objective Classical Reality

In search for the relation between quantum formalism and the real world we have weaved together several ideas that are very quantum to arrive at the *existential interpretation*. Its essence is the operational definition of objective existence of physical states: *To exist, a state must, at the very least, persist or evolve predictably in spite of the immersion of the system in its environment.* Predictability is the key to einselection, but persistence is only a necessary condition.

*Objective existence requires more: It should be possible to find out a state without perturbing it—without threatening its existence.* When that last desideratum is met, many observers will be able to reach consensus, a well—motivated practical criterion of objective existence.

Let us briefly recapitulate how objective existence arises in the quantum setting: We started with axioms (i) and (ii) that sum up mathematics of quantum theory: They impose the quantum principle of superposition, and demand unitarity, but make no connection with the “real world”. Addition of predictability (via the repeatability postulate (iii), the only uncontroversial measurement axiom), and recognition that our Universe consists of systems (axiom (o)) leads to *preferred pointer states*. This is a new insight into the quantum origin of quantum discreteness. Predictability is the “root cause” of the wavepacket collapse and quantum jumps. It also justifies Hermitian nature of quantum observables and explains breaking of the unitary symmetry, the crux of the collapse axiom (iv).

Our next task was to understand *the origin of probabilities and Born’s rule*, axiom (v). We have done this without appealing to decoherence (as this would have courted circularity in the derivation). Nevertheless, decoherence—inspired view is reflected in the envariant approach: To assign probabilities to pointer states we first had to show how to get decoherence without using tools of decoherence such as reduced density matrices and demonstrate that relative phases between outcomes do not matter.

Envariance provides the answer, but—strictly speaking—only for Schmidt states. We take this to mean that usual rules of the probability calculus will strictly hold for pointer states only after their superposition has been thoroughly decohered. Pointer states—Schmidt states coincidence is expected to be very good indeed: Probabilities one has in mind ultimately refer to pointer states of measuring or recording devices. These are usually macroscopic, so their interaction with the environment will quickly align Schmidt basis with pointer states. Born’s rule (with all the consequences for the frequencies of events) follows.

Probabilities derived from envariance are objective: They reflect an objective—and experimentally testable—symmetry of the global state (usually involving the measured system S, the apparatus A, and its environment E). Before interacting with A or its E observer does not know the outcome, but will know the set of pointer states—the menu of possibilities. This ignorance reflects objective symmetries of the global state of SAE that lead to Born’s rule.

Last question was the origin of *objective existence* in the quantum world: How can we find states of systems we encounter in our everyday experience without redefining them by our measurements?

We started by noting that in contrast to fragile arbitrary superpositions in the Hilbert space of the system, pointer states are robust.

Crucially, in the real world observers find out pointer states by letting natural selection take its course: Pointer states are the “robust species”, adapted to their environments. They survive intact through such “environmental monitoring”. More importantly, multiple records about S are deposited in E. They record pointer states, which are the “fittest”—they can survive the process of copying and so the information about them can multiply.

There is an extent to which “it had to be so”: In order to make one more copy one needs to preserve the original. However, there is a more subtle part of this relation between decoherence, einselection, and quantum Darwinism. Hamiltonians of interaction that allow for copying of certain observables necessarily leave them unperturbed. This conspiracy was noticed early: It is the basis of the commutation criterion for pointer observables (Zurek, 1981; 1982) [24,25]: When $H_{SE}$ is a function of some observable Λ of the system, it will also necessarily commute with it, $[H_{SE},$Λ]=0.

### 5.4. Extantons and the Existential Interpretation

Existential interpretation of quantum theory assigns “relatively objective existence” [6,7,36,198]—key to effective classicality—to widely broadcast quantum states. Objective existence is relative to the redundant records about what persists and (in that sense) exists—evidence of the pointer states imprinted on the environment.

Existential interpretation is obviously consistent with the relative states interpretation: Redundancy of the records disseminated throughout the environment suggests a natural definition of branches that are classical in the sense that an observer can find out a branch with indirect measurements and stay on it, rather than “cut off the branch he is sitting on” with a direct measurement. This is more than einselection, and much more than decoherence, although the key ingredient—environment—is still key, and the key criterion is “survival of the fittest”— immunity of the pointer states to monitoring by E reflected also in the predictability sieve.

The role of E is, however, upgraded from a passive “quantum information dump” to a communication channel. Information deposited in E in the process of decoherence is not lost. Rather, it is stored there, in multiple copies, often—but not always—for all to see. The emphasis on information theoretic significance of quantum states cuts both ways: Environment—as—a—witness paradigm supplies operational definition of objective existence, and shows why and how pointer states can be found out without getting disrupted. However, it also shows that objective existence is not an intrinsic attribute of quantum states, but that it arises along with—and as a result of—information transfer from S to E.

Extantons combine the source of information (extanton core) with the means of its transmission (halo, often consisting of photons). Extanton is an extant composite entity with the object of interest in its core that is decohering and imprinting qmemes on the information-laden halo, part of the environment that pointer states of the core. Information about them is disseminated by the halos throughout the Universe.

Photohalos are especially efficient in such “advertising”. Fragments of the halo intercepted by the observers inform them about the state of its core. Extanton cores persist as classical states would—independent of what is known about them. While they are not fundamental, they are fundamentally important for our perception of the quantum Universe we inhabit as a classical world. Extantons fulfill John Bell’s desiderata for “beables”.

### 5.5. Decoherence and Information Processing

Decoherence affects record keeping and information processing hardware (and, hence, information retention and processing abilities) of observers. Therefore, it is relevant for our consciousness, as agents’ consciousness presumably reflects states and processes of their neural networks. As was already noted some time ago by Tegmark (2000) [65], individual neurons decohere on a timescale very short compared to, e.g., the “clock time” on which the human brain operates or other relevant timescales. Moreover, even if somehow one could initiate our information processing hardware in a superposition of its pointer states (which would open up a possibility of being conscious of superpositions), it would decohere almost instantly. The same argument applies to the present day classical computers. Thus, even if information that is explicitly quantum (that is, involves superpositions or entanglement) was inscribed in computer memory, it would decohere and (at best) become classical. (More likely, it would become random nonsense.)

It is a separate and intriguing question whether a robot equipped with a quantum computer could “do better” and perceive quantumness we are bound to miss. We note, however, that the relevance of this question for the subject at hand—i.e., why does our quantum Universe appear to us as a classical world devoid of quantum weirdness—is at best marginal. After all, as already noted, rapid decoherence in the neural networks of our brains precludes quantum information processing.

Moreover, if such a robot relied (as we do) on the fragments of the environment for the information about the system of interest, it could access only the same information we can access. This information is redundant—hence, classical (see Girolami et al., 2022 [142])—and quantum information processing capabilities would not help; only pointer states can be accessed through this communication channel.

Existential interpretation—as defined originally (e.g., Zurek, 1993 [6])—relied primarily on decoherence. Decoherence of systems immersed in their environments leads to einselection: Only preferred pointer states of a system are stable, so only they can persist. Moreover, when both systems of interest and means of perception and information storage are subject to decoherence, only the correlations of the pointer states of the measured systems with the corresponding pointer states of agent’s memory that store the outcomes are stable—only such einselected correlations can persist.

Decoherence, one might say, “strikes twice”: It selects preferred states of the systems, thus defining what can persist, hence, exist. It also limits correlations that can persist, so that the observer’s memory or the apparatus pointer will only preserve correlations with the einselected states of the measured system when they are recorded in the einselected memory states.

This is really the already familiar discussion of the quantum measurement problem with one additional twist: Not just the apparatus, but also the systems of everyday interest to observers are subject of decoherence. Thus, the post-measurement correlations (when investigated using discord) must now be “classical-classical” using terminology of Piani, Horodecki, Horodecki, 2008 [135], while in quantum measurements only the apparatus side was guaranteed to be einselected (hence, certifiably classical).

### 5.6. Quantum Darwinism and “Life as We Know It”

Quantum Darwinism has a Darwinian name, but how Darwinian is it really? This is a vague question about nomenclature, and we shall pass it by. A related and more pointed question is: Does quantum Darwinism have any significance for the evolution of living organisms and for their survival? There are then two even more focused questions one can pose: Are the processes of imprinting information and passing it on (that are central to quantum Darwinism) relevant for natural evolution? Moreover, do living organisms take advantage of the proliferation of information about pointer states?

We note at the outset that natural selection and Darwinian evolution require multiple redundant records (e.g., as DNA). Redundant records imply (as we have seen already in Section 2) preferred set of states—preferred einselected classical basis. Therefore, natural selection and Darwinian evolution take place within a classical post-decoherence realm.

Darwinian evolution depends on the (nearly perfect) preservation of information and its propagation. DNA is disseminated by various means, but DNA molecules are thoroughly decohered carriers of information about an even more decohered parent organism.

The goals and the nature of natural evolution are somewhat different but the transfer of actionable information (see Section 2, Ref. [43]) is the essence of both the “original” and quantum Darwinism. Quantumness in a DNA “blueprint” is long gone, suppressed by decoherence, but the correlation between the pointer state and individual fragment F of E is also classical (as quantum discord is suppressed by surplus decoherence).

The key difference with quantum Darwinism appears to be the presence of the feedback in Darwinian evolution: It depends on the ability of the DNA to reproduce a copy of their progenitor so it can in turn produce more DNA. Evolutionary success therefore is judged not by redundancy in amplifying information in a single episode but through multiple generations of the organisms and their DNA memes. Iteration through multiple generations allows for variation and for natural selection. Fitness is still evaluated based on the ability to amplify, but in an iterative process.

The copies of DNA are imperfect—there are mutations. They are not deliberate—error correction guards against them–but they still happen. That imperfection in combination with the feedback is what allows for gradual change of the DNA and for the adaptation of the organisms driven by the natural selection and survival of the fittest.

By contrast, amplification and dissemination of quantum information appear to be enough to attain the goal of quantum Darwinism: They suffice to account for the emergence of objective classical reality. The ultimate fate of qmeme carriers such as photons is—at first sight—unimportant, and so, one might think, is the fate of the information: Once the state of the system or an event are made “objective”, they are an element of reality, and that is it.

Or is it? After all, following their detection photons (and other carriers of information accessible through other senses that also deliver data about the Universe) are no longer needed. Nevertheless, there is the next generation of records—in the apparatus, in the retina, or in the brain of the observer. Such information repositories inherit the memes—the essence of the information detected by the senses.

That leads to further consequences: Agents (living organisms) react to data. Their actions (hence, the state of the part of the Universe affected by them) depend on what they have perceived. In the evolutionary context such actions are taken to optimize their chances of survival. Hence, evolutionary success in the original Darwinian sense depends on the ability to acquire and process information about extantons that is obtained through quantum Darwinian processes that starts with decoherence and proliferation of qmemes.

Redundancy facilitates accessibility and explains the objective nature of events, and, hence, emergence of the objective classical reality. Consequences of such events influence multiple generations of their records and have implications for their recipients: After all, actions taken by the living organisms are based on perceptions—on what was recorded. Knowledge of events that take place is essential for survival. This is feedback. It does not necessarily require conscious decisions (e.g., sunflowers following the sun are taking advantage of such feedback), but it allows, and indeed, calls for, adaptation. Feedback is also what allows for learning from experience. Thus, while natural selection and Darwinism can be analyzed without any reference to quantum goings on, it does involve steps that depend on decoherence and proliferation of information.

#### Seeing Is Believing

Our senses did not evolve to test and verify quantum theory. Rather, they evolved through natural selection where survival of the fittest played a decisive role. When there is nothing to be gained from prediction, there is no evolutionary reason for perception. Only classical states that pass through the predictability sieve and deposit redundant easily accessible records in their environment are robust and easy to access.

Quantum Darwinian requirement of redundancy appears to be built into our senses, and, in particular, into our eyesight: The wiring of the nerves that pass on the signals from the rods in the eye—cells that detect light when illumination is marginal, and that appear sensitive to individual photons (Nam et al., 2014) [199]—tends to dismiss cases when fewer than ∼7 neighboring rods fire simultaneously (Rieke and Baylor, 1998) [200]. Thus, while there is evidence that even individual photons can be (occasionally and unreliably) detected by humans (Tinsley et al., 2016) [201], redundancy (more than one photon) is needed to pass the signal onto the brain.

This makes evolutionary sense—rods can misfire, so such built-in veto threshold suppresses false alarms. Frogs and toads have apparently lower veto thresholds, possibly because they are cold-blooded, so they may not need to contend with as much noise, as thermal excitation of rods appears to be the main source of “false positives”.

Quantum Darwinism relies on repeatability. As observers perceive outcomes of measurements indirectly—e.g., by looking at the pointer of the apparatus or at the photographic plate that was used in a double-slit experiment—they will depend, for their perceptions, on redundant copies of photons that are scattered from (or absorbed by) the apparatus pointer or the blackened grains of photographic emulsion. Thus, repeatability is not just a convenient assumption of a theorist: This hallmark of quantum Darwinism is built into our senses. And—as we have seen in Section 2—the discreteness of the possible measurement outcomes—possible perceptions —follows from the distinguishability of the preferred states that can be redundantly recorded in the environment [42,43].

What we are conscious of is then based on redundant evidence. Quantum Darwinist update to the existential interpretation is to demand that states exist providing that one can acquire redundant evidence about them indirectly, from the environment. This of course presumes stability in spite of decoherence (so there is no conflict with the existential interpretation that was originally formulated primarily on the basis of decoherence [6,36]) but the threshold for the state to objectively exist is nevertheless raised.

### 5.7. Bohr, Everett, and Wheeler

This paper has largely avoided issues of interpretation, focusing instead on consequences of quantum theory that are “interpretation independent”, but may constrain interpretational options. We have been led by quantum formalism to our conclusions, but they are largely beyond interpretational disputes. Our “existential interpretation” is in that sense not an interpretation—it simply points out the consequences of quantum formalism and some additional rudimentary assumptions. It recognizes that quantum states are epiontic: Like photons or electrons that can act as waves or particles, quantum states can exhibit epistemic or ontic side of their nature, depending on circumstances. In contrast to Bohr (who regarded them as purely epistemic) of Everett (who thought the universal state vector was ontic) they can perform either as robust elements of reality or as information carriers. These two roles are complementary: both are essential for extantons.

It is nevertheless useful to see how the two best known interpretations of quantum theory—Bohr’s “Copenhagen Interpretation” (CI) and Everett’s “Relative State Interpretation” (RSI) fit within the constraints that we have derived above by acknowledging the paramount role of the environment. To anticipate the conclusion, we quote John Archibald Wheeler (1957) [12], who—comparing CI with RSI—wrote: “*(1) The conceptual scheme of “relative state” quantum mechanics is completely different from the conceptual scheme of the conventional “external observation” form of quantum mechanics and (2) The conclusions from the new treatment correspond completely in familiar cases to the conclusions from the usual analysis*”.

Bohr insisted on preexistence of the classical domain of the Universe to render outcomes of quantum measurements firm and objective. Quantum Darwinism accomplishes that goal: Decoherence takes away quantumness of the system, but a system that is not quantum need not be immediately classical: Objective nature of events and, above all, of extantons arises as a result of redundancy. Consensus about what happened and what exists is reached only in presence of large redundancy. Large redundancy yields a very good approximation of “the classical”, like finite many-body systems that have a critical point marking a phase transition which is, strictly speaking, precisely defined only in the infinite size limit. Indeed, Quantum Darwinism might be regarded as a purely quantum implementation of the “irreversible act of amplification” that was such an important element of CI.

Physical significance of a quantum state in CI was purely epistemic (Bohr, 1928 [8]; Peres, 1993 [202]; Fuchs and Peres, 2000 [203]; Fuchs and Schack, 2013 [204]): Quantum states were only carriers of information—they correlated outcomes of measurements. Only the classical part of the Universe existed in the sense we are used to.

In the account we have given here there are really several different sorts of states. There are pure states—vectors in the Hilbert space. However, there are also objectively unknown states defined by the “Facts 1–3” of Section 3. They describe a subsystem and derive from the pure state of the whole using envariance, the symmetries of entanglement. And there are states defined through the spectral decomposition of a quantum operator. Last not least, there are decoherence-resistant pointer states that retain correlations and allow for prediction based on indirect measurements, as they are best known to and widely advertised by the environment.

In contrast to CI that split the Universe into only two domains—quantum and classical—we have seen that classicality is a matter of degree, and a matter of a criterion. For example, objectivity (which is in a sense the strongest criterion) is attained only in the limit of large redundancy. It is clear why this is a good approximation in the case of macroscopic systems. However, it is also clear that there are intermediate stages on the way from quantum to classical, and that a system can be no longer quantum but be still far away from classical objective existence.

There are two key ideas in Everett’s writings. The first one is to let quantum theory dictate its own interpretation. We took this “let quantum be quantum” point very seriously. The second message (that often dominates in popular accounts) is the Many Worlds mythology. In contrast “let quantum be quantum” it is less clear what it means, so—in the opinion of this author—there is less reason to take it at face value.

It is encouraging for the relative states point of view that the long - standing problem of the origin of probabilities has an elegant solution that is very much “relative state” in the spirit. We have relied on symmetries of entangled states. This allowed us to derive objective probabilities for individual events. We note that this is the first such *objective* derivation of probabilities not just in the quantum setting, but also in the history of the concept of probability.

Envariant derivation of Born’s rule is based on entanglement (which is at the heart of the relative states approach). We have not followed either proposals that appeal directly to invariant measures on the Hilbert space [10,11], or attempts to derive Born’s rule from frequencies by counting many worlds branches (Everett, 1957 [11]; DeWitt, 1970 [14], 1971 [15]; Graham 1973 [72], Geroch, 1984 [73]): As noted by DeWitt (1971) [15] and Kent (1990) [38], Everett’s appeal to invariance of Hilbert space measures makes no contact with physics, and makes less physical sense than the mathematically rigorous proof of Gleason (1957) [84]. In addition, frequentist derivations are circular—in addition to counting branches they implicitly use Born’s rule to dismiss “maverick universes”.

### 5.8. Closing Remarks

The emergence of the classical world from within our quantum Universe is a difficult problem. The traditional (and still occasionally encountered) expectation—that it will be somehow resolved by a single new idea—did not pan out. Rather, several interdependent new insights were needed to account for quantum jumps, for the appearance of the collapse, for the preferred pointer states, for probabilities, and for the perception of the objective reality—for all the familiar ingredients of ‘the classical’.

Our strategy was to avoid purely interpretational issues and to focus instead on technical questions. They can often be answered in a definitive manner. In this way, we have gained new insights into selection of preferred pointer states that go beyond decoherence, found out how probabilities arise from entanglement, and discovered how objectivity follows from redundancy.

All of that fits well with the relative states point of view and with a similar although less Everettian approach of, e.g., Rovelli [205]. There are also questions that are related to the technical developments we have discussed but are, at present, less definite—less technical—in nature. We signal some of them here.

The first point concerns the nature of quantum states, and its implications for the interpretation. One might regard states as purely epistemic (as did Bohr) or attribute to them “existence”. Technical results described above suggest that truth lies somewhere between these two extremes, and these two aspects are complementary in the sense of Bohr. It is therefore doubtful whether one is forced to attribute “reality” to all of the branches of the universal state vector. Indeed, such a view combines a very quantum idea of a state in the Hilbert space with a very classical literal ontic interpretation of that concept.

These two views of the universal state vector are incompatible. As we have emphasized, an unknown quantum state cannot be found out. It can acquire objective existence only by “advertising itself” in the environment[note 20]. This is obviously impossible for the universal state vector as the Universe has no environment.

The insistence on the absolute existence of the universal state vector as an indispensable prerequisite in interpreting quantum theory brings to mind the insistence on the absolute time and space. They seemed indispensable since Newton, yet both became relative and observer-dependent in special relativity. The absolute universal state vector may be—like the Newtonian absolute space and time, or for that matter, like isolated systems of classical physics—an idealization that is untenable in the quantum realm.

As noted in Section 2.6, observations reset the state of the Universe—they re-adjust initial conditions relevant for the future of the observer who has the record of their outcome. The rest of the state vector becomes unreachable. Thus, the relative state reading of Everett rests on a safer foundation than the “Many Worlds” alternative, which is—as Wheeler (1957) [12] pointed out—compatible with the consequences of Bohr’s views. There is nothing in the relative state interpretation that would elevate all the branches—especially the ones that “did not happen” to the observer—to the same ontological status as the one that is consistent with the observer’s perceptions.

Objective existence can be acquired (via quantum Darwinism) only by a relatively small fraction of all degrees of freedom within the quantum Universe: The rest is needed to “keep records”. Extantons (with a classical core and a vast information carrying halo) are a good illustration. Clearly, there is only a limited (if large) memory space available for this at any time. This limitation on the total memory available means that not all quantum states that exist or quantum events that happen now “really happen” in the sense of the existential interpretation: Only a small fraction of what occurs will be still available from the records in the future. So the finite memory capacity of the Universe implies indefiniteness of the present and impermanence of the past.

To sum it up, one can extend John Wheeler’s dictum “the past exists only insofar as it is recorded in the present”. This is one of the topics that could have been discussed in this review, but was not. Fortunately, Ref. [210] provides an introduction to the quantum Darwinian view of the emergence of objective past in our Universe.

Consensus can be reached about objective histories, a more selective set than histories that are just consistent. This may help settle the so-called class selection problem—that is, selecting candidates for physically relevant histories from among the set of all consistent histories—which is one of the central unresolved issues of the consistent histories interpretation of quantum physics (see Griffiths, 1984 [118]; 2002 [119]; Gell-Mann and Hartle, 1990 [120]; 1993 [121]; 2012 [211]; Omnès, 1992 [122]; Griffiths and Omnès, 1999 [123]; Halliwell, 1999 [212] for a selection of points of view on this subject).

As long as we are discussing subjects that are related, but beyond the scope of this review, we should mention the so-called “strong quantum Darwinism” and “spectrum broadcast structures”. They share with quantum Darwinism the main idea—that the objectively existing states are reproduced in many copies and can be independently accessed without a disturbance. They differ in the details of its implementation (e.g., by invoking different measures of what and how much the environment fragments know about the system). Fortunately, there are papers by Horodecki, Korbicz, and Horodecki (2015) [213], Le and Olaya-Castro (2018; 2020) [214,215], and there is a recent review by Korbicz (2021) [216]. These references discuss ‘strong quantum Darwinism” and “spectrum broadcast structures” and provide useful entries into the relevant literature.

Some of the quantum information theoretic tools used to implement criteria for objective existence have also been ‘beyond the scope’. In particular, quantum Chernoff bound has been successfully used in models [146], and we expect it to be useful in the future. Fortunately, a recent paper by Zwolak (2022) [141] discusses how quantum Chernoff bound can be used to analyze quantum Darwinism (e.g., in the c-maybe model [131]).

These results confirm that there is no need for a unique best quantum information-theoretic tool to study quantum Darwinism. We have already seen that Shannon and von Neumann mutual entropy, and various half-way quantities including Holevo χ (as well as Chernoff bound) lead to useful estimates of redundancy.

In presence of sufficiently large redundancy the precise number of records in the environment does not matter: As long as redundancy is large, emergence of the consensus—hence, objective classical reality—will be confirmed by perceptions of observers.

All of our conclusions followed from the core quantum postulates (o)–(iii) and the insight that quantum states are epiontic. These two aspects of their nature are complementary (like the wave and particle traits of photons and electrons). They both play a role in the emergence of objective classical reality with information and existence seemingly separated – the state of affairs we are accustomed to in our everyday experience.

In the field known for divergent views and lively discussions it is too early to expect consensus on which of the mysteries of the quantum-classical correspondence have been explained, but deducing inevitability of the discreteness and of quantum jumps, a simple and physically transparent derivation of Born’s rule, and—above all—accounting for the emergence of objective existence from the fragile quantum states mark a significant progress. Moreover, there are no obvious obstacles in pursuing the program outlined here to provide an even fuller account of the interrelation of the epistemic and ontic aspects of quantum states, or to incorporate evolving quantum states—thus exploring emergence of objective histories—within the framework of quantum Darwinism.

## Figures and Tables

**Figure 1 entropy-24-01520-f001:**
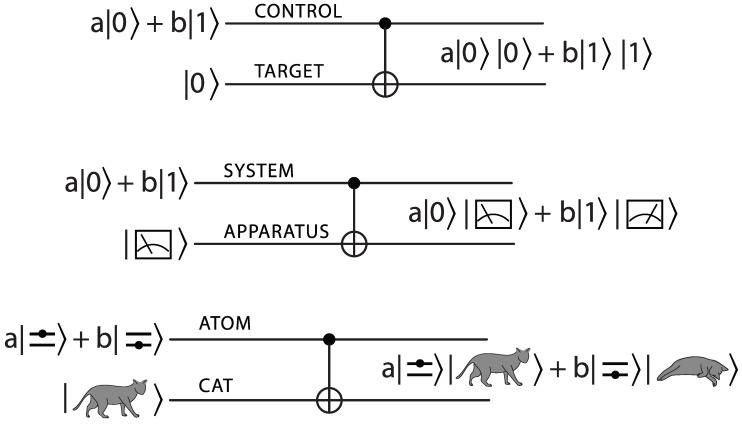
**Controlled-not, measurement, and Schrödinger’s cat**: We expect this figure to be self-explanatory. It is included primarily to establish the nomenclature (i.e., “control” and “target”), to illustrate Equation (1.1), and to emphasize the parallels be between the three situations illustrated above.

**Figure 2 entropy-24-01520-f002:**
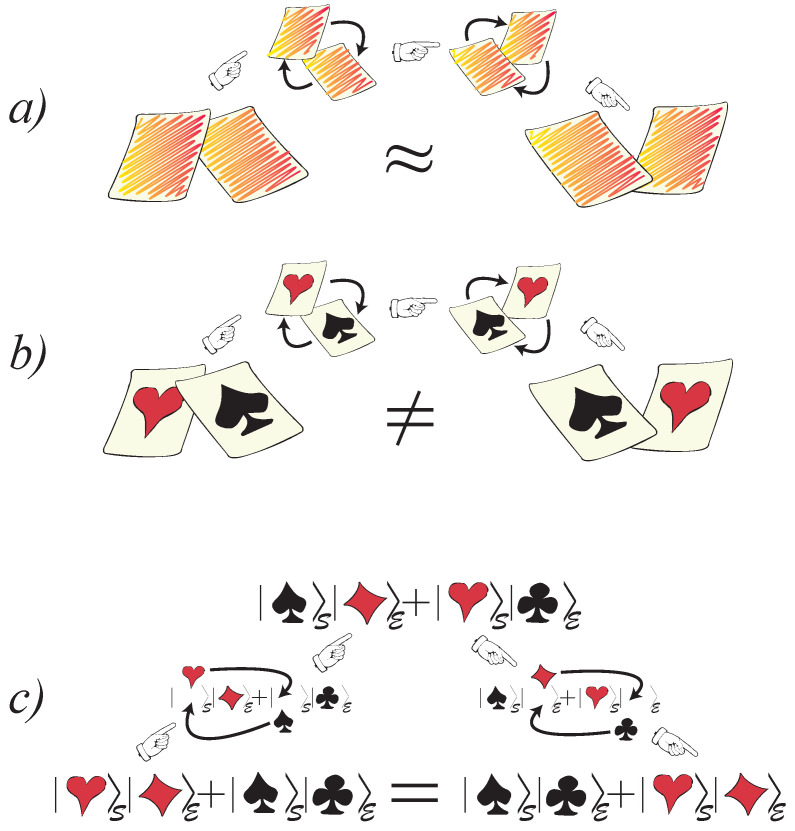
**Probabilities and symmetry**: (**a**) Laplace (1820) [69] appealed to subjective invariance (associated with ‘indifference’ based on observer’s ignorance of the real physical state) to define probability via *principle of equal likelihood*: When ignorance means observer is indifferent to swapping (e.g., of cards), alternative events should be considered equiprobable. So, for the cards above, subjective probability $p_{♠}=\frac{1}{2}$ would be inferred by an observer who does not know their face value, but knows that one (and only one) of the two cards is a spade. (**b**) Nevertheless, the real physical state of the system is independent from what is known about it. Moreover, the order of the cards is altered by the swap—it is not ‘indifferent’—illustrating subjective nature of Laplace’s approach. Subjectivity of equal likelihood probabilities poses foundational problems for, e.g., statistical physics. This led to an alternative definition employing relative frequency—an objective property (albeit of a fictitious—and, hence, subjective—infinite ensemble). (**c**) Quantum theory allows for an objective definition of probabilities based on *a perfectly known state* of a composite system and on the symmetries of entanglement [7,35,48,77,78]. When two systems (S and E) are maximally entangled (i.e., Schmidt coefficients differ only by phases, as in the Bell state above), a swap $|♠\rangle \langle ♡|+|♡\rangle \langle ♠|$ in S can be undone by ‘counterswap’ $|♣\rangle \langle ♦|+|♦\rangle \langle ♣|$ in E. Certainty about entangled state of the whole in combination with the symmetry between $|♠\rangle$ and $|♡\rangle$—certainty that $p_{♠}=p_{♡}$—means that $p_{♠}=p_{♡}=\frac{1}{2}$, Equiprobability follows from the objective symmetry of entanglement. This *en*tanglement—assisted in*variance* (*envariance*) also establishes decoherence of Schmidt states, allowing for additivity of probabilities of the effectively classical pointer states. Probabilities derived through envariance quantify indeterminacy of the state of S alone given the global entangled state of SE.

**Figure 3 entropy-24-01520-f003:**
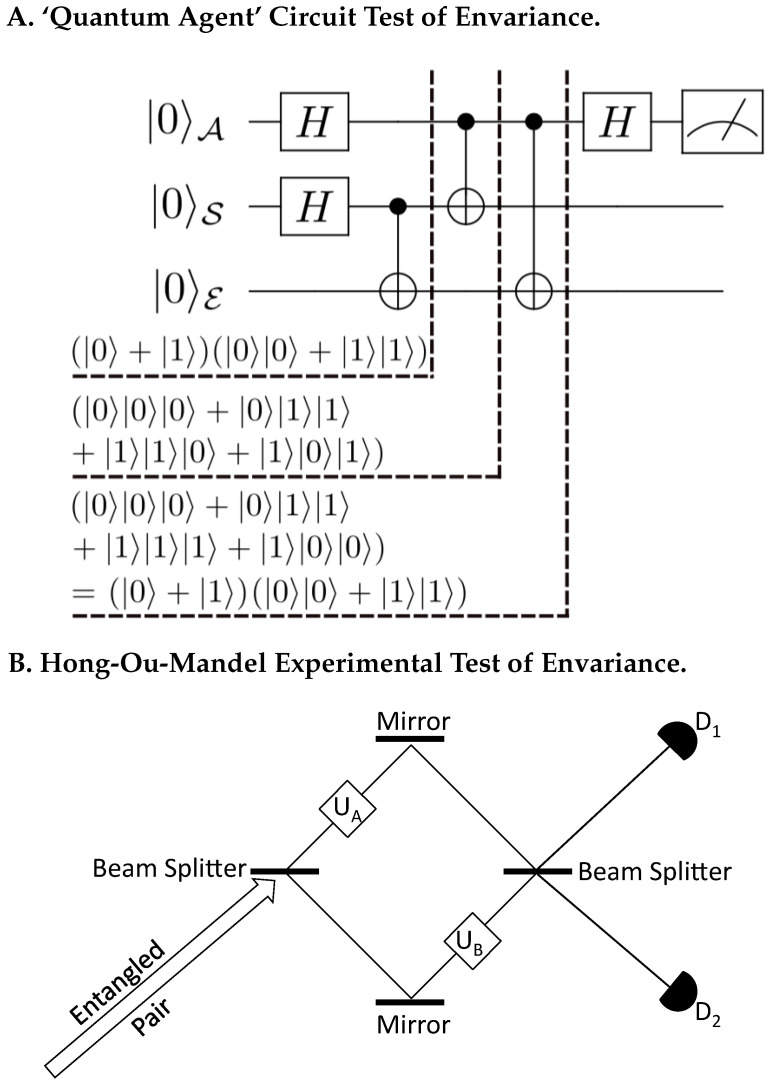
**Testing envariance:** A. “Quantum agent” circuit test of envariance: Hadamard gates put agent A and the system S in a superposition, and the leftmost c-not creates an entangled SE state. The next c-not performs a superposition of a swap on S (when A is in the state $|1\rangle$) or does nothing (when A is in $|0\rangle$). After the second c-not the SE pair is in a superposition of swapped an untouched, and state of A is mixed. However, after the last gate performs a conditional counterswap on E, the entangled state of SE should be restored, and the state of A should disentangle from SE and become pure again. B. Testing envariance with an entangled photon pair and a Hong-Ou-Mandel [115] interferometer. The initial state is $|0\rangle |1\rangle -|1\rangle |0\rangle$. As photons in the two arms of the interferometer are distinguishable (have “opposite” polarizations) they should not emerge together at the end of the interferometer. As a result, detectors should click in coincidence. However, after a swap $U_{A}=\left|0\rangle \;\langle 1\right|+\left|1\rangle \;\langle 0\right|$ on one of the photons changes the state into $|0\rangle |0\rangle -|1\rangle |1\rangle$, both photons should emerge at the same output. This will suppress coincidence clicks of the two detectors. This swap can be of course undone by a counterswap $U_{B}=\left|0\rangle \;\langle 1\right|+\left|1\rangle \;\langle 0\right|$ on the second photon, restoring the initial state, and restoring coincidences between the two detectors. Partial rotations of polarization and phase shifts can probe envariance for more general transformations.

**Figure 4 entropy-24-01520-f004:**
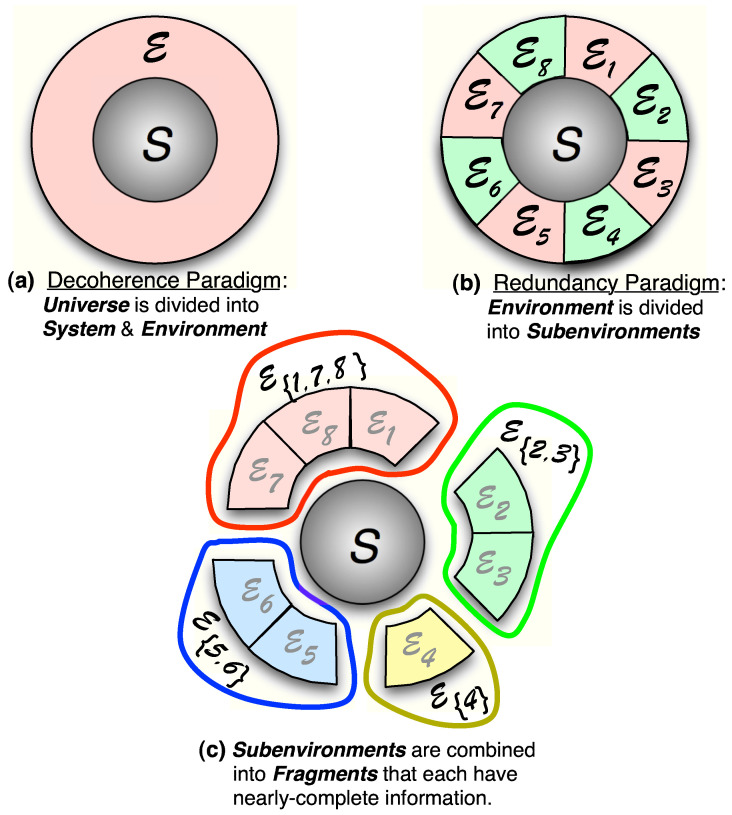
**Quantum Darwinism and the structure of the environment**. The decoherence paradigm distinguishes between a system (S) and its environment (E) as in (**a**), but makes no further recognition of the structure of E; it could as well be monolithic. In the environment-as-a-witness paradigm, we recognize subdivision of E into subenvironments—its natural subsystems, as in (**b**). The only essential requirement for a subsystem is that it should be individually accessible to measurements; observables corresponding to different subenvironments commute. To obtain information about the system S from its environment E one can then carry out measurements of *fragments*F of the environment—non-overlapping collections of its subsystems. Sufficiently large fragments of the environment that has monitored (and, therefore, decohered) S can often provide enough information to infer the state of S, by combining subenvironments as in (**c**). There are then many copies of the information about S in E, which implies that information about the “fittest” observables that survived monitoring by E has proliferated throughout E, leaving multiple qmemes, their (quantum) informational offspring. This proliferation of the information about the fittest (pointer) states defines quantum Darwinism. Multiple copies allow many observers to find out the state of S: Environment becomes a reliable witness with redundant copies of information about preferred observables, which accounts for the objective existence of preferred pointer states.

**Figure 5 entropy-24-01520-f005:**
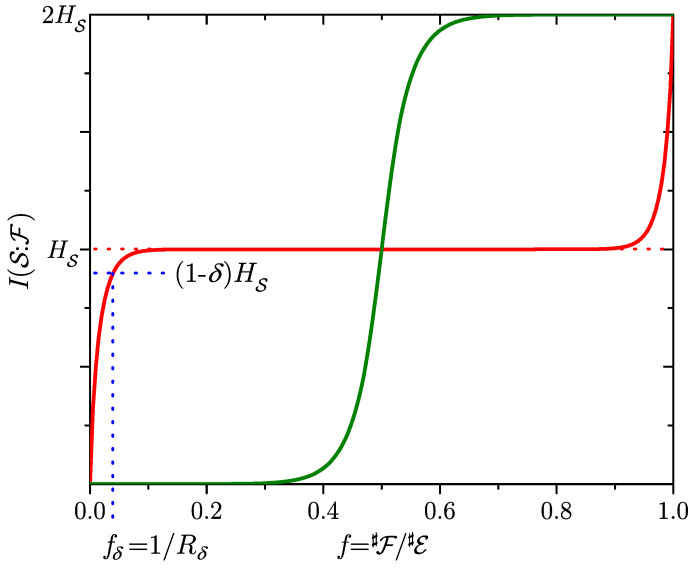
**Partial Information Plot (PIP) and redundancy $R_{\delta }$ of the information about a system S stored in its environment E**. When global state of SE is pure, mutual information that can be attributed to a typical fraction *f* of the environment is antisymmetric around $f=\frac{1}{2}$ and monotonic in *f*. For pure states picked out at random from the combined Hilbert space $H_{SE}$, there is very little mutual information between S and a typical fragment F smaller than about half of E. However, once threshold fraction $\frac{1}{2}$ is attained, nearly all information is in principle at hand. Thus, such random states (green line above) exhibit only small redundancy. (Strictly speaking redundancy is 2: The environment can be split into two halves, each supplying $H_{S}$ of information.) By contrast, states of SE created by decoherence (where the environment E monitors preferred observables of S) allow one to gain almost all (all but δ) of the information about S accessible through local measurements from a small fraction $f_{\delta }=1/R_{\delta }$ of E. The corresponding PIP (red line above) quickly asymptotes to $H_{S}$—the entropy of the system S (either preexisting or caused by decoherence)—which is all of the information about S available from measurements on either E of S. (More information about SE can be ascertained only through global measurements on S and a fragment F corresponding to more than half—indeed, nearly all—of E). $H_{S}$ is therefore the *classically accessible information*. As $\left(1-\delta \right)H_{S}$ of information can be obtained from a fraction $f_{\delta }=1/R_{\delta }$ of E, there are $R_{\delta }$ such fragments in E, and $R_{\delta }$ is the *redundancy* of the information about S. Large redundancy implies objectivity: The state of the system can be found out independently and indirectly (from fragments of E) by many observers, who will agree about it. In contrast to direct measurements, S will not be perturbed. Thus, *quantum Darwinism accounts for the emergence of objective existence*.

**Figure 6 entropy-24-01520-f006:**
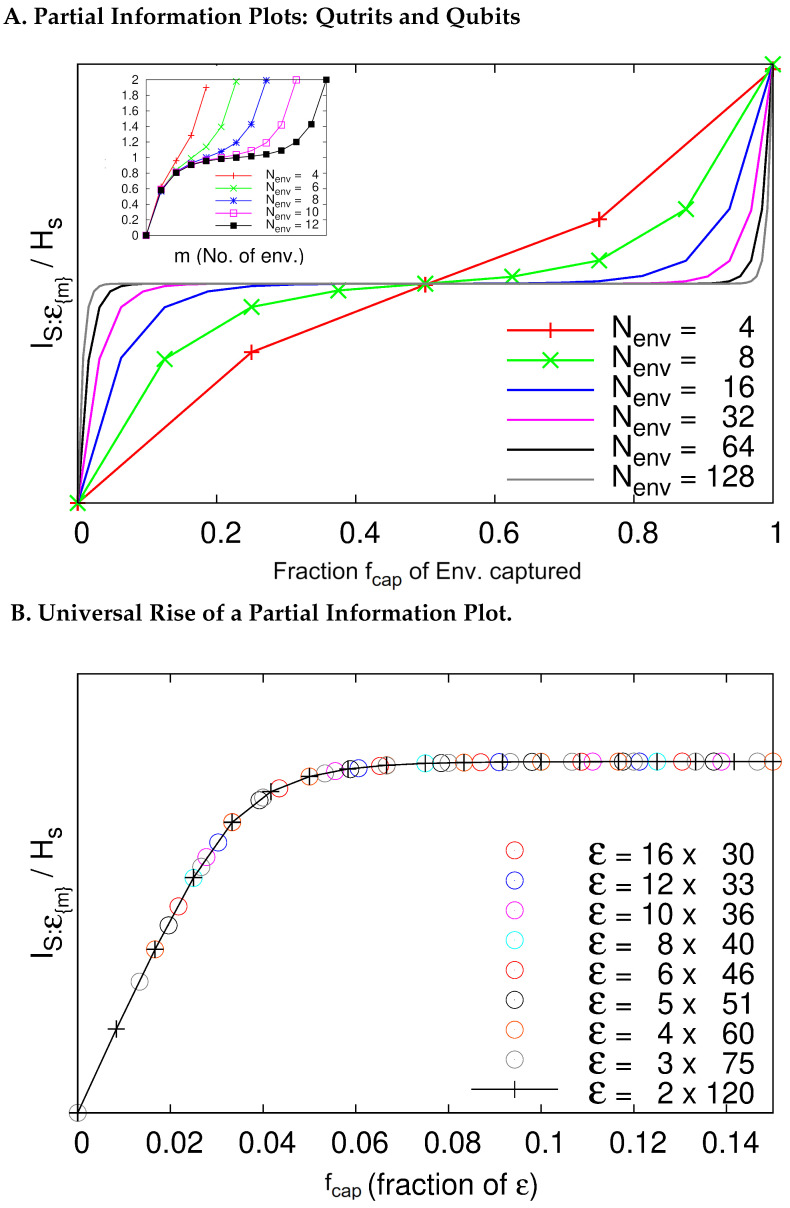
(**A**). **Partial information plots for pure decoherence:** (**A**) A qutrit coupled to N=4…128 qutrit environment plotted against the fraction $f={}^{♯}\;F/{}^{♯}\;E$ (see Blume-Kohout and Zurek, 2006 [152], for details). As the number of the environment subsystems increases, redundancy grows, which is reflected by the increasingly steep slope of the initial part of the plot. The inset [152] depicts rescaled mutual information of a qubit plotted against the number *m* of the environment qubits (rather than their fraction). Elongation of the plateau leads to the increase of redundancy: $R_{\delta }$ is the length of the PIP measured in units defined by the size (e.g., in the number of environment subsystems needed $m_{\delta }$, see Figure 5) of the part of the PIP that corresponds to I(S:F) rising from 0 to $\left(1-\delta \right)H_{S}$. (**B**). Universal rise of partial information plots of a qutrit coupled to nine different environments with different *subsystem size $d_{E}$* and cardinality ${}^{♯}\;E$, but with nearly identical total information capacity plotted as a function of their fractional information capacity.

**Figure 7 entropy-24-01520-f007:**
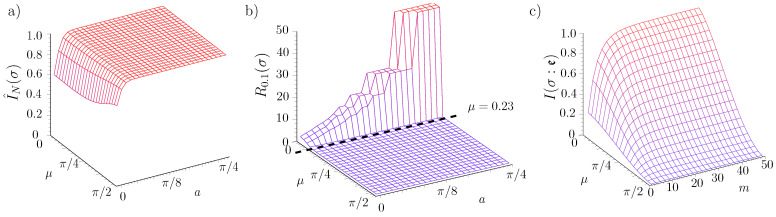
**Selection of the preferred observable in Quantum Darwinism in a simple model of decoherence** (Ollivier, Poulin, and Zurek, 2004 [149]).The system S, a spin-$\frac{1}{2}$ particle, interacts with N=50 qubits of E through the Hamiltonian $H=\sigma^{z}\sum_{k}g_{k}\sigma_{k}^{z}$ for a time *t*. The initial state of S⊗E is $\frac{1}{\sqrt{2}}\left(|0\rangle +|1\rangle \right)⊗{|0\rangle }_{E_{1}}⊗\ldots ⊗{|0\rangle }_{E_{N}}$. Couplings are selected randomly with uniform distribution in the interval (0,1]. All the plotted quantities are a function of the system’s observable $\sigma \left(\mu \right)=\cos\left(\mu \right)\sigma^{z}+\sin\left(\mu \right)\sigma^{x}$, where μ is the angle between its eigenstates and the pointer states of S—here the eigenstates of $\sigma^{z}$. (**a**) Information acquired by the optimal measurement on the whole environment, ${\hat{I}}_{N}\left(\sigma \right)$, as a function of the inferred observable σ(μ) and the average interaction action $\langle g_{k}t\rangle =a$. Nearly all information about every observable of S is accessible in the *whole* environment for any observables σ(μ) except when the action *a* is very small (so that E does not know much about S). Thus, complete imprinting of an observable of S in E is not sufficient to claim objectivity. (**b**) Redundancy of the information about the system as a function of the inferred observable σ(μ) and the average action $\langle g_{k}t\rangle =a$. It is measured by $R_{\delta =0.1}\left(\sigma \right)$, which counts the number of times 90% of the total information can be “read off” independently by measuring distinct fragments of the environment. For all values of the action $\langle g_{k}t\rangle =a$, redundant imprinting is sharply peaked around the pointer observable. Redundancy is a very selective criterion. The number of copies of relevant information is high only for the observables σ(μ) falling inside the theoretical bound (see Equation (4.29)) indicated by the dashed line. (**c**) Information about σ(μ) extracted by an observer restricted to local random measurements on *m* environmental subsystems. The interaction action $a_{k}=g_{k}t$ is randomly chosen in [0,π/4] for each *k*. Because of redundancy, pointer states—and only pointer states—can be found out through this far-from-optimal measurement strategy. Information about any other observable σ(μ) is restricted by the theorem discussed in this subsection (see also Refs. [130,149,151]) to be equal to the information about it revealed by the pointer observable $\Pi =\sigma^{z}$.

**Figure 8 entropy-24-01520-f008:**
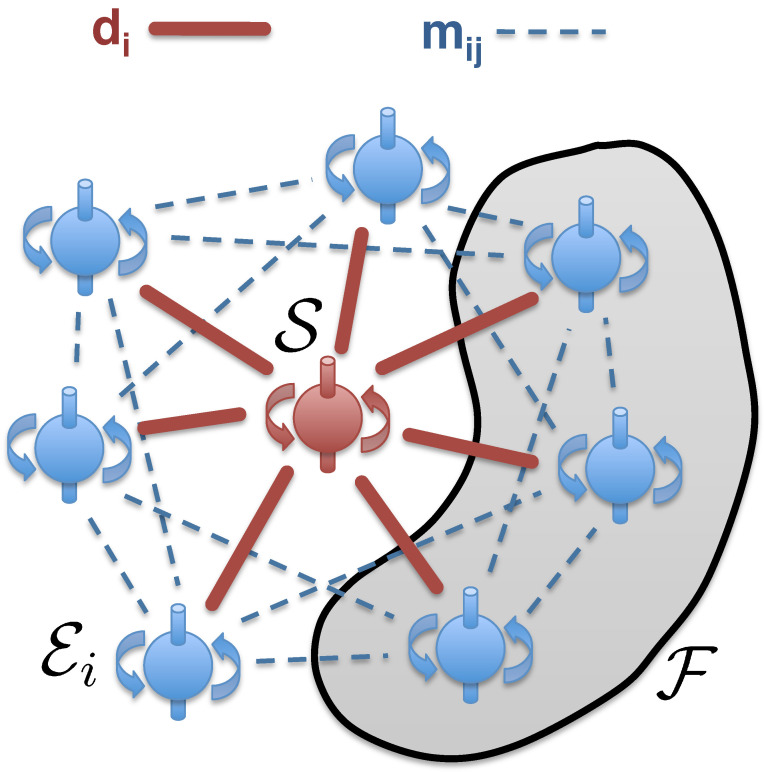
**Central spin model with interacting environment subsystems.** Decoherence is no longer pure: An environment of 16 spins $E_{i}$ coupled to a single system qubit S with Hamiltonian given by Equation (4.31b) is the basis of the results presented in this subsection and in Figure 9. As before, fragment F is a subset of the whole environment E. The couplings $d_{i}$ and $m_{ij}$ were selected from normal distributions with zero mean and standard deviations $\sigma_{d}=0.1$ and $\sigma_{m}=0.001$. Crucially, the interactions between S and the $E_{i}$ are much stronger than those within E. That is, $\sigma_{d}\gg \sigma_{m}$, so that information acquisition by the individual subsystems of the environment happens faster that the exchange of information between them. As a consequence, pure decoherence is initially a reasonable approximation. Eventually, however, interactions between the spins of the environment scramble the information so that fragments of E composed of individual spins reveal almost nothing about S.

**Figure 9 entropy-24-01520-f009:**
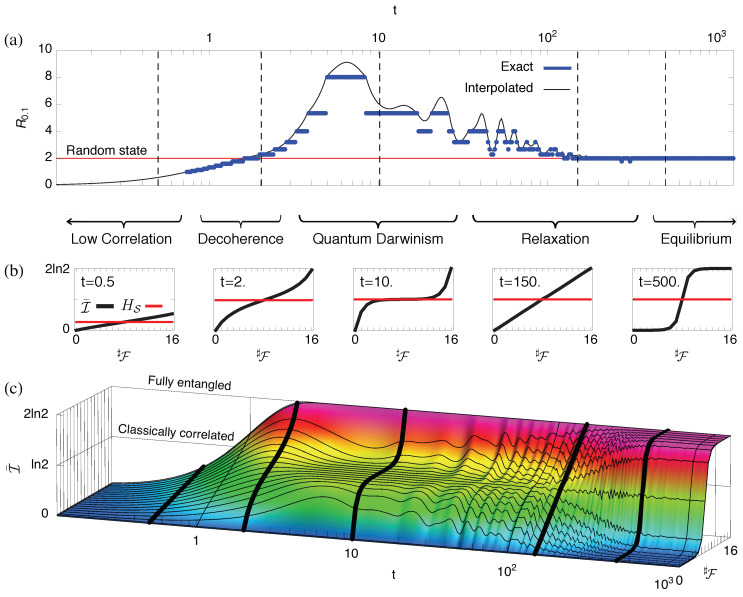
**Rise and fall of redundancy in the spin universe** of Figure 8 (see Riedel, Zurek, and Zwolak, 2012 [153]). (a) The redundancy $R_{\delta }$ is the number of fragments of E that provide (here, up to a fractional deficit δ=0.1) information about the system. The exact redundancy is supplemented with an estimate based on the interpolated value of $I(S:F_{f})$. The vertical dashed lines mark five instants. (b) The mutual information $I(S:F_{f})$ versus fragment size ${}^{♯}\;F$, and the entropy $H_{S}$ of the system at five instants corresponding to different qualitative behavior. (c) The mutual information $I(S:F_{f})$ versus fragment size ${}^{♯}\;F$ and time *t*. Thick black lines mark five instants. *Low correlation* (t=0.5) for small times means E “knows” very little about S. Each spin added to F reveals a bit more about S, resulting in the linear dependence $I(S:F_{f})$ on *f*. *Decoherence* (t=2) sets in near $\tau_{d}\equiv {\left(\sqrt{N}\sigma_{d}\right)}^{-1}=2.5$. By then the density matrix of S approaches a mixture of the two pointer states $|↑\rangle$ and $|↓\rangle$ singled out by the interaction and the state is approximately branching. Mutual information is still nearly linear in ${}^{♯}\;F$ and $R_{\delta }∼O\left(1\right)$. Mixing within E can be neglected as $t\ll \sigma_{m}^{-1}=1000$. *Quantum Darwinism* (t=10) is characterized by $I(S:F_{f})$ that rises to the plateau; the first few spins reveal nearly all classical information: Additional spins just confirm it. The quantum information (above the plateau) is still present in the global state but accessible only via an unrealistic global measurement of almost all of SE. After $t∼\sigma_{d}^{-1}=10$, few spins suffice to determine the state of S no matter how large *N* is, so $R_{\delta }∼N$. In the absence of the couplings $m_{ij}$ this (approximately pure decoherence) would persist. (For some environments, such as photons, this is indeed the case.) *Relaxation* (t=150) occurs near $t∼\tau_{m}\equiv {\left(\sqrt{N}\sigma_{m}\right)}^{-1}=250$. Mixing within the environment entangles any given spin’s information about S with the rest of E, reducing usefulness of the fragments. The mutual information plateau is destroyed, so redundancy plummets. *Equilibrium* (t=500) is reached for $t∼\sigma_{m}^{-1}=1000$, when the actions associated with interaction between spin pairs in the environment reach order unity. The state ceases to be branching. The mutual information plot takes the form of random states in the combined Hilbert space of SE. An observer can learn virtually nothing about the system unless almost half the environment is accessed.

**Figure 10 entropy-24-01520-f010:**
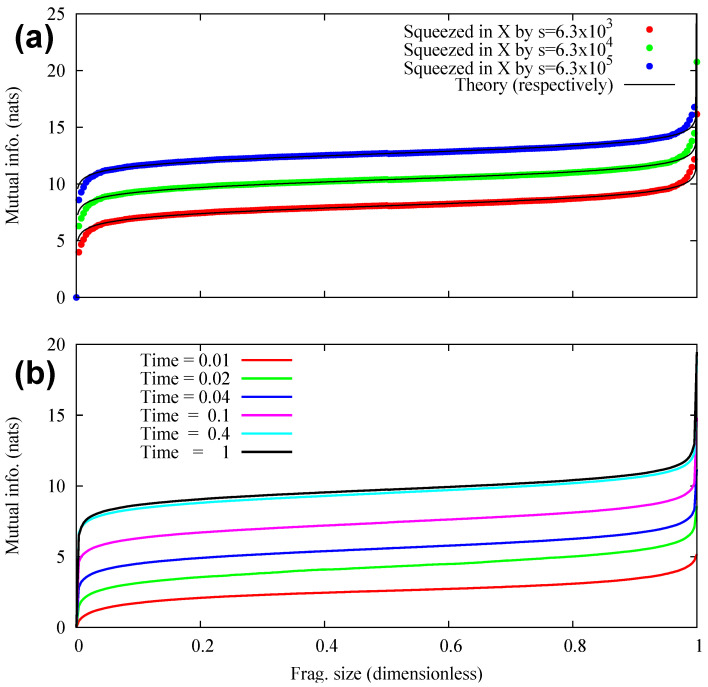
**Partial information plots for quantum Darwinism in quantum Brownian motion** (see Blume-Kohout and Zurek, 2008 [170]). The system S was initialized in an *x*-squeezed state, which decoheres as it evolves into a superposition of localized states. Plot (**a**) shows PIPs for three fully-decohered (t=4) states with different squeezing. Small fragments of E provide most of the information about S. Squeezing changes the amount of *redundant* information without changing the PIP’s shape. The numerics agree with the simple theory, Equation (4.36), discussed in Ref. [170]. Plot (**b**) tracks one state as decoherence progresses. PIPs’ shape is invariant; time only changes the redundancy of information.

**Figure 11 entropy-24-01520-f011:**
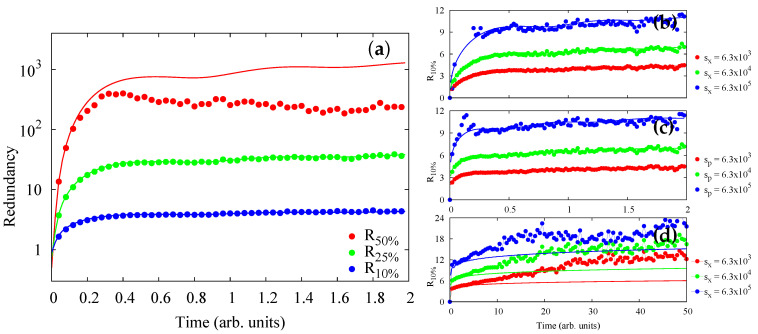
**Delocalized states of a decohering oscillator redundantly recorded by the environment** (Blume-Kohout and Zurek, 2008 [170]). Plot (**a**) shows redundancy $R_{\delta }$ vs. time for three information deficits δ, when the initial state of the system is a Gaussian squeezed in *x* by $s_{x}=6.3\times 10^{3}$. Plots (**b**–**d**) show $R_{10\%}$—redundancy of 90% of the available information—vs. initial squeezing ($s_{x}$ or $s_{p}$). Dots denote numerics; lines—theory. S has mass $m_{S}=1000$, $\omega_{S}=4$. E comprises oscillators with ω∈[0…16] and mass $m_{}=1$. The frictional coefficient is $\gamma =\frac{1}{40}$. Redundancy develops with decoherence: *p*-squeezed states [plot (**c**)] decohere almost instantly, while *x*-squeezed states [plot (**b**)] decohere as a $\frac{\pi }{2}$ rotation transforms them into *p*-squeezed states. Redundancy persists thereafter [plot (**d**)]; dissipation intrudes by $t∼O(\gamma^{-1})$, causing $R_{10\%}$ to rise above simple theory. Redundancy increases *exponentially*—as $R_{\delta }\approx s^{2\delta }$—with the information deficit [plot (**a**)]. So, while $R_{\delta }∼10$ may seem modest, δ=0.1 implies *very* precise knowledge (resolution of around 3 ground-state widths) of S. This is half an order of magnitude better than a recent results for measuring a micromechanical oscillator (see, e.g., Ref. [173]). At δ∼0.5—resolving $∼\sqrt{s}$ different locations within the wavepacket—redundancy reaches $R_{50\%}≳10^{3}$ (maximum numerical resolution of Ref. [170]).

**Figure 12 entropy-24-01520-f012:**
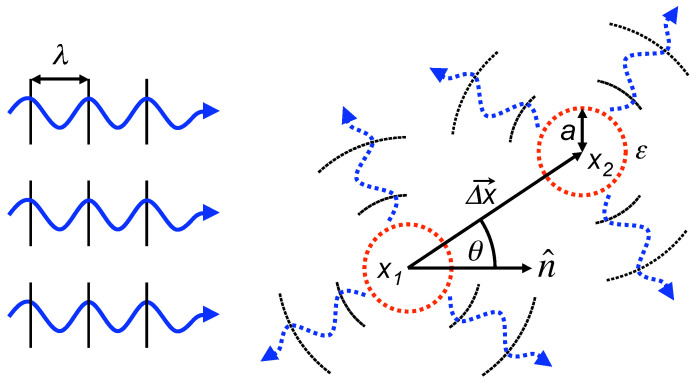
**Scattering photons from a dielectric sphere ‘Schrödinger’s cat’** (Riedel and Zurek, 2010; 2011 [158,181]). This is a realistic case of quantum Darwinism: Scattered photons carry multiple copies of the information about the location of S: A dielectric sphere of radius *r* and permittivity ϵ is initially in a superposition with separation $\Delta x=|x_{1}-x_{2}|$. This object—our systems S—scatters plane-wave radiation with thermally distributed photons of wavelength λ propagating in a direction $\hat{n}$ that makes an angle θ with the vector $\vec{\Delta x}$.

**Figure 13 entropy-24-01520-f013:**
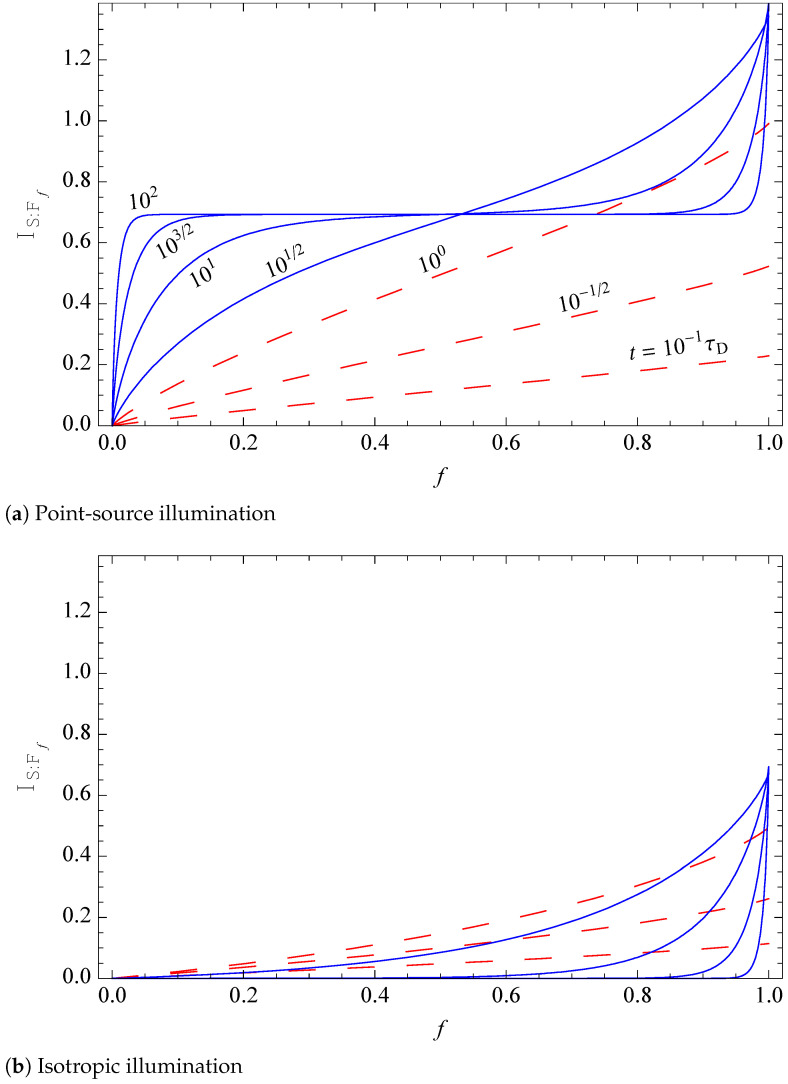
**Quantum Darwinism in a photon environment—the origin of the photohalo** (Riedel, and Zurek, 2010; 2011 [158,181]). The quantum mutual information $I(S:F_{f})$ vs. fragment size *f* at different elapsed times for an object illuminated by a point-source black-body radiation, and by an isotropic black-body radiation. (**a**) **For point-source illumination** individual curves are labeled by the time *t* in units of the characteristic time $\tau_{D}$, Equation (4.40). For $t\leq \tau_{D}$ (red dashed lines), the information about the system available in the environment is low. The linearity in *f* means each piece of the environment contains new, independent information. For $t>\tau_{D}$ (blue solid lines), the shape of the partial information plot indicates redundancy; the first few fragments of the environment give a lot of information, while the additional fragments only confirm what is already known. Such a photohalo contains many copies—multiple qmemes—of the record of the location of S. The remaining quantum information (i.e., mutual information above the plateau) is highly encrypted in the global state, in the sense that it can only be accessed by capturing almost all of E and measuring SE in the right way. (**b**) **For isotropic illumination** the same time-slicing is used as in (**a**) but there is greatly decreased mutual information because the directional photon states are “full” and cannot store more information about the state of the object.

**Figure 14 entropy-24-01520-f014:**
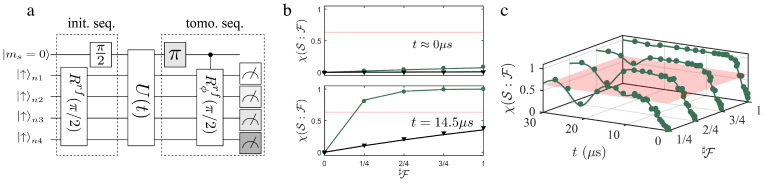
**The emergence of redundancy for an NV center decohered by its environment** (after Unden et al., 2019 [187]). (**a**) The measurement protocol starts with the initialization sequence and follows with a free decoherence-inducing evolution, *U*, for a duration of *t*. After initial polarization, two $\frac{\pi }{2}$-pulses transform the state into a product of $|+\rangle$ states which then evolve under the direct HF interaction between the NV center (the system S) and nuclear spins (the environment E). The tomography sequence follows. (**b**) Holevo information versus fraction size for a few different times. For small times, there are no correlations between S and E. However, as decoherence proceeds, information is transferred into E resulting in formation of a *classical plateau*. The plateau signifies the appearance of redundant information. When the data is not normalized to the initial degree of polarization (black), only the initial rise of information is seen. (**c**) Holevo information, χ(S:F), versus the environment fragment size ${}^{♯}\;F$ and free evolution time *t*. For small ${}^{♯}\;F$ one can see the initial rise in information with time followed by oscillations. This is due to information flowing into the fragment of the environment and then back into the system (i.e., environment spins will first gain information and then transfer it back to S). For larger fractions, however, one sees just a rise and a plateau with time. This is due to different interaction strengths with the environment spins, which favors one way information flow. The solid curves in (**b**,**c**) show the result of simulations with and without imperfect initial polarization. The dynamics in simulation are governed by the actual Hamiltonian. The semi-transparent red lines in (**b**) and the plane in (**c**) indicate an information deficit of 1/e, i.e., $I=\left(1-1/e\right)H_{S}$. Errors are smaller than the data points. (See Unden et al., 2019 [187], for further details.)

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
