# Peer review of "Quantum Theory of the Classical: Einselection, Envariance, Quantum Darwinism and Extantons"

_entropy, 2022, doi:10.3390/e24111520_

Round 1

Reviewer 1 Report

Please take a look at the attached pdf file for my comments and suggestions. 

Author Response

Thank you for the positive assessment. 

I went carefully through the whole paper and made changes, primarily perturbative, and primarily in reply to the comments of the two referees as well as to additional comments from several colleagues who read the manuscript.

There are now two substantial additions: 

(i)             Subsections 3.4.1 and 3.4.2 on additivity of probabilities and its consequences.

(ii)            Subsection 4.5 introducing the concept of an “extanton” (this neologism appears already in the title).

I believe these additions are useful. 

I appreciate the suggestion to expand the discussion of the experimental efforts. However, I have not followed the (optional) suggestion to add to Fig. 15 the illustration of the other quantum Darwinism experiments. I have a feeling all of the experiments to date (while useful) are still rather primitive. (This includes the one illustrated in Fig. 15, although the decoherence there is “natural”.)

Reviewer 2 Report

This is an excellent review of research aimed at understanding how the classical world we perceive emerges from the counterintuitive laws of quantum mechanics (QM).  In textbook presentations, the postulations of QM include the controversial collapse postulate (iv, in the notation of the paper), supplemented by Born's rule (v).  The author summarizes about two decades of research, in which he has played a (arguably, THE) pioneering role, leading to the conclusion that postulates iv and v are in fact unnecessary, as they can be obtained -- not obviously, but through careful analysis -- from the non-controversial postulates (o) - (iii).  The bottom line is that the measurement process can be described without resorting to postulates (iv) and (v).  The meat of the paper is found in Secs. II - IV, dealing with the collapse postulate, Born's rule and Quantum Darwinism, respectively.

This review offers a terrific introduction into this progress, and I recommend that it be accepted for publication in Entropy.

Below, I list a number of questions / comments that the author may wish to address, followed by a list of minor typos.

Page 7, Eq. 2.1:

The author supposes the existence of states |s_k> that satisfy Eq. 2.1.  But what is assumed about the initial state of the environment, |eps_0> ?  Is Eq. 2.1 being assumed valid for any choice of |eps_0>, or rather is it assumed that there exists a particular |eps_0>, together with |s_1>, |s_2>, ... such that Eq. 2.1 is satisfied?

Page 8, left column:

The no-cloning theorem is invoked.  It would be useful to briefly (one or two sentences) summarize what this theorem says.  A citation to Wootters and Zurek, 1982, would be appropriate.  This reference appears in the bibliography, but I don't think it is actually cited in the text.

Page 10, left column, near the bottom:

"a set of orthogonal states defines a Hermitian operator when supplemented with real eigenvalues".

This is true only if this set spans the Hilbert space (or am I missing something)?  Is it assumed somewhere that the states |s_1>, |s_2>, ... form a complete basis for the system's Hilbert space?

Page 11, right column: What is the meaning of the tilde immediately after Eq. 2.11?

Page 21, footnote 6:

I'm having difficulties following the argument here.  "One can combine two of these states to form a new merged state, which still has the same coefficient of 0".  I assume that "combining" two states |a> and |b> means taking a linear superposition to form a "new merged state" |c>, but in that case there will be an orthgonal state |d> such that |c> and |d> span that same subspace as |a> and |b>.  So how has the total number of states (with zero eigenvalue) been reduced from n to n-1?

Page 24, left column:

Footnote 10 seems redundant, given the sentence "Without normalization, amplitudes we are trying to deduce would have no meaning", which appears just before the last paragraph.

Page 28, left column, near the top:

The error bars that appear in the values 99.66+-0.04 and 99.963+-0.005 seem to suggest that these values are statistically significantly different from 100%.  E.g. 99.66 is nearly ten standard deviations away from 100, if the standard deviation is 0.04.  I'm not sure this is worthy of a comment (or an edit), but I noticed it.

Page 32, Eq. 4.12b and immediately below:

Is there a reason why the notation |pi_k> is used, rather than |s_k>?

Page 33, first paragraph of section 3:

The notation is a bit confusing.  Initially the subscript on F is k, which suggests the subscript indicates an index.  But then in Eq. 4.15 the subscript becomes delta, which denotes a real value rather than an index.  This should be clarified.

Page 36, left column, sixth line:

"However, when S is initially pure ..."

Then a few lines below that:

"In the 'opposite' case -- when S is initially pure ..."

This seems to be a contradiction, and should be rewritten for clarity.

Page 45, left column, near the bottom:

"In our case, the coupling between the spins of the environment is significantly stronger than the couplings between the spins of E."

I'm not sure how to interpret this, maybe it's a typo.

Typos:

Page 9, right column, a few lines above Eq. 2.7:

one is lead to  ->  one is led to

Page 15, top of left column:

"usual suspects  ->  "usual suspects"

Page 20, footnote 5:

as the allow  ->  as they allow

Page 22, Eq. 3.8a:

The final |0'> (just before the equals sign) should be |2'>

Page 24, just above Eq. 3.15a:

the number of equiprobable sequences -> the square root of the number of equiprobable sequences

Page 29, middle of the caption of Fig. 4:

of of  ->  of

Page 31, footnote 13:

who knows the A A  -> ??

Page 35, right column, near the center:

and its reminder  ->  and its remainder

Page 36, right column, 8 lines from the top:

This situations  ->  This situation

Page 49, right column, last line:

reverse reverse  ->  reverse

Page 53, left column, near the top:

run into problems  ->  runs into problems

Page 54, left column, top line:

he is doesn't know  ->  he doesn't know

Page 54, left column, first paragraph of section B:

interretation  ->  interpretation

Page 54, right column, first paragraph of section C:

"clock time  ->  "clock time"

Page 56, left column:

possibly because are cold-blooded  ->  possibly because they are cold-blooded

Page 57, right column, last paragraph before acknowledgements:

inevitability quantum jumps  ->  inevitably of quantum jumps

exist to address is it  ->  ??

Author Response

Thank you for the assessment. Please see attachment
